# DATA RELIABILITY SCORING

## ABSTRACT

How can we assess the reliability of a dataset without access to ground truth? We introduce the problem of *reliability scoring* for datasets collected from potentially strategic sources. The true data are unobserved, but we see outcomes of an unknown statistical experiment that depends on them. To benchmark reliability, we define ground-truth–based orderings that capture how much reported data deviate from the truth. We then propose the *Gram determinant score*, which measures the volume spanned by vectors describing the empirical distribution of the observed data and experiment outcomes. We show that this score preserves several ground-truth-based reliability orderings and, uniquely up to scaling, yields the same reliability ranking of datasets regardless of the experiment – a property we term experiment agnosticism. Experiments on synthetic noise models, CIFAR-10 embeddings, and real employment data demonstrate that the Gram determinant score effectively captures data quality across diverse observation processes.

## 1 INTRODUCTION

Reliable data can effectively inform decision-making. For example, vehicle condition and driving behavior data help insurance companies set policies; investor's positions guide regulators in adjusting financial market rules; and during the COVID-19 pandemic, case numbers were used by governments to allocate medical resources. Yet, such data are typically reported by people. They can be noisy, and more importantly, strategically or maliciously distorted. Direct verification is often impossible or impractical. This raises a central question: how can we tell whether a dataset is reliable? Answering this would greatly enhance the value of data-driven methods for decision-making.

Without further knowledge, this question is unresolvable. But in practice, we often have access to data that are related to the private data in question. For instance, insurance company may use telematic devices–albeit imperfect–to estimate vehicle condition; regulators can observe trading volumes correlated with investors' positions; and governments track COVID mortality numbers linked to true case counts through disease fatality rates. Such auxiliary observations can provide useful information to assess how well the reported data are consistent with the unobservable ground truth.

In this paper, we initiate the study of reliability scoring for datasets collected from potentially strategic or noisy sources. Although the underlying truth remains unknown, we assume access to outcomes of unknown statistical experiments that depend on it. Our contributions include:

- We formalize the problem of reliability scoring from observations generated by unknown experiments. (Section 2)
- We introduce ground-truth-based dataset reliability orderings as benchmarks for evaluating reliability scores. (Section 2.3)
- We propose a novel reliability measure, the **Gram Determinant Score**, along with its kernel variant, which preserves several ground-truth-based dataset reliability orderings under certain conditions. Moreover, we show that the Gram Determinant Score is, up to scaling, the unique reliability score that produces the same dataset ranking for all experiments – a property we term *experiment agnosticism*. (Section 4)
- We analyze the limitations of reliability scoring and show that the conditions under which the Gram Determinant Score preserves reliability orderings are nearly tight. (Section 3)
- We empirically validated the Gram Determinant Score using synthetic data, CIFAR-10 image dataset, and employment data. (Section 5)

The Gram Determinant Score admits a geometric interpretation: it measures the volume of the parallelepiped spanned by the joint distribution of the reported data and the experiment outcomes. As the reported data deviate further from the truth, this volume decreases. (Figure 1)

## 1.1 RELATED WORK

Early frameworks categorize data reliability into intrinsic, contextual, accessibility, and representational dimensions. (Wang & Strong, 1996; Priestley et al., 2023) Our work focuses on intrinsic reliability—the extent to which reported data match the true data—using auxiliary observations.

Our approach is inspired by information elicitation, which designs scoring mechanisms that incentivize truthful reporting. A key distinction is our emphasis on preserving ordinal relationships: assigning higher scores to more reliable data. Traditional elicitation instead focuses solely on ensuring that truthful reporting is strictly optimal among alternatives. Information elicitation has two main settings (1) when the scoring mechanism can access the ground truth, e.g., proper scoring rules for predictions of future observable events (Gneiting & Raftery, 2007; Osband, 1985; Lambert et al., 2008; Frongillo & Kash, 2015; Liu & Chen, 2018); and (2) peer prediction mechanisms, which do not have access to ground truth but rely on multiple agents' reports (Miller et al., 2005; Dasgupta & Ghosh, 2013). The most relevant work is Kong (2024), which introduces determinant mutual information and inspires our Gram Determinant Score. We provide a more detailed comparison with Kong (2024) in the Appendix. A recent works use Shannon (pointwise) mutual information to evaluate dataset and introduce Blackwell ordering to compare reported dataset. (Zheng et al., 2025)

Traditional statistical approaches (Huber, 2004; Meeker et al., 2021) often assess reliability under distributional assumptions. In contrast, our method evaluates reliability agnostic to the underlying distribution. There are several general-purpose score that measures the stochastic relationship between random variables, e.g., KL-divergence (Kullback & Leibler, 1951), $f$-divergence (Csiszár, 1972), determinant (Zou & Adams, 2012; Xu et al., 2019), PCA (Amiri et al., 2022). But they often lack clear connections to standard, interpretable criteria such as accuracy or data integrity. On the other hand, one line of data valuation focus on task-dependent utility—quantifying the value of a dataset or individual sample with respect to a specific objective. Examples include value of information in decision theory (Howard, 2007; Chen & Waggoner, 2016), influence-based valuation (Cook & Weisberg, 1980; Koh & Liang, 2017), and data Shapley (Ghorbani & Zou, 2019). In contrast, our reliability scoring aims to evaluate datasets in a task-agnostic and experiment-agnostic manner.

Other related areas include learning with noisy labels (Natarajan et al., 2013), which typically assumes that reports are corrupted by independent noise. Some works (e.g., (Liu & Guo, 2020)) relax this by allowing unknown noise, but our setting is more general: auxiliary observations may lie in an entirely different space. Anomaly detection (Chandola et al., 2009) addresses distribution shifts, but focuses on adaptive detection rather than reliability scoring. Finally, reliability theory primarily studies system robustness to failure (Gnedenko et al., 2014), a concept distinct from data reliability.

## 2 MODEL

In this section, we introduce the problem of designing data reliability scores to assess how much a dataset deviates from its inaccessible ground truth. To benchmark reliability, we propose ground-truth-based reliability orderings—partial orders over datasets that compare their relative deviations from the same true dataset. The ideal goal of a reliability score is to preserve these orderings, assigning higher scores to datasets that more faithfully reflect the true data.

## 2.1 BASIC SETUP

There is a single data source (an agent) who has access to a set of *true data* $\boldsymbol{x} = (x_1, \ldots, x_N)$ of size $N$.[1] The agent provides *reported data* $\hat{\boldsymbol{x}} = (\hat{x}_1, \ldots, \hat{x}_N)$, which can potentially be different from $\boldsymbol{x}$. Let $\mathcal{X} = [d]$ be the set of $d$ possible data values. Thus, $x_n \in \mathcal{X}$ and $\hat{x}_n \in \mathcal{X}$ for all $n$.

Our goal is to evaluate how reliably the reported data $\hat{\boldsymbol{x}}$ reflects the true data $\boldsymbol{x}$. Although $\boldsymbol{x}$ is unobserved, we have access to additional observable data $\boldsymbol{y} = (y_1, \ldots, y_N)$, called *observations*, which

---

[1] $\boldsymbol{x}$ is non-time-series data. Hence, the order of the data within the set is not important.

are indirectly related to $x$. The observation space $\mathcal{Y}$ may differ from $\mathcal{X}$. We model the relationship between $y$ and $x$ as an unknown, statistical *experiment*, represented by a column-stochastic matrix $P = (P_x)_{x \in \mathcal{X}}$, where each column $P_x$ is a distribution over $\mathcal{Y}$. Given true data $x = (x_1, \ldots, x_N)$, observations are generated according to $P$ with $y_n \sim P_{x_n}$ independently for all $n \in [N]$. We denote this generation as $y \sim P(x)$.

For instance, $x$ may represent patients' true disease state (having or not having the disease), $\hat{x}$ the diagnoses reported by a hospital to an insurance database for reimbursement, and $y$ the results of inexpensive blood tests or imaging biomarkers correlated with the disease. As another example, in an image-labeling dataset, $x$ denotes the true image labels, while $\hat{x}$ are the reported labels. The observations $y$ may come from encoder representations, such as those produced from contrastive learning methods (Zbontar et al., 2021).

Having access to $y$ and knowing that $y$ are generated by unknown experiment $P$, we want to design a *reliability score* $S : \mathcal{X}^N \times \mathcal{Y}^N \to \mathbb{R}$ such that, if a dataset $\hat{x}$ aligns with $x$ more than a dataset $\hat{x}'$ does, dataset $\hat{x}$ receives a higher reliability score in expectation than dataset $\hat{x}'$: $\mathbb{E}_{y \sim P(x)}[S(\hat{x}, y)] > \mathbb{E}_{y \sim P(x)}[S(\hat{x}', y)]$. However, to formalize this goal, we will first need metrics to quantify how much reported data align with the true data. In Section 2.2, we describe how to use a misreport matrix to represent the relationship between reported data and true data. Then, we introduce four notions of ground-truth-based reliability ordering of reported datasets in Section 2.3 before returning to define the ideal goal of reliability scoring in Section 2.4.

## 2.2 REPRESENTATION OF DATASET RELATIONSHIPS

The relationship between the true dataset $x$ and a reported dataset $\hat{x}$ can be summarized by the size of the datasets $N$ and a $d \times d$-dimension *misreport matrix* $Q$ where each entry $Q(i, j)$ represents the frequency of misreporting true value $i$ in $x$ for value $j$ in $\hat{x}$:

$$Q(i, j) = \frac{1}{N} \sum_{n=1}^{N} \mathbf{1}[x_n = i, \hat{x}_n = j].$$

$Q$ is the joint frequency of true data and reported data. It can be further decomposed into marginal frequency and conditional frequency. Let $q_x(i) = \frac{1}{N} \sum_{n=1}^{N} \mathbf{1}[x_n = i]$ and $q_{\hat{x}}(i) = \frac{1}{N} \sum_{n=1}^{N} \mathbf{1}[\hat{x}_n = i] \; \forall i \in \mathcal{X}$, the marginal frequency matrices are defined as $d \times d$ diagonal matrices $Q_x, Q_{\hat{x}}$ with $q_x$ and $q_{\hat{x}}$ respectively as diagonal and zeros everywhere else. We then define column-stochastic matrices $Q_{\hat{x}|x}, Q_{x|\hat{x}}$ for conditional frequency, where for all $i, j \in \mathcal{X}$, $Q_{\hat{x}|x}(i, j) = \frac{\sum_n \mathbf{1}[x_n = j, \hat{x}_n = i]}{\sum_n \mathbf{1}[x_n = j]}$ and $Q_{x|\hat{x}}(i, j) = \frac{\sum_n \mathbf{1}[x_n = i, \hat{x}_n = j]}{\sum_n \mathbf{1}[\hat{x}_n = j]}$. Hence,

$$Q = (Q_{\hat{x}|x} Q_x)^{\mathsf{T}} \text{ and } Q = Q_{x|\hat{x}} Q_{\hat{x}}. \tag{1}$$

These frequency matrices exist for any pairs of $x$ and $\hat{x}$, but $Q$, $Q_x$, $Q_{x|\hat{x}}$, and $Q_{\hat{x}|x}$ are not observed because $x$ is unknown. We introduce them to help us quantify a $\hat{x}$'s deviation from $x$. In this paper, we use $\mathcal{Q}$ to denote a set of misreporting matrices, and also, abusing the notation, use $\mathcal{Q}$ to refer pairs of $x, \hat{x}$ so that the associated misreport matrix is in $\mathcal{Q}$.

Given a statistical experiment $P$, the matrix product $PQ$ is a $|\mathcal{Y}| \times |\mathcal{X}|$ matrix representing the joint distribution of observations and reported data, with element at $(k, i)$ being $\Pr(y = k, \hat{x} = i)$. The matrix product $PQ_x$ is a $|\mathcal{Y}| \times |\mathcal{X}|$ matrix representing the joint distribution of observations and true data, with element at $(k, i)$ be $\Pr(y = k, x = i)$. While both $PQ$ and $PQ_x$ are unknown, $\hat{x}$ and $y$ are samples from distribution $PQ$, which are all that we can leverage in reliability scoring.

## 2.3 RELIABILITY ORDERINGS OF DATASETS

To compare the reliability of reported datasets relative to the true data $x$, some preference on relative dataset reliability is needed. While the preference may depend on applications, we suggest three natural strict partial orderings of reported datasets, each defined with respect to true data $x$.

1. **Exact Match Ordering**: $\hat{x} \succ^x_{\text{EXACT}} \hat{x}'$ if $\hat{x} = x$ but $\hat{x}' \neq x$. Equivalently, $Q'_{\hat{x}|x} \neq \mathbb{I}$ and $Q_{\hat{x}|x} = \mathbb{I}$. This ordering picks up only complete agreement with the true data, and does not differentiate any pair of reported datasets if neither agrees with the true data. This order captures the notion of data integrity. (Kim & Spafford, 1994)

2. **Blackwell dominant ordering**: $\hat{\boldsymbol{x}} \succ^{\boldsymbol{x}}_{\text{Blackwell}} \hat{\boldsymbol{x}}'$ if $\boldsymbol{Q}$ and $\boldsymbol{Q}'$ are both invertible and (row) diagonally maximized (i.e. $\boldsymbol{Q}(i,j) \leq \boldsymbol{Q}(i,i)$ and $\boldsymbol{Q}'(i,j) \leq \boldsymbol{Q}'(i,i)$ for all $i$ and $j$) and there exists a (column) stochastic matrix $\boldsymbol{T} \neq \mathbb{I}$ so that $\boldsymbol{T} \boldsymbol{Q}_{\hat{\boldsymbol{x}}|\boldsymbol{x}} = \boldsymbol{Q}'_{\hat{\boldsymbol{x}}|\boldsymbol{x}}$ (equivalently, $\boldsymbol{Q}' = \boldsymbol{Q}\boldsymbol{T}^\intercal$ by Eq. (1)). This ordering captures that post-processing that transforms $\hat{\boldsymbol{x}}$ into $\hat{\boldsymbol{x}}'$ only reduces the reliability or informativeness of the data. (Blackwell, 1953). In particular, this ordering ensures that the true data ranks the highest, and uninformative random reports ranks the lowest.

3. **dist ordering**: Given a distance function $\text{dist} : \mathcal{X} \times \mathcal{X} \to \mathbb{R}$ so that $\text{dist}(x,x') = \text{dist}(x',x)$, $\text{dist}(x,x) = 0$ and $\text{dist}(x,x') > 0$ if $x \neq x'$,[2] we say $\hat{\boldsymbol{x}} \succ^{\boldsymbol{x}}_{\text{dist}} \hat{\boldsymbol{x}}'$ if $\sum_{n=1}^{N} \text{dist}(\hat{x}_n, x_n) < \sum_{n=1}^{N} \text{dist}(\hat{x}'_n, x_n)$. This ordering captures the coordinate-wise difference between true and reported data. We may also consider a weaker notion, $\alpha$-dist *ordering* with some $\alpha \in (0,1]$. We say $\hat{\boldsymbol{x}} \succ^{\boldsymbol{x}}_{\text{dist},\alpha} \hat{\boldsymbol{x}}'$ if $\sum_{n=1}^{N} \text{dist}(\hat{x}_n, x_n) < \alpha \sum_{n=1}^{N} \text{dist}(\hat{x}'_n, x_n)$. In other words, the distance between $\hat{\boldsymbol{x}}$ and $\boldsymbol{x}$ is at least a factor of $\alpha$ smaller than that of $\hat{\boldsymbol{x}}'$ and $\boldsymbol{x}$, in order to rank $\hat{\boldsymbol{x}}$ and $\hat{\boldsymbol{x}}'$.

A special case of $\text{dist}$ ordering is **_Hamming ordering_**, when $\text{dist}$ is the discrete metric $\text{dist}(i,j) = \mathbf{1}[i \neq j]$ for all $i,j \in \mathcal{X}$. We say $\hat{\boldsymbol{x}} \succ^{\boldsymbol{x}}_{\text{Hamming}} \hat{\boldsymbol{x}}'$ if $\sum_{n=1}^{N} \mathbf{1}[\hat{x}_n \neq x_n] < \sum_{n=1}^{N} \mathbf{1}[\hat{x}'_n \neq x_n]$ or, equivalently, $\text{Tr}(\boldsymbol{Q}) > \text{Tr}(\boldsymbol{Q}')$. Hamming ordering counts the number of disagreements between the true data and the reported data. (Hamming, 1950)

Blackwell dominant ordering is intentionally defined for a subset of misreport matrices: $\boldsymbol{Q}, \boldsymbol{Q}' \in \mathcal{Q}_{\text{reg}}$, which is the collection of invertible and (row) diagonally maximal matrices so that $\boldsymbol{Q}(i,j) \leq \boldsymbol{Q}(i,i)$ for all $i$ and $j$. Intuitively, diagonally maximal requires the true data values dominate any misreport in a reported dataset. Restriction to $\mathcal{Q}_{\text{reg}}$ is necessary for Blackwell dominant ordering to be a strict partial ordering. In Appendix B, we formally prove that all above orderings are strict partial orders. In particular, the Blackwell ordering fails to be strict if either invertibility or diagonal maximal of $\boldsymbol{Q}$ and $\boldsymbol{Q}'$ is not enforced.[3]

These orderings reflect different ways of measuring the extent of misreporting, with some providing finer distinctions between datasets than others. Formally, given a set of misreport matrices $\mathcal{Q}$, partial ordering $\succ_1$ *refines* partial ordering $\succ_2$ on $\mathcal{Q}$ if $\forall \boldsymbol{x}, \hat{\boldsymbol{x}}, \hat{\boldsymbol{x}}'$ with associated misreport matrices $\boldsymbol{Q}, \boldsymbol{Q}' \in \mathcal{Q}$, $\hat{\boldsymbol{x}} \succ^{\boldsymbol{x}}_2 \hat{\boldsymbol{x}}' \Rightarrow \hat{\boldsymbol{x}} \succ^{\boldsymbol{x}}_1 \hat{\boldsymbol{x}}'$. The following proposition shows that Blackwell dominant ordering refines exact-match ordering, and Hamming ordering refines Blackwell dominant ordering. The proofs are in Appendix B

**Proposition 2.1** (Refinement). *The reliability orderings have the following relationships:*

1. *Blackwell dominant ordering refines the exact match ordering on $\mathcal{Q}_{reg}$.*

2. *Hamming ordering refines the Blackwell dominant ordering on $\mathcal{Q}_{reg}$.*

3. *For all $\alpha \geq \alpha'$ and distance function $\text{dist}$, $\alpha$-dist ordering refines $\alpha'$-dist ordering.*

## 2.4 RELIABILITY SCORING

We now return to formally define the ideal goals of reliability scoring.

**Definition 2.2.** *Given a reliability ordering $\succ$ over $\mathcal{X}^N$, a reliability score $S : \mathcal{X}^N \times \mathcal{Y}^N \to \mathbb{R}$ preserves partial ordering $\succ$ under experiment $\boldsymbol{P}$, if for all $\boldsymbol{x}, \hat{\boldsymbol{x}}, \hat{\boldsymbol{x}}' \in \mathcal{X}^N$ with $\hat{\boldsymbol{x}} \succ^{\boldsymbol{x}} \hat{\boldsymbol{x}}'$ we have*

$$\mathbb{E}_{\boldsymbol{y} \sim P(\boldsymbol{x})}[S(\hat{\boldsymbol{x}}, \boldsymbol{y})] > \mathbb{E}_{\boldsymbol{y} \sim P(\boldsymbol{x})}[S(\hat{\boldsymbol{x}}', \boldsymbol{y})]. \tag{2}$$

Given a set of experiments $\mathcal{P}$, a set of misreport matrices $\mathcal{Q}$, and a minimum size of reported datasets $N_0 \in \mathbb{N}$, we say that a reliability score preserves $\succ$ under $\mathcal{P}, \mathcal{Q}$ and $N_0$ if Eq. (2) holds for all

---

[2]Any metric, e.g., $\ell_2$-norm, satisfies the above three conditions. Additionally, a function with these properties is often referred to as a semimetric.

[3]Instead of $\mathcal{Q}_{\text{reg}}$, we can alternatively require (a) $\boldsymbol{Q}$ and $\boldsymbol{Q}'$ are invertible and (b) $\boldsymbol{T}$ is not a permutation matrix (i.e. $\boldsymbol{Q}\boldsymbol{T}^\intercal$ is not a permutation of columns of $\boldsymbol{Q}$) to ensure that Blackwell dominant ordering is strict. However, this set of conditions does not support the result in Proposition 2.1 that Hamming ordering refines the Blackwell dominant ordering.

$P \in \mathcal{P}$ and tuples $x, \hat{x}, \hat{x}'$ of size at least $N_0$ with $\hat{x} \succ^x \hat{x}'$ and $Q, Q' \in \mathcal{Q}$. We further call $S$ *asymptotically* preserves $\succ^{\cdot}$ under $\mathcal{P}, \mathcal{Q}$, if for all $P \in \mathcal{P}$ and $Q, Q' \in \mathcal{Q}$ there exists $N_0$ so that $S$ preserve $\succ^{\cdot}$ under $P$ for all $x, \hat{x}, \hat{x}'$ of size at least $N_0$ with $\hat{x} \succ^x \hat{x}'$ and misreport matrices $Q, Q'$.

In the remainder of the paper, we study the problem of designing reliability score $S(\hat{x}, y)$ that preserves partial orderings of interest. We refer to this as the detail-free setting, since scoring does not rely on knowledge of $Q$ or $P$. For the analysis, however, we also consider a partial-knowledge setting, where the score can take the joint distribution $PQ$ as input, $S(PQ)$. This setting serves as a technical tool: it allows us to establish impossibility results (Section 3) and to illustrate the core ideas underlying our approach to detail-free scoring (Section 4).

## 3 IMPOSSIBILITY RESULTS FOR RELIABILITY SCORING

We explore innate limitations of reliability scoring. These impossibility results form a foundation for charting the feasible combinations of $\mathcal{P}$ and $\mathcal{Q}$ for reliability scoring and motivate Section 4.

This section focuses on the partial knowledge setting, where the joint distribution of observations and reported data, $PQ$, is assumed to be known, and provided as input to the score. Impossibility results in this setting extend to the detail-free setting for reliability scores that rely on estimates of $PQ$. In particular, the impossibility results apply to the Gram determinant score that we'll introduce in Section 4. We provide a more detailed discussion in Appendix C.

We first introduce the class of independent experiments and a few classes of misreport matrices that'll be used in this paper.

- $\mathcal{P}_{\text{indep}}$: the set of linearly independent experiments, where $P \in \mathcal{P}_{\text{indep}}$ if and only if all columns of $P$ are linearly independent.
- $\mathcal{Q}_{\text{nonperm}}$: the set of misreport matrices $Q$ so that the associated $Q_{\hat{x}|x}$ is neither a permutation matrix nor an identity matrix.
- $\mathcal{Q}_{\text{reg}}$: the set of invertible and diagonally maximal misreport matrices where $Q(i, j) \leq Q(i, i)$ for all $i$ and $j$. This was also defined earlier in Section 2.3.
- $\mathcal{Q}_{\text{dom}}$: the set of (row) diagonally dominant misreport matrices where $\sum_{j:j \neq i} |Q(i, j)| \leq |Q(i, i)|$ for all $i$.[4]
- $\mathcal{Q}_{L,\delta}$: the set of (row) diagonally dominant misreport matrices where the true data are $L$ balanced and the Hamming distance is bounded above by $N\delta$. True data $x$ is $L$-balanced if $q_x(x) \leq L q_x(x')$ for all $x, x' \in \mathcal{X}$. We use $\mathcal{Q}_L := \mathcal{Q}_{L,1}$ to denote the set of (row) diagonally dominant misreport matrices where the true data are $L$ balanced, with no restriction on Hamming distance.

We note that $\mathcal{Q}_{L,\delta} \subseteq \mathcal{Q}_L \subset \mathcal{Q}_{\text{dom}} \subset \mathcal{Q}_{\text{reg}} \subset \mathcal{Q}_{\text{nonperm}}$ for all $L$ and $\delta$.

**Proposition 3.1.** *In the partial-knowledge setting, it is sometimes impossible for any reliability score to preserve reliability orderings. In particular,*

1. ***Exact match ordering:*** *There exists a $\mathcal{P}$ so that no score preserves the exact match ordering under $\mathcal{P}$ and $\mathcal{Q}_{nonperm}$. Additionally, for all $\mathcal{Q} \supsetneq \mathcal{Q}_{nonperm}$, no score preserves the exact match ordering on $\mathcal{P}_{indep}$ and $\mathcal{Q}$.*

2. ***Blackwell dominant ordering:*** *For any $\mathcal{P}$, if there exists $P \in \mathcal{P}$ and a rational vector $v \neq 0$ so that $Pv = 0$, no score preserves the Blackwell ordering on $\mathcal{P}$ and $\mathcal{Q}_{reg}$.*

3. ***Hamming and dist orderings:*** *No score preserves the Hamming ordering under $\mathcal{P}_{indep}$ and $\mathcal{Q}_{dom}$. Additionally, no score preserves the dist ordering under $\mathcal{P}_{indep}$ and $\mathcal{Q}_{dom}$ for any dist.*

The first part of Proposition 3.1 establishes that no score can respect the exact-match reliability ordering across all experiment sets. The non-permutation condition is needed here to exclude degenerate cases such as label permutations. By Proposition 2.1, these impossibility results also extend

---

[4]Note that diagonally dominant matrices are invertible by Gershgorin circle theorem.

to the other orderings. The second part further shows that even a single linearly dependent experiment is enough to make preservation of the Blackwell ordering impossible. We therefore focus on the class of linearly independent experiments, $\mathcal{P}_{\text{indep}}$. Finally, the third part shows that no reliability score can preserve the Hamming or any other dist ordering, even under diagonally dominant misreport matrices $\mathcal{Q}_{\text{dom}}$. In Section 4, we thus further restrict our attention to $\mathcal{Q}_{L,\delta}$.

# 4 GRAM DETERMINANT RELIABILITY SCORE

Our idea for measuring data reliability is to leverage the diversity of observations. We formalize this idea with the Gram determinant score—the determinant of a Gram matrix of the observation distributions conditional on reported labels.

**Definition 4.1.** *Given finite sets $\mathcal{X} = [d]$ and $\mathcal{Y}$, and an experiment $\boldsymbol{P}$, we define Gram matrix of labels as $\boldsymbol{G} = \boldsymbol{P}^\intercal \boldsymbol{P} \in \mathbb{R}^{|\mathcal{X}| \times |\mathcal{X}|}$ where $\boldsymbol{G}(x, x') = \langle P_x, P_{x'} \rangle = \Pr_{y \sim P_x, y' \sim P_{x'}}[y = y']$. Moreover, given $\boldsymbol{x}$ and $\hat{\boldsymbol{x}}$, we define the Gram matrix of reports $\hat{\boldsymbol{x}}$ as $\hat{\boldsymbol{G}} = (\boldsymbol{P}\boldsymbol{Q})^\intercal (\boldsymbol{P}\boldsymbol{Q}) \in \mathbb{R}^{|\mathcal{X}| \times |\mathcal{X}|}$ where $\hat{\boldsymbol{G}}(x, x') := \frac{1}{N^2} \sum_{n,n':\hat{x}_n=x,\hat{x}_{n'}=x'} \langle P_{x_n}, P_{x_{n'}} \rangle$. The **Gram determinant score** is*

$$\Gamma := \det\left(\hat{\boldsymbol{G}}\right) = \sum_{\sigma \in symm(d)} sgn(\sigma) \prod_{i=1}^{d} \hat{\boldsymbol{G}}(i, \sigma(i)). \tag{3}$$

*where $symm(d)$ is the set of all permutations on $[d]$ and $sgn$ the sign function of permutations. We further denote $\Gamma(\boldsymbol{P}\boldsymbol{Q}) := \Gamma$ to highlight that the Gram determinant score takes $\boldsymbol{P}\boldsymbol{Q}$ as input.*

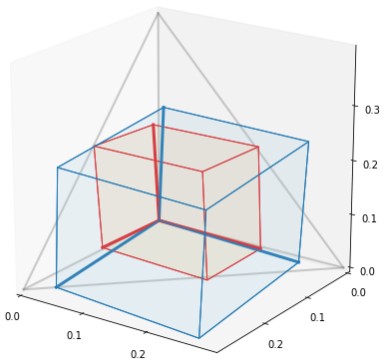

Figure 1: Gram determinant scores and parallelepipeds. [5]

Before proving properties of the Gram determinant score, we first present some intuitions. $\boldsymbol{G}(x, x')$ corresponds to the probability that true data $x$ and $x'$ have the same observation. The Gram matrix of reports

$$\hat{\boldsymbol{G}} = \boldsymbol{Q}^\intercal \boldsymbol{P}^\intercal \boldsymbol{P} \boldsymbol{Q} = \boldsymbol{Q}^\intercal \boldsymbol{G} \boldsymbol{Q}. \tag{4}$$

Moreover, $\det(\hat{\boldsymbol{G}}) = \det((\boldsymbol{P}\boldsymbol{Q})^\intercal \boldsymbol{P}\boldsymbol{Q})$ is the Gram determinant of $\boldsymbol{P}\boldsymbol{Q} \in \mathbb{R}^{|\mathcal{Y}| \times |\mathcal{X}|}$ which is the square of the volume of the parallelepiped spanned by the column vectors of $\boldsymbol{P}\boldsymbol{Q}$ (Horn & Johnson, 2012), as illustrated in Fig. 1. The Gram determinant score of true data, $\Gamma(\boldsymbol{P}\boldsymbol{Q}_{\boldsymbol{x}})$, is the squared volume of the blue parallelepiped spanned by column vectors in $\boldsymbol{P}\boldsymbol{Q}_{\boldsymbol{x}}$, $vol(\boldsymbol{P}\boldsymbol{Q}_{\boldsymbol{x}})^2$. As $\Gamma(\boldsymbol{P}\boldsymbol{Q}) = \Gamma(\boldsymbol{P}\boldsymbol{Q}_{\boldsymbol{x}}\boldsymbol{Q}_{\hat{\boldsymbol{x}}|\boldsymbol{x}}^\intercal) = \Gamma(\boldsymbol{P}\boldsymbol{Q}_{\boldsymbol{x}}\boldsymbol{Q}_{\hat{\boldsymbol{x}}|\boldsymbol{x}})$, the Gram determinant score of reported data is the squared volume of the red parallelepiped, $vol(\boldsymbol{P}\boldsymbol{Q}_{\boldsymbol{x}}\boldsymbol{Q}_{\hat{\boldsymbol{x}}|\boldsymbol{x}})^2$, which is smaller than that of the true data because $\boldsymbol{Q}_{\hat{\boldsymbol{x}}|\boldsymbol{x}}$ is column stochastic and each column of $\boldsymbol{P}\boldsymbol{Q}_{\boldsymbol{x}}\boldsymbol{Q}_{\hat{\boldsymbol{x}}|\boldsymbol{x}}$ is a convex combination of columns of $\boldsymbol{P}\boldsymbol{Q}_{\boldsymbol{x}}$. A symbolic example is presented in Appendix D.1.

In the remainder of this section, we first show that the Gram determinant score preserves several reliability orderings and is invariant under experiments (Section 4.1). We then introduce two estimators of the Gram determinant score for the detail-free setting (Section 4.2). Finally, we introduce kernels to generalize Gram determinant score to handle non-finite observation spaces $\mathcal{Y}$ (Section 4.3).

## 4.1 PRESERVING RELIABILITY ORDERINGS AND INVARIANCE

We show that Gram determinant reliability score preserves the exact, the Blackwell dominant, and the approximated Hamming (or dist) ordering.

---

[5]Figure 1 uses $\boldsymbol{P} = \begin{pmatrix} 0.1 & 0.1 & 0.7 \\ 0.9 & 0.1 & 0.2 \\ 0 & 0.8 & 0.1 \end{pmatrix}$, $\boldsymbol{Q}_{\boldsymbol{x}} = \begin{pmatrix} 0.3 & 0 & 0 \\ 0 & 0.3 & 0 \\ 0 & 0 & 0.4 \end{pmatrix}$, and $\boldsymbol{Q}_{\hat{\boldsymbol{x}}|\boldsymbol{x}} = \begin{pmatrix} 0.1 & 0.1 & 0.7 \\ 0.9 & 0.1 & 0.2 \\ 0 & 0.8 & 0.1 \end{pmatrix}$.

**Theorem 4.2.** *Given $\mathcal{X} = [d]$, a finite set $\mathcal{Y}$, and $L \geq 1$, the Gram determinant score in Definition 4.1 preserves*

1. *exact match ordering under $\mathcal{P}_{indep}$ and $\mathcal{Q}_{nonperm}$,*

2. *Blackwell ordering under $\mathcal{P}_{indep}$ and $\mathcal{Q}_{reg}$, and*

3. *$\frac{1}{4L\Delta}$-dist ordering under $\mathcal{P}_{indep}$ and $\mathcal{Q}_{L,1/64L^2d^2}$ for all dist with $\Delta = \frac{\max_{x,x'\in\mathcal{X}}\mathrm{dist}(x,x')}{\min_{x\neq x'\in\mathcal{X}}\mathrm{dist}(x,x')}$.*

Theorem 4.2 covers any linearly independent experiment—required by the impossibilities in Section 3—and places minimal assumptions on misreports, nearly matching our impossibility results. In particular, Propositions 2.1 and 3.1 show: 1) no score preserves exact ordering for any superset of $\mathcal{Q}_{nonperm}$; 2) the Blackwell relation is only a strict partial order on $\mathcal{Q}_{reg}$; and 3) no score preserves Hamming ordering or any other dist ordering on $\mathcal{Q}_{dom}$. The third part of Theorem 4.2 implies the score preserves $\frac{1}{4L}$-Hamming ordering, because the aspect ratio for Hamming distance is $\Delta = 1$.

The key idea of the proof is that the determinant has the multiplicative property and Eq. (4),

$$\Gamma(\boldsymbol{PQ}) = \det(\boldsymbol{Q}^\intercal \boldsymbol{P}^\intercal \boldsymbol{PQ}) = \det(\boldsymbol{Q}^\intercal)\det(\boldsymbol{P}^\intercal \boldsymbol{P})\det(\boldsymbol{Q}) = \det(\boldsymbol{P}^\intercal \boldsymbol{P})\det(\boldsymbol{Q})^2$$

because $\boldsymbol{Q}$ and $\boldsymbol{P}^\intercal\boldsymbol{P}$ are squared matrices. Hence, we can decouple the misreport matrix $\boldsymbol{Q}$ from the quality of the experiment $\boldsymbol{P}$. In particular, it is sufficient to focus on misreport matrices as the Gram matrix of labels is positive definite $\boldsymbol{P}^\intercal\boldsymbol{P}$, $\det(\boldsymbol{P}^\intercal\boldsymbol{P}) > 0$, for all $\boldsymbol{P} \in \mathcal{P}_{indep}$. This observation may provide a recipe for considering other reliability orderings in the Gram determinant score. The formal proof is deferred to Appendix D.2.

We now establish an invariance principle: the induced ranking of datasets should be invariant to the unknown experiment, to relabelings, and to priors. The latter two are straightforward by the multiplicative property of Gram determinant. For the first one, we show that the Gram determinant is *experiment-agnostic* so that the reliability ranking of a dataset $\hat{\boldsymbol{x}}$ should depend only on $\hat{\boldsymbol{x}}$ and the true data $\boldsymbol{x}$ (defined in Eq. (5)). Thus the choice of experiment does not affect which reported dataset is deemed more reliable. Moreover, we show that the Gram determinant score is the unique experiment agnostic score up to scaling under mild coherence assumption.

**Proposition 4.3.** *Given $\mathcal{X} = [d]$ and a finite set $\mathcal{Y}$, the Gram determinant score in Definition 4.1 is experiment agnostic so that for all $\boldsymbol{Q}, \boldsymbol{Q}' \in GL_d$ general linear group and $\boldsymbol{P} \in \mathcal{P}_{indep}$,*

$$\Gamma(\boldsymbol{Q}) \geq \Gamma(\boldsymbol{Q}') \Leftrightarrow \Gamma(\boldsymbol{PQ}) \geq \Gamma(\boldsymbol{PQ}'). \tag{5}$$

*Moreover, if there exists a continuous function $S : GL_d \to \mathbb{R}_{>0}$ with a continuous $c : \mathbb{R}_{>0} \to \mathbb{R}_{>0}$ so that for all $\boldsymbol{Q}, \boldsymbol{Q}', \boldsymbol{P} \in GL_d$, and $t > 0$, Eq. (5) holds and $S(t\boldsymbol{Q}) = c(t)S(\boldsymbol{Q})$, there exists $\alpha > 0, \beta \neq 0$ so that $S(\boldsymbol{Q}) = \alpha \det(\boldsymbol{Q}^\intercal\boldsymbol{Q})^\beta$.*

As discussed above, the first part follows directly from multiplicative property of determinant, $\Gamma(\boldsymbol{PQ}) = \det(\boldsymbol{P}^\intercal\boldsymbol{P})\det(\boldsymbol{Q})^2 = \det(\boldsymbol{P}^\intercal\boldsymbol{P})\Gamma(\boldsymbol{Q})$. We deter the proof for the second part to Appendix D.3. Finally, since $GL_d \subset \mathcal{P}_{indep}$, the second part of Proposition 4.3 implies that even when we restrict to settings where the observation space has the same dimension as the data space $|\mathcal{Y}| = |\mathcal{X}|$, the Gram determinant score remains unique up to scaling.

## 4.2 Estimators for Gram Determinant Scores

We introduce two estimators for the Gram determinant score in the detail-free setting: plug-in and stratified matching estimator. The second estimator and proofs are deferred to Appendix E.

**Definition 4.4** (plug-in Gram determinant reliability score). *Given $\hat{\boldsymbol{x}}$ and $\boldsymbol{y}$ of size $N$, define $\bar{\boldsymbol{G}} \in \mathbb{R}^{d\times d}$ so that for all $x, x' \in \mathcal{X}$ $\bar{\boldsymbol{G}}(x,x') = \frac{1}{N^2}\sum_{n,n'\in[N]:\hat{x}_n=x,\hat{x}_{n'}=x'} \mathbf{1}[y_n = y_{n'}]$. The plug-in Gram determinant reliability score is then defined as $\bar{S}(\hat{\boldsymbol{x}}, \boldsymbol{y}) = \det(\bar{\boldsymbol{G}})$.*

The plug-in estimator first estimates $\hat{\boldsymbol{G}}$ using empirical distribution between reports $\hat{\boldsymbol{x}}$ and observations $\boldsymbol{y}$ and computes the determinants of $\hat{\boldsymbol{G}}$. Note that the probability of $y_n = y_{n'}$ is simply the inner product of $P_{x_n}$ and $P_{x_{n'}}$ if $n \neq n'$. Proposition 4.5 shows that the plug-in estimator asymptotically preserves all reliability orderings in Theorem 4.2.

**Proposition 4.5.** *Given $\mathcal{X} = [d]$, finite set $\mathcal{Y}$ and $L \geq 1$, the plug-in Gram determinant score in Definition 4.4 asymptotically preserves reliability orderings in Theorem 4.2.*

### 4.3 GRAM DETERMINANT SCORE WITH KERNELS

The Gram determinant score in Definition 4.1 has two limitations. First, it cannot handle continuous or general observation space $\mathcal{Y}$. Second, it ignores any intrinsic structure in the observation space, e.g., prediction or feature embedding. We extend the Gram determinant score with kernels. We provide examples of different kernels that can be used in practice, together with a reliability-ordering result analogous to Theorem 4.2 in Appendix F.

**Definition 4.6** (kernelized Gram determinant score). *Given a finite set $\mathcal{X}$, an experiment $\boldsymbol{P}$, and $\mathcal{Y}$ with a kernel $K : \mathcal{Y} \times \mathcal{Y} \to \mathbb{R}$, we define Gram matrix of labels as $\boldsymbol{G}_K \in \mathbb{R}^{d \times d}$ where for all $x, x' \in \mathcal{X}$, $\boldsymbol{G}_K(x, x') = \langle P_x, P_{x'} \rangle_K := \mathbb{E}_{y \sim P_x, y' \sim P_{x'}}[K(y, y')]$. Given $\boldsymbol{x}$ and $\hat{\boldsymbol{x}}$, we define the Gram matrix of reports as $\hat{\boldsymbol{G}}_K \in \mathbb{R}^{d \times d}$ where $\hat{\boldsymbol{G}}_K(x, x') = \frac{1}{N^2} \sum_{n,n' : \hat{x}_n = x, \hat{x}_{n'} = x'} \langle P_{x_n}, P_{x_{n'}} \rangle_K$.*

*Finally, we define the* Gram determinant score *with kernel $K$ as $\Gamma_K := \det\left(\hat{\boldsymbol{G}}_K\right)$.*

## 5 EXPERIMENTS

We evaluate the Gram determinant score in three parts: (Exp. 1) synthetic categorical data with six label-manipulation policies; (Exp. 2) real image data (CIFAR-10 embeddings) with the same six manipulations using the kernelized score; (Exp. 3) real employment data, treating CES vintage revisions as naturally occurring manipulations.

**Experiment 1: Gram Determinant Score on Synthetic Data**   In this experiment, we evaluate how well the Gram determinant score captures label reliability under categorical observations, as summarized in Figs. 2 and 2d. Specifically, we first generate a ground-truth dataset $(\boldsymbol{x}, \boldsymbol{y})$ of size $N = 4000$ with $d = 5$. Each label $x_k$ is drawn uniformly from $1, \ldots, d$ for $k \in [N]$, and each outcome $y_k$ is sampled from the distribution $\boldsymbol{P}(\cdot \mid x_k)$, where the experiment distribution matrix $\boldsymbol{P} \in [0, 1]^{d \times d}$ is constructed by sampling $\boldsymbol{P}(i, j) \sim \mathrm{Uniform}(0, 1)$ independently and normalizing rows to be stochastic. The ground-truth dataset $(\boldsymbol{x}, \boldsymbol{y})$ is fixed across all trials. To model varying reliability, for each $p \in \{0.00, 0.05, \ldots, 0.50\}$ we corrupt the labels according to

$$\hat{x}_k = \begin{cases} x_k, & \text{with probability } 1 - p, \\ Z_k, & \text{with probability } p, \end{cases} \tag{6}$$

where $Z_k \sim \pi(\cdot \mid x_k)$ is independently drawn from a corruption policy $\pi$; in our experiments, $\pi$ is instantiated by one of the six manipulations below.

- Uniformly random: $Z_k \sim \mathrm{Uniform}\{1, \ldots, d\}$.
- Asym neighbor: with probability $0.85$ set $Z_k = \min\{x_k + 1, d\}$, otherwise sample $Z_k$ uniformly from $\{1, \ldots, d\} \setminus \{x_k\}$.
- Row-sim 2nd: $Z_k = \arg\max_{j \neq x_k} \frac{\langle \boldsymbol{P}_{x_k,\cdot}, \boldsymbol{P}_{j,\cdot} \rangle}{\|\boldsymbol{P}_{x_k,\cdot}\| \, \|\boldsymbol{P}_{j,\cdot}\|}$, the label with closest observation distribution.
- Merge $0/1 \to 0$: if $x_k \in \{1, 2\}$ then set $Z_k = 1$; otherwise $Z_k = x_k$.
- Group up/down: $Z_k = \min\{x_k + 1, d\}$ with probability $1/2$, or $Z_k = \max\{x_k - 1, 1\}$ otherwise.
- Mixed: sample $Z_k \sim \pi_{\mathrm{mixed}}(\cdot | x_k)$ where each row $\pi_{\mathrm{mixed}}(i, \cdot)$ is drawn from $\mathrm{Dirichlet}\big(\alpha_i(1), \ldots, \alpha_i(d)\big)$ with

  $\alpha_i(j) = \alpha_{\mathrm{off}} + \alpha_{\mathrm{diag}}\mathbf{1}\{j = i\} + \lambda_{\mathrm{loc}} \exp\big(-\mathrm{dist}_{\mathrm{ring}}(i, j)\big) + \lambda_{\mathrm{up}} \exp\big(\gamma(j - i)\big) + \lambda_{\mathrm{def}}\mathbf{1}\{j = j_0\}$,

  $\mathrm{dist}_{\mathrm{ring}}(i, j) = \min(|i - j|, d - |i - j|)$, and $j_0$ a salient default label; rows are normalized to be stochastic, where $\alpha_{\mathrm{off}} = 0.2, \alpha_{\mathrm{diag}} = 6, \lambda_{\mathrm{loc}} = 1.0, \lambda_{\mathrm{up}} = 0.4, \gamma = 0.5, \lambda_{\mathrm{def}} = 0.6, j_0 = 1$. This policy mimics human labeling: diagonal dominance (keep $i$), locality on the ring (near-class confusions), mild upcoding (asymmetric mistakes), and a default-label bias—yielding structured, non-uniform noise beyond uniform corruption.

Fix a ground-truth dataset $(\boldsymbol{x}, \boldsymbol{y})$.   For each manipulation and corruption level $p \in \{0.0, 0.1, \ldots, 0.5\}$, in Figs. 2a to 2c, we run $M = 100$ independent trials, producing corrupted

reports $\hat{\boldsymbol{x}}^m$. In every trial, we compute 1) the plug-in Gram determinant reliability score in Definition 4.4, 2) the Hamming error $\sum_{n=1}^{N} \mathbf{1}[x_n \neq \hat{x}_n^m]$, and 3) the $\ell_2$ error $\|\boldsymbol{x} - \hat{\boldsymbol{x}}^m\|_2$. We then report the mean and standard deviation of each metric across the $M$ trials. In Fig. 2a, the plug-in Gram-determinant score falls steadily as the corruption probability $p$ increases. Figures 2b and 2c show that higher scores correspond to lower Hamming error and smaller $\ell_2$ deviation, respectively, demonstrating a clear negative correlation between our score and these conventional error measures regardless of the manipulation policy (i.e., across all corruption schemes considered).

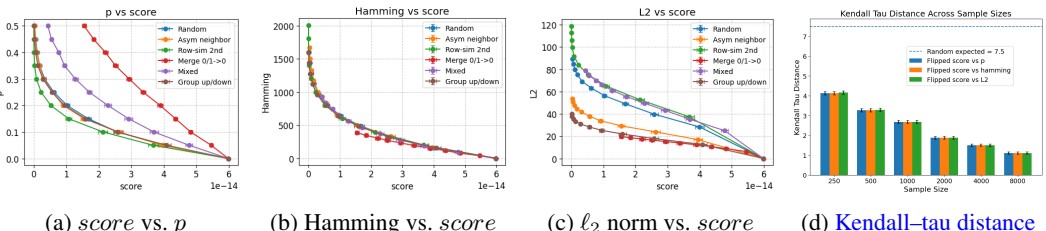

(a) *score* vs. $p$    (b) Hamming vs. *score*    (c) $\ell_2$ norm vs. *score*    (d) Kendall–tau distance

Figure 2: Gram determinant reliability score on categorical synthetic data.

In Fig. 2d, we vary data sizes $N \in \{250, 500, \ldots, 8000\}$ and generate 1000 datasets for each $N$. In each dataset and corruption level $p \in \{0.0, 0.1, \ldots, 0.5\}$, we use the uniform random manipulation strategy, and compute the plug-in Gram determinant, Hamming-distance error, and $\ell_2$ error, then rank the six corrupted reports. We report the average Kendall–tau distance between the reversed Gram–determinant ranking and the orderings induced by $p$, the Hamming distance, and the $\ell_2$ error. As shown in Figure 2d, the fraction of correctly recovered rankings increases with the sample size, confirming that the Gram–determinant score is a consistent indicator of true-label reliability.

**Experiment 2: Gram Determinant Score with Kernels on Image Data**   We evaluate the Gram determinant score with continuous observations by using image embeddings. We train a SimCLR model (Chen et al., 2020) with a ResNet-18 backbone and an 8-dimensional projection head on CIFAR-10 (Krizhevsky et al., 2009). The model is optimized for 60 epochs using the InfoNCE loss with batch size $B = 256$, temperature $\tau = 0.5$, and the Adam optimizer at learning rate $5 \times 10^{-3}$. After training, we extract normalized projections $y_n \in \mathbb{R}^8$ for each of the $N = 10000$ test images, denote the true labels by $\boldsymbol{x} \in \{0, \ldots, 9\}^N$, and the embeddings by $\boldsymbol{y} \in \mathbb{R}^{N \times 8}$.

To simulate corrupted reports, we use the same six corruption policies with $p \in \{0.00, 0.04, \ldots, 0.40\}$. As $\mathcal{Y} = \mathbb{R}^8$ is continuous, we use plug-in Gram determinant with kernel $K(y, y') = \langle y, y' \rangle$ as the score. For each $p$ and policy we repeat the procedure over $M = 100$ random trials to obtain the mean and standard error. As shown in Figs. 3a to 3c, the score increases monotonically with $p$ across all six manipulations, and higher score is associated with lower Hamming error and smaller $\ell_2$ deviation, mirroring the trends observed in categorical setting.

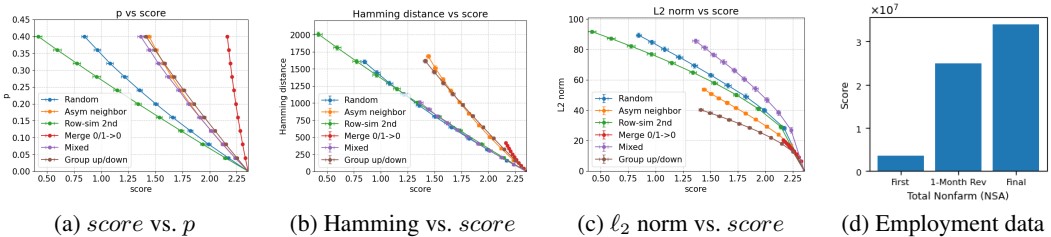

(a) *score* vs. $p$    (b) Hamming vs. *score*    (c) $\ell_2$ norm vs. *score*    (d) Employment data

Figure 3: Gram determinant reliability for image–label experiments under six manipulation policies

**Experiment 3: Gram Determinant Score on Real-World Employment Data**   We evaluate three vintages of the CES total nonfarm employment series (not seasonally adjusted) from Oct 2005–Feb 2023, using the CES vintage dataset (U.S. Bureau of Labor Statistics, 2025), and take as external $\boldsymbol{y}$ the monthly changes in Withheld Income & Employment Taxes from Treasury deposits (U.S. Department of the Treasury, Bureau of the Fiscal Service, 2025). For each month we use: 1) first

release: initial estimate, published the next month; 2) one-month revision: first revision, one month later; and 3) final value: last available vintage including benchmark revisions. We discretize month-to-month differences into four quantile buckets as $x$ and $y$ with $N = 209$ and compute Gram determinant scores with the plug-in estimator. Figure 3d shows that revisions substantially improve reliability according to our score, with the final series most aligned with fiscal benchmarks.

## 6 CONCLUSION

We introduce the Gram determinant score — a metric that intuitively measures the volume of class-conditional observation distributions. Under mild independence assumptions, it exactly preserves exact-match and Blackwell orderings and closely approximates Hamming orderings. We develop plug-in and stratified-matching estimators with finite-sample guarantees and extend the method to continuous or structured spaces via kernel embeddings. Experiments on synthetic data, CIFAR-10 embeddings, and employment data demonstrated its effectiveness.

Looking ahead, it's interesting to design scalable estimators for high-dimensional or continuous label domains using dimensionality-reduction (e.g., PCA, DPP sampling) and learned encoders. Moreover, we conjecture that other singular-value–based criteria can also serve as reliability scores. Appendix G briefly discusses additional candidates beyond the Gram determinant score and reports synthetic-data experiments evaluating them. However, formal guarantees remain to be established; each candidate will require tailored analysis to show it preserves the relevant reliability orderings. In real-world settings, the Gram determinant score is applicable wherever labels are noisy or manipulated – for example, by detecting incoherent star ratings in product reviews – and could help platforms like Amazon and Yelp enhance consumer protection.

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
