APPENDIX

# A  PRELIMINARY: MATRICES AND KERNELS

This section provides basic definitions and theorems for matrices and kernels. Given a $d \times d$ matrix $\boldsymbol{A}$, the determinant of $\boldsymbol{A}$ is

$$\det(\boldsymbol{A}) = \sum_{\sigma \in symm(d)} sgn(\sigma) \prod_{i=1}^{d} A(i, \sigma(i)),$$

where $symm(m)$ is the set of all permutations of $[d]$ and $sgn(\sigma)$ is the sign function of a permutation.

Given two $d \times d$ matrices $\boldsymbol{A}$ and $\boldsymbol{B}$, the Frobenius inner product between them is $\langle \boldsymbol{A}, \boldsymbol{B} \rangle_F := \sum_{i,j \in [d]} \boldsymbol{A}(i,j)\boldsymbol{B}(i,j)$.

We introduce two approximation results for determinants. The first one shows that $\det(\boldsymbol{A})$ can be approximated by the determinant of its diagonal matrix, and the second shows that the determinant is smooth under small perturbation.

**Theorem A.1** ((Ipsen & Lee, 2011)). *Let $\boldsymbol{A}$ be a $d$-dimensional squared matrix, $\boldsymbol{A}_D$ be the associated diagonal matrix, and $\boldsymbol{A}_E = \boldsymbol{A} - \boldsymbol{A}_D$. If $\boldsymbol{A}_D$ is non-singular and spectral norm $\rho := \|\boldsymbol{A}_D^{-1} \boldsymbol{A}_E\|_2 < 1$ then*

$$\frac{|\det(\boldsymbol{A}) - \det(\boldsymbol{A}_D)|}{|\det(\boldsymbol{A}_D)|} \leq c\rho e^{c\rho}, \text{ where } c = -d \ln(1 - \rho)$$

*Moreover, if $c\rho < 1$, $\frac{|\det(\boldsymbol{A}) - \det(\boldsymbol{A}_D)|}{|\det(\boldsymbol{A}_D)|} \leq \frac{7}{4}c\rho$.*

**Theorem A.2** ((Ipsen & Rehman, 2008)). *Let $\boldsymbol{A}$ and $\boldsymbol{E}$ be $d \times d$ matrices. If $A$ is nonsingular, then*

$$\frac{|\det(\boldsymbol{A} + \boldsymbol{E}) - \det(\boldsymbol{A})|}{|\det(\boldsymbol{A})|} \leq \left(1 + \kappa \frac{\|\boldsymbol{E}\|_2}{\|\boldsymbol{A}\|_2}\right)^d - 1$$

*where $\kappa = \|\boldsymbol{A}\|_2 \|\boldsymbol{A}^{-1}\|_2$ and $\|\cdot\|_2$ is the spectral norm.*

**Lemma A.3.** *Given $\boldsymbol{A}, \boldsymbol{B} \in \mathbb{R}^{d \times d}$, if $\boldsymbol{B} \neq \mathbb{I}$ is column stochastic and $\boldsymbol{A}, \boldsymbol{BA}$ are column diagonally maximal, $\boldsymbol{B}$ is not a permutation matrix.*

*Proof of Lemma A.3.* Suppose not and there exists a permutation $\sigma : [d] \to [d]$ and $\iota \in [d]$ so that $B(i, j) = \mathbf{1}[j = \sigma(i)]$ and $\sigma(\iota) \neq \iota$. Because $\boldsymbol{A}$ is column diagonally maximal

$$(\boldsymbol{BA})(\iota, \iota) = \sum_j \boldsymbol{B}(\iota, j)\boldsymbol{A}(j, \iota) = \boldsymbol{A}(\sigma(\iota), \iota) < \boldsymbol{A}(\iota, \iota).$$

Additionally,

$$(\boldsymbol{BA})(\sigma^{-1}(\iota), \iota) = \boldsymbol{B}(\sigma^{-1}(\iota), \iota)\boldsymbol{A}(\iota, \iota) = \boldsymbol{A}(\iota, \iota) > (\boldsymbol{BA})(\iota, \iota).$$

Therefore, $\boldsymbol{BA}$ is not column diagonally maximal which is a contradiction. □

Now we introduce kernel.

**Definition A.4.** *A function $K : \mathcal{Y} \times \mathcal{Y} \to \mathbb{R}$ is* positive definite kernel *if for all $\{y_1, \ldots, y_m\} \subseteq \mathcal{Y}$, the matrix $[K(y_i, y_j)]_{ij} \in \mathbb{R}^{m \times m}$ is symmetric positive semi definite. Additionally, it is* strictly positive definite *if the matrix is positive definite.*

By Moore-Aronszajn theorem (Aronszajn, 1950), given a positive definite kernel $K$, there exists a Hilbert space $\mathcal{H}$ known as a reproducing kernel Hilbert space so that for any $y \in \mathcal{Y}$, $K(\cdot, y) \in \mathcal{H}$ and for all $h \in \mathcal{H}$, $h(y) = \langle h, K(y, \cdot) \rangle$. This allows us to think of a kernel defines a feature map $\phi : y \mapsto K(\cdot, x) \in \mathcal{H}$ where the inner product in the embedded space reduces to kernel evaluation, because $\langle K(\cdot, y), K(\cdot, y') \rangle = K(y, y')$

Moreover, given a measurable kernel, we can define the *kernel mean embedding* (Berlinet & Thomas-Agnan, 2011) of probability measures on $\mathcal{Y}$, $P \in \Delta(\mathcal{Y})$, into $\mathcal{H}$ where

$$\phi(P) := \int K(\cdot, y) dP(y) = \mathbb{E}_{y \sim P}[\phi(y)].$$

Here we slightly abuse the notations, and note that $\phi$ is linear in $P$ by linearity of integration. We can further extend this to signed measures $\phi(\mu) := \int K(\cdot, y) d\mu(y)$. Finally, a kernel $K$ is *integrally strictly positive definite* if the $\int\int_{\mathcal{Y}} K(y, y') d\mu(y) d\mu(y') > 0$ for all finite non-zero signed measures $\mu$.

# B   PROOFS AND DETAILS IN SECTION 2

We show that the reliability orderings are well-defined ordering. Formally, a binary relationship $\succ$ on $\Omega$ is a *strict partial order* if it satisfies the following conditions for all $a, b, c \in \Omega$

1. anti-reflexive: no element is larger than itself
2. asymmetry: if $a \succ b$ then not $b \succ a$
3. Transitivity: if $a \succ b$ and $b \succ c$, then $a \succ c$.

Next, we show that the reliability orderings defined in Section 2 form a strict partial order over reports, given a fixed true data.

**Proposition B.1.** *For any $\boldsymbol{x} \in \mathcal{X}^N$, the exact match ordering $\succ_{\text{EXACT}}^{\boldsymbol{x}}$ is a strict partial order on all $\hat{\boldsymbol{x}}$ and $\hat{\boldsymbol{x}}' \in \mathcal{X}^N$.*

*Proof.* The first two are trivial. For transitivity, if $\hat{\boldsymbol{x}}_1 \succ_{\text{EXACT}}^{\boldsymbol{x}} \hat{\boldsymbol{x}}_2$, then $\hat{\boldsymbol{x}}_2 \neq \boldsymbol{x}$ so there is no $\hat{\boldsymbol{x}}_3$ with $\hat{\boldsymbol{x}}_2 \succ_{\text{EXACT}}^{\boldsymbol{x}} \hat{\boldsymbol{x}}_3$. $\qquad\square$

The following shows that Blackwell dominant ordering is a strict partial order over subsets of reports under the invertible and diagonally maximal conditions. Those conditions are essential. If the misreport matrices are not invertible, the Blackwell ordering may fail to be asymmetric: it is possible for two distinct reports to Blackwell-dominate each other, violating the strictness of the relation. Similarly, if the misreport matrices are not diagonally maximal, the ordering also fails asymmetry via non-trivial permutation.

**Proposition B.2.** *For any $\boldsymbol{x} \in \mathcal{X}^N$, Blackwell dominant ordering $\succ_{\text{Blackwell}}^{\boldsymbol{x}}$ is a strict partial order on all $\hat{\boldsymbol{x}}$ and $\hat{\boldsymbol{x}}' \in \mathcal{X}^N$ so that the associated misreport matrices $\boldsymbol{Q}, \boldsymbol{Q}' \in \mathcal{Q}_{reg}$ are invertible and diagonally maximal.*

*Proof.* Suppose $\succ_{\text{Blackwell}}^{\boldsymbol{x}}$ is not anti-reflective. There exists $\hat{\boldsymbol{x}} \succ_{\text{Blackwell}}^{\boldsymbol{x}} \hat{\boldsymbol{x}}$ with misreport matrix $\boldsymbol{Q}$ and a column stochastic matrix $\boldsymbol{T} \neq \mathbb{I}$ so that

$$\boldsymbol{T}\boldsymbol{Q}_{\hat{\boldsymbol{x}}|\boldsymbol{x}} = \boldsymbol{Q}_{\hat{\boldsymbol{x}}|\boldsymbol{x}}.$$

Because $\boldsymbol{Q} = (\boldsymbol{Q}_{\hat{\boldsymbol{x}}|\boldsymbol{x}} \boldsymbol{Q}_{\boldsymbol{x}})^{\mathsf{T}}$ is invertible, $\boldsymbol{Q}_{\hat{\boldsymbol{x}}|\boldsymbol{x}}$ is also invertible and $\boldsymbol{T} = \mathbb{I}$ which is a contradiction.

For asymmetry, if $\hat{\boldsymbol{x}} \succ_{\text{Blackwell}}^{\boldsymbol{x}} \hat{\boldsymbol{x}}'$ and $\hat{\boldsymbol{x}}' \succ_{\text{Blackwell}}^{\boldsymbol{x}} \hat{\boldsymbol{x}}$, there exist column stochastic matrices $\boldsymbol{T}$ and $\boldsymbol{T}'$ so that

$$\boldsymbol{T}\boldsymbol{Q}_{\hat{\boldsymbol{x}}|\boldsymbol{x}} = \boldsymbol{Q}'_{\hat{\boldsymbol{x}}|\boldsymbol{x}} \text{ and } \boldsymbol{T}'\boldsymbol{Q}'_{\hat{\boldsymbol{x}}|\boldsymbol{x}} = \boldsymbol{Q}_{\hat{\boldsymbol{x}}|\boldsymbol{x}}.$$

Because $\boldsymbol{Q}, \boldsymbol{Q}'$ are invertible, $\boldsymbol{T}\boldsymbol{T}' = \mathbb{I}$, and both $\boldsymbol{T}$ and $\boldsymbol{T}'$ are permutation matrices. (https://math.stackexchange.com/users/436618/angina seng) However, because $\boldsymbol{Q}$ and $\boldsymbol{Q}'$ are (row) diagonally maximal, $\boldsymbol{Q}_{\hat{\boldsymbol{x}}|\boldsymbol{x}}$ and $\boldsymbol{Q}'_{\hat{\boldsymbol{x}}|\boldsymbol{x}}$ are column diagonally maximal. Therefore by Lemma A.3, $\boldsymbol{T} = \boldsymbol{T}' = \mathbb{I}$ which is a contradiction.

Transitivity is trivial, because the product of column stochastic matrices is still stochastic. $\qquad\square$

**Proposition B.3.** *For any $\boldsymbol{x} \in \mathcal{X}^N$, dist ordering $\succ_{\text{dist}}^{\boldsymbol{x}}$ is a strict partial order on all $\hat{\boldsymbol{x}}$ and $\hat{\boldsymbol{x}}' \in \mathcal{X}^N$*

*Proof.* The first two are trivial. For transitivity, given $\boldsymbol{x}, \boldsymbol{x}'$ let $\text{dist}(\boldsymbol{x}, \boldsymbol{x}') := \sum_n \text{dist}(x_n, x'_n)$. If $\hat{\boldsymbol{x}}_1 \succ_{\text{dist}}^{\boldsymbol{x}} \hat{\boldsymbol{x}}_2$ and $\hat{\boldsymbol{x}}_2 \succ_{\text{dist}}^{\boldsymbol{x}} \hat{\boldsymbol{x}}_3$ then $\text{dist}(\boldsymbol{x}, \hat{\boldsymbol{x}}_1) < \text{dist}(\boldsymbol{x}, \hat{\boldsymbol{x}}_2)$ and $\text{dist}(\boldsymbol{x}, \hat{\boldsymbol{x}}_2) < \text{dist}(\boldsymbol{x}, \hat{\boldsymbol{x}}_3)$. Therefore, $\hat{\boldsymbol{x}}_1 \succ_{\text{dist}}^{\boldsymbol{x}} \hat{\boldsymbol{x}}_3$. $\qquad\square$

### B.1 PROOF OF PROPOSITION 2.1

*Proof of Proposition 2.1.* Given $\boldsymbol{x}, \hat{\boldsymbol{x}},$ and $\hat{\boldsymbol{x}}'$, if $\hat{\boldsymbol{x}} \succ^{\boldsymbol{x}}_{\text{EXACT}} \hat{\boldsymbol{x}}'$, $\boldsymbol{Q}_{\hat{\boldsymbol{x}}|\boldsymbol{x}} = \mathbb{I}$ and $\boldsymbol{Q}'_{\hat{\boldsymbol{x}}|\boldsymbol{x}} \neq \mathbb{I}$. If we set a column stochastic $\boldsymbol{T} = \boldsymbol{Q}'_{\hat{\boldsymbol{x}}|\boldsymbol{x}}$, $\boldsymbol{Q}'_{\hat{\boldsymbol{x}}|\boldsymbol{x}} = \boldsymbol{T}\boldsymbol{Q}_{\hat{\boldsymbol{x}}|\boldsymbol{x}}$. Therefore, $\hat{\boldsymbol{x}} \succ^{\boldsymbol{x}}_{\text{Blackwell}} \hat{\boldsymbol{x}}'$.

If $\hat{\boldsymbol{x}} \succ^{\boldsymbol{x}}_{\text{Blackwell}} \hat{\boldsymbol{x}}'$, there is $\boldsymbol{T} \neq \mathbb{I}$ so that $\boldsymbol{Q}'_{\hat{\boldsymbol{x}}|\boldsymbol{x}} = \boldsymbol{T}\boldsymbol{Q}_{\hat{\boldsymbol{x}}|\boldsymbol{x}}$. With Eq. (1) we have $\boldsymbol{Q}' = (\boldsymbol{Q}'_{\hat{\boldsymbol{x}}|\boldsymbol{x}}\boldsymbol{Q}_{\boldsymbol{x}})^{\intercal} = (\boldsymbol{T}\boldsymbol{Q}_{\hat{\boldsymbol{x}}|\boldsymbol{x}}\boldsymbol{Q}_{\boldsymbol{x}})^{\intercal} = \boldsymbol{Q}\boldsymbol{T}^{\intercal}$, and

$$
\begin{aligned}
\text{Tr}(\boldsymbol{Q}') = \text{Tr}(\boldsymbol{Q}\boldsymbol{T}^{\intercal}) &= \sum_{i,j} \boldsymbol{Q}(i,j)\boldsymbol{T}(i,j) \\
&= \sum_{i} \boldsymbol{Q}(i,i)\boldsymbol{T}(i,i) + \sum_{i,j:i\neq j} \boldsymbol{Q}(i,j)\boldsymbol{T}(i,j) \\
&\leq \sum_{i} \boldsymbol{Q}(i,i)\boldsymbol{T}(i,i) + \sum_{i,j:i\neq j} \boldsymbol{Q}(i,i)\boldsymbol{T}(i,j) \quad (\boldsymbol{Q} \text{ is diagonally maximal and } \boldsymbol{T} \neq \mathbb{I}) \\
&= \sum_{i} \boldsymbol{Q}(i,i) = \text{Tr}(\boldsymbol{Q}) \quad (\boldsymbol{T} \text{ is column stochastic})
\end{aligned}
$$

Therefore, $\hat{\boldsymbol{x}} \succ^{\boldsymbol{x}}_{\text{Hamming}} \hat{\boldsymbol{x}}'$. The third on is straightforward by definition of refinement. $\square$

## C PROOFS AND DETAILS IN SECTION 3

We discuss the connection between detail-free setting and partial knowledge setting. First note that as the order of data is not relevant, given $\hat{\boldsymbol{x}}, \boldsymbol{y}$ of size $N$, it is sufficient to consider the histogram of $\boldsymbol{R} \in \mathbb{R}^{|\mathcal{Y}|\times|\mathcal{X}|}$ and $N$ where

$$
\boldsymbol{R}(y,x) = \frac{1}{N}\sum_{n} \mathbf{1}[y_n = y, \hat{x}_n = x].
$$

By symmetrization, we can write a reliability score in detail-free setting as a stochastic function on the histogram $\boldsymbol{R}$ and $N$ that have the same expected score. (Zheng et al., 2021) The expectation of $\boldsymbol{R}$ over the randomness of experiment is $\mathbb{E}[\boldsymbol{R}] = \boldsymbol{P}\boldsymbol{Q}$. This leads to two key implications. First when the data size $N$ is large, $\boldsymbol{R}$ converges to $\boldsymbol{P}\boldsymbol{Q}$ so that the expectation of any smooth reliability score

$$
\mathbb{E}[S(\boldsymbol{R})] \to S(\mathbb{E}[\boldsymbol{R}]) = S(\boldsymbol{P}\boldsymbol{Q}).
$$

Second, if we consider any *empirical risk-based scores* so that has $\ell : \mathcal{X} \times \mathcal{Y} \to \mathbb{R}$ so that

$$
S(\hat{\boldsymbol{x}}, \boldsymbol{y}) = \frac{1}{N}\sum_{n} \ell(\hat{x}_n, y_n).
$$

This includes common metrics like empirical risk and log-likelihood function. We can rewrite it as a linear function of $\boldsymbol{R}$

$$
\begin{aligned}
S(\hat{\boldsymbol{x}}, \boldsymbol{y}) &= \frac{1}{N}\sum_{n} \ell(\hat{x}_n, y_n) \\
&= \frac{1}{N}\sum_{x,y}\sum_{n} \mathbf{1}[\hat{x}_n = x, y_n = y]\ell(x,y) \\
&= \sum_{x,y} \boldsymbol{R}(y,x)\ell(x,y)
\end{aligned}
$$

which is simply the Frobenius inner product between $\boldsymbol{R}$ and the score matrix based on $\ell$.

Finally, as Definition 4.1, our Gram determinant score is also a function of $\boldsymbol{P}\boldsymbol{Q}$. Consequently, the impossibility results presented in Section 3 for the partial knowledge setting apply not only to the Gram determinant score but also to any empirical risk-based score.

We provide the proof of Proposition 3.1 consists of three parts: exact, Blackwell, and Hamming and other dist orderings.

**Proof of Exact orderings in Proposition 3.1**  For the exact ordering setting, we motivate the independence condition on experiments and non-permutation condition on misreport matrix. First we show that we need additional condition on experiments $\mathcal{P}$, even restricting to $\mathcal{Q}_{\text{nonperm}}$. Second, we show that $\mathcal{Q}_{\text{nonperm}}$ is the maximal set of misreport matrices to have a reliability score that respects the exact match ordering on $\mathcal{P}_{\text{indep}}$.

Both parts use the idea that if two labels in $\mathcal{X}$ induce the same distribution over observations, it becomes impossible to determine whether the reports agree with the true data.

For the first part, if $\boldsymbol{P}$ consists of identical columns, we can find a diagonal matrix $\boldsymbol{Q}_{\boldsymbol{x}}$ and a doubly stochastic $\boldsymbol{Q}_{\hat{\boldsymbol{x}}|\boldsymbol{x}} \neq \mathbb{I}$ so that $\boldsymbol{P}(\boldsymbol{Q}_{\hat{\boldsymbol{x}}|\boldsymbol{x}}\boldsymbol{Q}_{\boldsymbol{x}})^{\mathsf{T}} = \boldsymbol{P}\boldsymbol{Q}_{\boldsymbol{x}}\boldsymbol{Q}_{\hat{\boldsymbol{x}}|\boldsymbol{x}}^{\mathsf{T}} = \boldsymbol{P}\boldsymbol{Q}_{\boldsymbol{x}}$. Hence, we can set $\boldsymbol{x}, \hat{\boldsymbol{x}}$ with misreport matrices $\boldsymbol{Q} = (\boldsymbol{Q}_{\hat{\boldsymbol{x}}|\boldsymbol{x}}\boldsymbol{Q}_{\boldsymbol{x}})^{\mathsf{T}}$ so that $\boldsymbol{x} \succ_{\text{EXACT}}^{\boldsymbol{x}} \hat{\boldsymbol{x}}$, but have the same joint distribution between reports and observations. Therefore, no score in the partial knowledge setting can distinguish them and preserves exact match ordering.

For the second part, because we can only observe the observations and reports, it would be impossible to always score true data over relabeled reports (permutation). Suppose not and there exists a score $S$ in partial knowledge setting that preserves all misreport matrices. Given any $\boldsymbol{P} \in \mathcal{P}_{\text{indep}}$, the uniform marginal distribution $\boldsymbol{Q}_{\boldsymbol{x}} := \frac{1}{d}\mathbb{I}$, and permutation $\boldsymbol{T} \neq \mathbb{I}$, there exist $\boldsymbol{x}$ and $\hat{\boldsymbol{x}}$ so that the misreport matrix equals $\boldsymbol{T}\boldsymbol{Q}_{\boldsymbol{x}} = \frac{1}{d}\boldsymbol{T}$ and $\boldsymbol{x} \succ_{\text{EXACT}}^{\boldsymbol{x}} \hat{\boldsymbol{x}}$. Because the joint distribution between reports and observations is $\frac{1}{d}\boldsymbol{P}$ for $(\boldsymbol{x}, \boldsymbol{y})$, and $\frac{1}{d}\boldsymbol{P}\boldsymbol{T}^{\mathsf{T}}$ for $(\hat{\boldsymbol{x}}, \boldsymbol{y})$, we have

$$S\left(\frac{1}{d}\boldsymbol{P}\right) > S\left(\frac{1}{d}\boldsymbol{P}\boldsymbol{T}^{\mathsf{T}}\right).$$

Conversely, we can set an new experiment $\boldsymbol{P}' = \boldsymbol{P}\boldsymbol{T}^{\mathsf{T}}$ and $\boldsymbol{x}' = \hat{\boldsymbol{x}}$ and $\hat{\boldsymbol{x}}' = \boldsymbol{x}$ so that the misreport matrix equals $\frac{1}{d}\boldsymbol{T}^{\mathsf{T}}$ and $\boldsymbol{x}' \succ_{\text{EXACT}}^{\boldsymbol{x}'} \hat{\boldsymbol{x}}'$. First, because $\boldsymbol{T}$ is a permutation $\boldsymbol{P}' = \boldsymbol{P}\boldsymbol{T}^{\mathsf{T}} \in \mathcal{P}_{\text{indep}}$ and the joint distributions becomes $\frac{1}{d}\boldsymbol{P}' = \frac{1}{d}\boldsymbol{P}\boldsymbol{T}^{\mathsf{T}}$ for $(\boldsymbol{x}', \boldsymbol{y}')$ and $\frac{1}{d}\boldsymbol{P}'\boldsymbol{T} = \frac{1}{d}\boldsymbol{P}\boldsymbol{T}^{\mathsf{T}}\boldsymbol{T} = \frac{1}{d}\boldsymbol{P}$ for $(\hat{\boldsymbol{x}}', \boldsymbol{y}')$. Therefore,

$$S\left(\frac{1}{d}\boldsymbol{P}\boldsymbol{T}^{\mathsf{T}}\right) > S\left(\frac{1}{d}\boldsymbol{P}\right)$$

which is a contradiction.

**Proof of Blackwell orderings in Proposition 3.1**  For Blackwell ordering, we further show that the existence of *any* linearly dependent experiment $\boldsymbol{P}$ (i.e. columns of $\boldsymbol{P}$ are linearly dependent) in $\mathcal{P}$ makes it impossible to preserve Blackwell ordering on $\mathcal{P}$ and $\mathcal{Q}_{\text{reg}}$. The $\mathcal{Q}_{\text{reg}}$ restriction rules out the possibility of improving data reliability through simple post-processing operations like (noisy) relabeling. In addition, recall that Blackwell ordering requires $\mathcal{Q}_{\text{reg}}$ to be a strict partial ordering.

The proof idea is similar to that of the exact ordering setting: it is impossible to detect misreporting when two labels induce identical observation distributions—i.e., when $\boldsymbol{P}$ has identical columns. The main challenge in Proposition 3.1, however, is to show that for any linearly dependent $\boldsymbol{P}$ (which may not have identical columns), we can construct a misreport matrix $\boldsymbol{Q}$ such that $\boldsymbol{P}\boldsymbol{Q}$ has identical columns.[6]

If we can find $\boldsymbol{P} \in \mathcal{P}$, a misreport matrix $\boldsymbol{Q}$, and column stochastic $\boldsymbol{T} \neq \mathbb{I}$ with $\boldsymbol{P}\boldsymbol{Q} = \boldsymbol{P}\boldsymbol{Q}\boldsymbol{T}^{\mathsf{T}}$, we have $\boldsymbol{x}, \hat{\boldsymbol{x}}, \hat{\boldsymbol{x}}'$ with misreport matrices $\boldsymbol{Q}$ and $\boldsymbol{Q}\boldsymbol{T}^{\mathsf{T}}$ so that $\hat{\boldsymbol{x}} \succ_{\text{Blackwell}}^{\boldsymbol{x}} \hat{\boldsymbol{x}}'$, but have the same joint distribution between reports and observations. Therefore, no score in the partial knowledge (and detail-free) setting can distinguish them and preserves the Blackwell ordering.

Now we construct $\boldsymbol{P}, \boldsymbol{Q}$, and $\boldsymbol{T}$. By the condition in Proposition 3.1 there exists $\boldsymbol{P} \in \mathcal{P}$ and $\boldsymbol{v} \neq \boldsymbol{0} \in \mathbb{Q}^d$ so that $\boldsymbol{P}\boldsymbol{v} = \boldsymbol{0}$. We decompose $\boldsymbol{v}$ as $\boldsymbol{v} = \boldsymbol{v}_+ - \boldsymbol{v}_-$ where $\boldsymbol{v}_+$ and $\boldsymbol{v}_-$ are nonnegative and have disjoint support, so

$$\boldsymbol{P}\boldsymbol{v}_+ = \boldsymbol{P}\boldsymbol{v}_- \tag{7}$$

and $\boldsymbol{v}_+, \boldsymbol{v}_- \neq \boldsymbol{0}$ because $\boldsymbol{P}$ is a collection of distributions. Let $\iota_+ \in [d]$ be the index of the largest entry in $\boldsymbol{v}_+$, and $\iota_-$ for $\boldsymbol{v}_-$ similarly, breaking ties arbitrarily. Note that $\iota_+ \neq \iota_-$, because $\boldsymbol{v}_+$ and $\boldsymbol{v}_-$ have disjoint supports. We first construct $\boldsymbol{A}$ by replacing the $\iota_+$ column of the identity

---

[6]We require $\boldsymbol{v} \in \mathbb{Q}^d$ to have rational coefficients to ensure the resulting $\boldsymbol{Q}$ has rational coefficients to be a valid misreport matrix.

matrix $\mathbb{I} \in \mathbb{R}^d$ by $\boldsymbol{v}_+$ and $\iota_-$ column by $\boldsymbol{v}_-$, and set $\boldsymbol{Q} = \frac{1}{Z}\boldsymbol{A}$ where $Z = \sum_{i,j} \boldsymbol{A}(i,j)$. This normalization ensures that $\boldsymbol{Q}$ forms a distribution as $\boldsymbol{v}_+$ and $\boldsymbol{v}_-$ are non-negative. By construction, $\boldsymbol{Q}$ is diagonally maximized by the choice of $\iota_+, \iota_-$, and invertible because $\boldsymbol{v}_+, \boldsymbol{v}_- \neq \boldsymbol{0}$ and using Gaussian elimination. Most importantly, the $\iota_+$ and $\iota_-$ columns of $\boldsymbol{P}\boldsymbol{Q}$ are identical by Eq. (7).

To complete the construction, given $\epsilon > 0$ we set $\boldsymbol{T} \neq \mathbb{I}$ so that

$$
\boldsymbol{T}(i,j) = \begin{cases}
1 & \text{if } i = j \text{ and } \{i,j\} \cap \{\iota_+, \iota_-\} = \emptyset \\
0 & \text{if } i \neq j \text{ and } \{i,j\} \cap \{\iota_+, \iota_-\} = \emptyset \\
\epsilon & \text{if } i = \iota_+, j = \iota_- \text{ or } i = \iota_-, j = \iota_+ \\
1 - \epsilon & \text{if } i = j \in \{\iota_+, \iota_-\} \\
0 & \text{if } i \neq j \text{ and } |\{i,j\} \cap \{\iota_+, \iota_-\}| = 1
\end{cases}
$$

which is the identical matrix excepts for the $\iota_+$ and $\iota_-$ columns and rows. Note that $\boldsymbol{T}$ is a column stochastic matrix, $\boldsymbol{Q}\boldsymbol{T}^\mathsf{T}$ is still invertible and diagonally maximal when $\epsilon$ is small enough. Finally, $\boldsymbol{P}\boldsymbol{Q}\boldsymbol{T}^\mathsf{T}$ mixes the $\iota_+$ and $\iota_-$ columns. However, because the $\iota_+$ and $\iota_-$ columns of $\boldsymbol{P}\boldsymbol{Q}$ are identical, $\boldsymbol{P}\boldsymbol{Q} = \boldsymbol{P}\boldsymbol{Q}\boldsymbol{T}^\mathsf{T}$ which completes our proof.

**Proof of Hamming and** dist **orderings in Proposition 3.1** Finally, we show that there does not exist a reliability score that preserves the Hamming and dist distance ordering, even restricting to diagonally dominant misreport matrices $\mathcal{Q}_{\text{dom}} \subset \mathcal{Q}_{\text{reg}}$.

We begin the proof with the Hamming ordering. Suppose we can find two settings: one has $\boldsymbol{Q}_1, \boldsymbol{Q}_1' \in \mathcal{Q}_{\text{dom}}$ and $\boldsymbol{P}_1 \in \mathcal{P}_{\text{indep}}$, the other has $\boldsymbol{Q}_2, \boldsymbol{Q}_2' \in \mathcal{Q}_{\text{dom}}$ and $\boldsymbol{P}_2 \in \mathcal{P}_{\text{indep}}$ so that

$$\text{Tr}(\boldsymbol{Q}_1) > \text{Tr}(\boldsymbol{Q}_1'), \text{Tr}(\boldsymbol{Q}_2) < \text{Tr}(\boldsymbol{Q}_2'), \text{ but } \boldsymbol{P}_1\boldsymbol{Q}_1 = \boldsymbol{P}_2\boldsymbol{Q}_2, \boldsymbol{P}_1\boldsymbol{Q}_1' = \boldsymbol{P}_2\boldsymbol{Q}_2'.$$

Then we can find $\boldsymbol{x}_1, \hat{\boldsymbol{x}}_1, \hat{\boldsymbol{x}}_1', \boldsymbol{x}_2, \hat{\boldsymbol{x}}_2, \hat{\boldsymbol{x}}_2'$ so that $\hat{\boldsymbol{x}}_1 \succ_{\text{Hamming}}^{\boldsymbol{x}_1} \hat{\boldsymbol{x}}_1'$ and $\hat{\boldsymbol{x}}_2' \succ_{\text{Hamming}}^{\boldsymbol{x}_2} \hat{\boldsymbol{x}}_2$ by setting the misreport matrix of $\boldsymbol{x}_1, \hat{\boldsymbol{x}}_1$ be $\boldsymbol{Q}_1$, the misreport matrix $\boldsymbol{x}_1, \hat{\boldsymbol{x}}_1'$ as $\boldsymbol{Q}_1'$, the misreport matrix of $\boldsymbol{x}_2, \hat{\boldsymbol{x}}_2$ be $\boldsymbol{Q}_2$, the misreport matrix $\boldsymbol{x}_2, \hat{\boldsymbol{x}}_2'$ as $\boldsymbol{Q}_2'$. If there is a reliability score that preserves the Hamming ordering on $\mathcal{P}_{\text{indep}}, \mathcal{Q}_{\text{dom}}$,

$$\mathbb{E}[S(\boldsymbol{P}_1\boldsymbol{Q}_1)] > \mathbb{E}[S(\boldsymbol{P}_1\boldsymbol{Q}_1')] \text{ and } \mathbb{E}[S(\boldsymbol{P}_2\boldsymbol{Q}_2)] < \mathbb{E}[S(\boldsymbol{P}_2\boldsymbol{Q}_2')] \tag{8}$$

which reaches a contradiction as $\boldsymbol{P}_1\boldsymbol{Q}_1 = \boldsymbol{P}_2\boldsymbol{Q}_2$ and $\boldsymbol{P}_1\boldsymbol{Q}_1' = \boldsymbol{P}_2\boldsymbol{Q}_2'$

We construct

$$
\boldsymbol{P}_1 = \begin{pmatrix} 0.74 & 0 & 0.26 \\ 0.26 & 0.74 & 0 \\ 0 & 0.26 & 0.74 \end{pmatrix}, \boldsymbol{Q}_1 = \frac{1}{3}\begin{pmatrix} 0.8 & 0 & 0.2 \\ 0.2 & 0.8 & 0 \\ 0 & 0.2 & 0.8 \end{pmatrix}, \quad \boldsymbol{Q}_1' = \frac{1}{3}\begin{pmatrix} 0.7 & 0.3 & 0 \\ 0 & 0.7 & 0.3 \\ 0.3 & 0 & 0.7 \end{pmatrix}.
$$

For the second setting, we define $\boldsymbol{P}_2 = \mathbb{I}$, and

$$
\boldsymbol{Q}_2 = \boldsymbol{P}_1\boldsymbol{Q}_1 = \frac{1}{3}\begin{pmatrix} 0.592 & 0.052 & 0.356 \\ 0.356 & 0.592 & 0.052 \\ 0.052 & 0.356 & 0.592 \end{pmatrix}
$$

$$
\boldsymbol{Q}_2' = \boldsymbol{P}_1\boldsymbol{Q}_1' = \frac{1}{3}\begin{pmatrix} 0.596 & 0.222 & 0.182 \\ 0.182 & 0.596 & 0.222 \\ 0.222 & 0.182 & 0.596 \end{pmatrix}
$$

Therefore, $\boldsymbol{P}_1\boldsymbol{Q}_1 = \boldsymbol{P}_2\boldsymbol{Q}_2$ and $\boldsymbol{P}_1\boldsymbol{Q}_1' = \boldsymbol{P}_2\boldsymbol{Q}_2'$. By direct computation, we have $\text{Tr}(\boldsymbol{Q}_1) = \frac{24}{30} > \text{Tr}(\boldsymbol{Q}_1') = \frac{21}{30}$ and $\text{Tr}(\boldsymbol{Q}_2) = \frac{1776}{3000} < \text{Tr}(\boldsymbol{Q}_2') = \frac{1788}{3000}$. Finally, note that we can easily generalize this construction beyond three dimensions by padding the other dimension with identity.

Interestingly, the same construction works for general dist-ordering, due to the symmetry in $\boldsymbol{Q}_1, \boldsymbol{Q}_1', \boldsymbol{Q}_2$ and $\boldsymbol{Q}_2'$. First note that $\sum_{n=1}^N \text{dist}(\hat{x}_n, x_n) = N\sum_{i,j\in[d]} \boldsymbol{Q}(i,j)\,\text{dist}(i,j) = N\langle \boldsymbol{Q}, \text{dist}\rangle_F$ where $\langle \cdot, \cdot \rangle_F$ is the Frobenius inner product defined in Appendix A. Hence, with Eq. (8), it is sufficient to show the above construction satisfies

$$\langle \boldsymbol{Q}_1, \text{dist}\rangle_F > \langle \boldsymbol{Q}_1', \text{dist}\rangle_F \text{ and } \langle \boldsymbol{Q}_2, \text{dist}\rangle_F < \langle \boldsymbol{Q}_2', \text{dist}\rangle_F.$$

Let $A = \text{dist}(1,2) + \text{dist}(2,3) + \text{dist}(3,1) = \text{dist}(1,3) + \text{dist}(2,1) + \text{dist}(3,2) > 0$ as $\text{dist}(x, x') = \text{dist}(x', x)$ for all $x, x'$. By symmetry, we note the Frobenius inner product only depends on $A$,

$$\langle \boldsymbol{Q}_1, \text{dist} \rangle_F - \langle \boldsymbol{Q}_1', \text{dist} \rangle_F = \frac{1}{3}(0.2A - 0.3A) < 0 \qquad (\text{dist}(x,x) = 0 \text{ for all } x)$$

$$\langle \boldsymbol{Q}_2, \text{dist} \rangle_F - \langle \boldsymbol{Q}_2', \text{dist} \rangle_F = \frac{1}{3}(0.408A - 0.404A) > 0$$

which completes the proof.

# D  PROOFS AND DETAILS IN SECTION 4.1

*Proof of Eq. (4).* For all $\hat{x}, \hat{x}' \in \mathcal{X}$,

$$\hat{\boldsymbol{G}}(\hat{x}, \hat{x}') = \frac{1}{N^2} \sum_{n,n': \hat{x}_n = x, \hat{x}_{n'} = x'} \langle P_{x_n}, P_{x_{n'}} \rangle$$

$$= \frac{1}{N^2} \sum_{x,x' \in \mathcal{X}} \sum_{\substack{n,n': \\ \hat{x}_n = \hat{x}, \hat{x}_{n'} = \hat{x}', \\ x_n = x, x_{n'} = x'}} \boldsymbol{G}(x, x')$$

$$= \sum_{x,x' \in \mathcal{X}} \boldsymbol{Q}(x, \hat{x}) \boldsymbol{G}(x, x') \boldsymbol{Q}(x', \hat{x}')$$

which proves Eq. (4). $\square$

## D.1  AN EXAMPLE OF GRAM DETERMINANT SCORE

Here we provide a simple example for Gram determinant score. Consider $\mathcal{X} = \mathcal{Y} = \{1, 2\}$, $\boldsymbol{P} = \begin{pmatrix} 1 - p_1 & 1 - p_2 \\ p_1 & p_2 \end{pmatrix}$, and the misreport matrix $\boldsymbol{Q} = \begin{pmatrix} \frac{1-\delta}{4} & \frac{\delta}{4} \\ \frac{\delta}{4} & \frac{1-\delta}{4} \end{pmatrix}$ with $\delta \geq 0$ where $\boldsymbol{x} = \hat{\boldsymbol{x}}$ if $\delta = 0$ whereas increasing $\delta$ makes the reports less reliable. By Eq. (4) and direct computation, the Gram determinant score is

$$\det(\hat{\boldsymbol{G}}) = \det(\boldsymbol{Q}^\intercal) \det(\boldsymbol{G}) \det(\boldsymbol{Q})$$
$$= \det(\boldsymbol{P})^2 \det(\boldsymbol{Q})^2 \tag{9}$$
$$= \frac{1}{2^8}(p_1 - p_2)^2 (1 - 2\delta)^2.$$

Given a fixed experiment $\boldsymbol{P}$, the Gram determinant score Eq. (9) decreases as $\delta$ increases from $\delta = 0$ to $1/2$. In particular, it maximizes at $\delta = 0$, when the reported data exactly match the true data, and drops to zero at $\delta = 1/2$, where all reports contain the same uniform mixture of the true labels. Additionally, the score also depends on the quality of the experiment $\boldsymbol{P}$. If $p_1 = p_2$, columns of $\boldsymbol{P}$ are linearly dependent and Gram determinant score become zero. In contrast, if $p_1 \neq p_2$ and $\delta < 1/2$, the score is strictly positive.

## D.2  LEMMAS AND PROOFS FOR THEOREM 4.2

*Proof of Theorem 4.2.* The key idea of proving Theorem 4.2 is that the determinant has multiplicative property Eq. (4) which allows us to decouple the misreport matrix $\boldsymbol{Q}$ from the quality of the experiment $\boldsymbol{P}$, and $\det(\boldsymbol{G}) = \det(\boldsymbol{P}^\intercal \boldsymbol{P}) > 0$, for all $\boldsymbol{P} \in \mathcal{P}_{\text{indep}}$. Therefore,

$$\Gamma > \Gamma' \text{ if and only if } \det(\boldsymbol{Q}^\intercal \boldsymbol{Q}) > \det((\boldsymbol{Q}')^\intercal \boldsymbol{Q}').$$

Thus, the following Lemmas D.1 and D.2 prove the first and second cases. For the third case, Lemma D.3 proves the score preserves the approximate Hamming ordering, as $\Delta = 1$ for Hamming distance. For general distance, let $\text{Hamming}(\boldsymbol{x}, \hat{\boldsymbol{x}}) = \sum_n \mathbf{1}[\hat{x}_n \neq x_n]$ and $\text{dist}(\boldsymbol{x}, \hat{\boldsymbol{x}}) = \sum_n \text{dist}(\hat{x}_n, x_n)$ be the Hamming distance and dist between $\hat{\boldsymbol{x}}$ and $\boldsymbol{x}$. Because

$$\min_{x \neq x'} \text{dist}(x, x') \, \text{Hamming}(\hat{\boldsymbol{x}}, \boldsymbol{x}) \leq \text{dist}(\boldsymbol{x}, \hat{\boldsymbol{x}}) \leq \max_{x, x'} \text{dist}(x, x') \, \text{Hamming}(\hat{\boldsymbol{x}}, \boldsymbol{x}),$$

and $\Delta = \frac{\max_{x,x'} \text{dist}(x,x')}{\min_{x \neq x'} \text{dist}(x,x')}$, $\hat{\boldsymbol{x}} \succ_{\text{dist}, 1/(4\Delta L)}^{\boldsymbol{x}} \hat{\boldsymbol{x}}'$ implies $\hat{\boldsymbol{x}} \succ_{\text{Hamming}, 1/(4L)}^{\boldsymbol{x}} \hat{\boldsymbol{x}}'$, which completes the proof. $\square$

**Lemma D.1.** *For all $\boldsymbol{x}, \hat{\boldsymbol{x}}, \hat{\boldsymbol{x}}'$ if $\hat{\boldsymbol{x}} \succ_{\mathrm{EXACT}}^{\boldsymbol{x}} \hat{\boldsymbol{x}}'$ and $\boldsymbol{Q}, \boldsymbol{Q}' \in \mathcal{Q}_{\mathit{nonperm}}$, $\det(\boldsymbol{Q}^{\mathsf{T}}\boldsymbol{Q}) > \det((\boldsymbol{Q}')^{\mathsf{T}}\boldsymbol{Q}')$.*

*Proof of Lemma D.1.* As $\boldsymbol{x}, \hat{\boldsymbol{x}}$, and $\hat{\boldsymbol{x}}'$ with $\hat{\boldsymbol{x}} \succ_{\mathrm{EXACT}}^{\boldsymbol{x}} \hat{\boldsymbol{x}}'$, $\boldsymbol{Q}_{\hat{\boldsymbol{x}}|\boldsymbol{x}} = \mathbb{I}$ and there is $\boldsymbol{T} \neq \mathbb{I}$ so that $\boldsymbol{Q}'_{\hat{\boldsymbol{x}}|\boldsymbol{x}} = \boldsymbol{T}\boldsymbol{Q}_{\hat{\boldsymbol{x}}|\boldsymbol{x}} = \boldsymbol{T}$. By Eq. (1) we have $\boldsymbol{Q}' = \boldsymbol{Q}\boldsymbol{T}^{\mathsf{T}} = \boldsymbol{Q}_{\boldsymbol{x}}\boldsymbol{T}^{\mathsf{T}}$ and $\boldsymbol{Q} = \boldsymbol{Q}_{\boldsymbol{x}}$. Therefore

$$\det((\boldsymbol{Q}')^{\mathsf{T}}\boldsymbol{Q}') = \det(\boldsymbol{T}\boldsymbol{Q}^{\mathsf{T}}\boldsymbol{Q}\boldsymbol{T}^{\mathsf{T}}) = \det(\boldsymbol{T}\boldsymbol{T}^{\mathsf{T}})\det(\boldsymbol{Q}^{\mathsf{T}}\boldsymbol{Q}) \tag{10}$$

Because the diagonal matrix $\boldsymbol{Q} = \boldsymbol{Q}_{\boldsymbol{x}}$ has positive diagonals, and $\boldsymbol{T}$ is column stochastic and not a permutation matrix, the Perron–Frobenius theorem (or (Kong, 2020)) implies $|\det(\boldsymbol{T})| < 1$ and $\det((\boldsymbol{Q}')^{\mathsf{T}}\boldsymbol{Q}') = \det(\boldsymbol{T}\boldsymbol{T}^{\mathsf{T}})\det(\boldsymbol{Q}^{\mathsf{T}}\boldsymbol{Q}) < \det(\boldsymbol{Q}^{\mathsf{T}}\boldsymbol{Q})$. $\qquad\square$

**Lemma D.2.** *For all $\boldsymbol{x}, \hat{\boldsymbol{x}}, \hat{\boldsymbol{x}}'$ if $\hat{\boldsymbol{x}} \succ_{\mathrm{Blackwell}}^{\boldsymbol{x}} \hat{\boldsymbol{x}}'$ and $\boldsymbol{Q}, \boldsymbol{Q}' \in \mathcal{Q}_{\mathit{reg}}$, $\det(\boldsymbol{Q}^{\mathsf{T}}\boldsymbol{Q}) > \det((\boldsymbol{Q}')^{\mathsf{T}}\boldsymbol{Q}')$.*

*Proof of Lemma D.2.* As $\hat{\boldsymbol{x}} \succ_{\mathrm{Blackwell}}^{\boldsymbol{x}} \hat{\boldsymbol{x}}'$, there is a column stochastic $\boldsymbol{T} \neq \mathbb{I}$ so that $\boldsymbol{Q}'_{\hat{\boldsymbol{x}}|\boldsymbol{x}} = \boldsymbol{T}\boldsymbol{Q}_{\hat{\boldsymbol{x}}|\boldsymbol{x}}$. By Eq. (10),

$$\det((\boldsymbol{Q}')^{\mathsf{T}}\boldsymbol{Q}') = \det(\boldsymbol{T}\boldsymbol{Q}^{\mathsf{T}}\boldsymbol{Q}\boldsymbol{T}^{\mathsf{T}}) = \det(\boldsymbol{T}\boldsymbol{T}^{\mathsf{T}})\det(\boldsymbol{Q}^{\mathsf{T}}\boldsymbol{Q})$$

Because $\boldsymbol{Q} \in \mathcal{Q}_{\mathrm{reg}}$ is invertible, $\det(\boldsymbol{Q}) \neq 0$. By Lemma A.3, $\boldsymbol{T}$ is not a permutation matrix, so $|\det(\boldsymbol{T})| < 1$, and $\det((\boldsymbol{Q}')^{\mathsf{T}}\boldsymbol{Q}') = \det(\boldsymbol{T}\boldsymbol{T}^{\mathsf{T}})\det(\boldsymbol{Q}^{\mathsf{T}}\boldsymbol{Q}) < \det(\boldsymbol{Q}^{\mathsf{T}}\boldsymbol{Q})$.

$\qquad\square$

**Lemma D.3.** *Given $\mathcal{X} = [d]$ and $L \geq 1$, for all $\boldsymbol{x}, \hat{\boldsymbol{x}}, \hat{\boldsymbol{x}}'$ if $\hat{\boldsymbol{x}} \succ_{\mathrm{Hamming}, \frac{1}{4L}}^{\boldsymbol{x}} \hat{\boldsymbol{x}}'$ and $\boldsymbol{Q}, \boldsymbol{Q}' \in \mathcal{Q}_{L, 1/(64L^2 d^2)}$, $\det(\boldsymbol{Q}^{\mathsf{T}}\boldsymbol{Q}) > \det((\boldsymbol{Q}')^{\mathsf{T}}\boldsymbol{Q}')$.*

**Proof of Lemma D.3** Lemma D.3 establishes that the Gram determinant score approximately preserves the Hamming ordering under balancedness and small Hamming distance conditions. The main technical challenge lies in deriving upper and lower bounds on the Gram determinant in terms of the Hamming distance Lemma D.4.

**Lemma D.4.** *For all $\delta \geq 0$, and $\boldsymbol{x}, \hat{\boldsymbol{x}}$ with diagonally dominant $\boldsymbol{Q}$, if $\delta = 1 - \mathrm{Tr}(\boldsymbol{Q})$ and $\delta < \frac{\min_i q_{\boldsymbol{x}}(i)}{4}$,*

$$\left(1 - \frac{8d\delta^2}{\min_i q_{\boldsymbol{x}}(i)^2}\right)\left(1 - \frac{\delta}{\min_i q_{\boldsymbol{x}}(i)}\right) \leq \frac{\det(\boldsymbol{Q})}{\prod_i q(i)} \leq \left(1 + \frac{8d\delta^2}{\min_i q_{\boldsymbol{x}}(i)^2}\right)\left(1 - \frac{\delta}{2\max_i q_{\boldsymbol{x}}(i)}\right).$$

**Lemma D.5.** *Given $L \geq 1$, if $a_1, \ldots, a_d \geq 0$, $\sum_{i \in [d]} a_i = 1$ and $a_i \leq La_j$ for all $i, j \in [d]$, then*

$$\frac{1}{Ld - L + 1} \leq a_i \leq \frac{L}{d + L - 1}, \text{ for all } i$$

*Proof of Lemma D.3.* If $\boldsymbol{x}, \hat{\boldsymbol{x}}, \hat{\boldsymbol{x}}'$ with $\boldsymbol{Q}, \boldsymbol{Q}' \in \mathcal{Q}_{L, 1/(64L^2 d^2)}$, the true labels are $L$ balanced, and Hamming distances $\delta = 1 - \mathrm{Tr}(\boldsymbol{Q}), \delta' = 1 - \mathrm{Tr}(\boldsymbol{Q}')$ are less than $\frac{1}{64L^2 d^2}$. If $\hat{\boldsymbol{x}} \succ_{\mathrm{Hamming}, 1/(4L)}^{\boldsymbol{x}} \hat{\boldsymbol{x}}'$, we want to show $\left(\frac{\det(\boldsymbol{Q})}{\det(\boldsymbol{Q}')}\right)^2 > 1$.

Note that as $\boldsymbol{Q}, \boldsymbol{Q}'$ are diagonally dominant $\det(\boldsymbol{Q}), \det(\boldsymbol{Q}') > 0$, and we use Lemma D.4 to show that the lower bound of $\det(\boldsymbol{Q})$ is larger than the upper bound of $\det(\boldsymbol{Q}')$,

$$\left(1 + \frac{8d(\delta')^2}{\min_i q_{\boldsymbol{x}}(i)^2}\right)\left(1 - \frac{\delta'}{2\max_i q_{\boldsymbol{x}}(i)}\right) < \left(1 - \frac{8d\delta^2}{\min_i q_{\boldsymbol{x}}(i)^2}\right)\left(1 - \frac{\delta}{\min_i q_{\boldsymbol{x}}(i)}\right).$$

By taking the difference, we have

$$\left(1 - \frac{8d\delta^2}{\min_i q_{\boldsymbol{x}}(i)^2}\right)\left(1 - \frac{\delta}{\min_i q_{\boldsymbol{x}}(i)}\right) - \left(1 + \frac{8d(\delta')^2}{\min_i q_{\boldsymbol{x}}(i)^2}\right)\left(1 - \frac{\delta'}{2\max_i q_{\boldsymbol{x}}(i)}\right)$$

$$> \frac{\delta'}{2\max_i q_{\boldsymbol{x}}(i)} - \frac{8d\delta^2}{\min_i q_{\boldsymbol{x}}(i)^2} - \frac{\delta}{\min_i q_{\boldsymbol{x}}(i)} - \frac{8d(\delta')^2}{\min_i q_{\boldsymbol{x}}(i)^2}$$

(The second order terms are positive)

$$\geq \frac{\delta'}{2\max_i q_{\boldsymbol{x}}(i)} - \frac{\delta}{\min_i q_{\boldsymbol{x}}(i)} - \frac{16d(\delta')^2}{\min_i q_{\boldsymbol{x}}(i)^2} \qquad (\delta < \delta')$$

$$\geq \frac{\delta'}{4\max_i q_{\boldsymbol{x}}(i)} - \frac{16d(\delta')^2}{\min_i q_{\boldsymbol{x}}(i)^2} \qquad (\delta' > 4L\delta)$$

$$= \frac{\delta'}{4\max_i q_{\boldsymbol{x}}(i)}\left(1 - \frac{64d\max_i q_{\boldsymbol{x}}(i)\delta'}{\min_i q_{\boldsymbol{x}}(i)^2}\right)$$

$$\geq \frac{\delta'}{4\max_i q_{\boldsymbol{x}}(i)}\left(1 - \frac{64Ld}{\min_i q_{\boldsymbol{x}}(i)}\delta'\right) \qquad (\max_i q_{\boldsymbol{x}}(i) < L\min_i q_{\boldsymbol{x}}(i))$$

$$> 0 \qquad (\delta' < \frac{1}{64L^2d^2} \text{ and } \min_i q_{\boldsymbol{x}}(i) \geq \frac{1}{Ld} \text{ by Lemma D.5})$$

$\square$

*Proof of Lemma D.4.* We want to estimate $\det(\boldsymbol{Q})$ by the Hamming distance. Let $\boldsymbol{Q} = \boldsymbol{D} + \boldsymbol{E}$ where $\boldsymbol{D}$ is a diagonal matrix and $\boldsymbol{E}$ has zero diagonal, and $\delta_i = \sum_{j \neq i} \boldsymbol{E}(i,j) = q_{\boldsymbol{x}}(i) - \boldsymbol{D}(i,i) \geq 0$ for all $i \in \mathcal{X}$ which is the off-diagonal weight of row $i$. With above notations, $1 - \mathrm{Tr}(\boldsymbol{Q}) = \sum_{i \in \mathcal{X}} \delta_i = \delta$ and $\det(\boldsymbol{D}) = \prod(q_{\boldsymbol{x}}(i) - \delta_i)$. If $\rho = \|\boldsymbol{D}^{-1}\boldsymbol{E}\|_2$ and $\delta_Q = -\rho d \ln(1 - \rho)$, by Theorem A.1

$$1 - \delta_Q \leq \frac{\det(\boldsymbol{Q})}{\det(\boldsymbol{D})} \leq 1 + \delta_Q. \tag{11}$$

As $\boldsymbol{D}^{-1}\boldsymbol{E}$ is a nonnegative matrix, by Gershgorin theorem, the spectral radius $\rho$ can be bounded by the row sum $\delta_i/\boldsymbol{Q}(i,i) \leq \frac{2\delta}{\min_i q_{\boldsymbol{x}}(i)}$ since $\boldsymbol{Q}$ is diagonally dominant. Because $-\ln(1-t) \leq 2t$ for all $t < 1/2$ and $\delta \leq \frac{\min_i q_{\boldsymbol{x}}(i)}{4}$, we have

$$\delta_Q \leq 2d\rho^2 \leq \frac{8d\delta^2}{\min_i q_{\boldsymbol{x}}(i)^2} \tag{12}$$

Now we bound the ratio $\frac{\det(\boldsymbol{D})}{\prod_i q_{\boldsymbol{x}}(i)} = \prod_i \left(1 - \frac{\delta_i}{q_{\boldsymbol{x}}(i)}\right)$. By union bound,

$$\prod_i \left(1 - \frac{\delta_i}{q_{\boldsymbol{x}}(i)}\right) \geq 1 - \sum_i \frac{\delta_i}{q_{\boldsymbol{x}}(i)} \geq 1 - \frac{\delta}{\min_i q_{\boldsymbol{x}}(i)} \qquad (\delta = \sum \delta_i)$$

On the other hand,

$$\prod_i \left(1 - \frac{\delta_i}{q_{\boldsymbol{x}}(i)}\right) \leq \exp\left(-\sum \frac{\delta_i}{q_{\boldsymbol{x}}(i)}\right) \qquad (1 - t \leq e^{-t} \text{ for all } t)$$

$$\leq \exp\left(-\frac{\delta}{\max_i q_{\boldsymbol{x}}(i)}\right) \qquad (\delta = \sum \delta_i)$$

$$\leq 1 - \frac{\delta}{2\max_i q_{\boldsymbol{x}}(i)} \qquad (\delta < \max q_{\boldsymbol{x}}(i) \text{ and } e^{-t} \leq 1 - \frac{1}{2}t \text{ if } 0 \leq t \leq 1)$$

Therefore,

$$1 - \frac{\delta}{\min_i q_{\boldsymbol{x}}(i)} \leq \frac{\det(\boldsymbol{D})}{\prod_i q(i)} \leq 1 - \frac{\delta}{2\max_i q_{\boldsymbol{x}}(i)} \tag{13}$$

By Eqs. (11) and (13), we have

$$(1 - \delta_Q)\left(1 - \frac{\delta}{\min_i q_{\boldsymbol{x}}(i)}\right) \leq \frac{\det(\boldsymbol{Q})}{\prod_i q(i)} \leq (1 + \delta_Q)\left(1 - \frac{\delta}{2\max_i q_{\boldsymbol{x}}(i)}\right)$$

which completes the proof by plugging in Eq. (12) $\square$

*Proof of Lemma D.5.* Because $a_j \geq \frac{1}{L} a_i$ for all $i \neq j$, $1 = \sum a_j \geq a_i + \frac{d-1}{L} a_i \leq \frac{L+d-1}{L} a_i$, and

$$a_i \leq \frac{L}{L+d-1}.$$

On the other hand, because $a_j \leq L a_i$, $1 = \sum a_j \leq a_i + (d-1) L a_i$, and

$$a_i \geq \frac{1}{Ld - L + 1}.$$

$\square$

## D.3 PROOFS FOR EXPERIMENT AGNOSTIC

*Proof of Proposition 4.3.* We first show that there is $\alpha = 1/S(\mathbb{I}) > 0$ so that for all $P, Q \in GL_d$

$$S(PQ) = \alpha S(P) S(Q). \tag{14}$$

Since $S$ is experiment agonistic, given any $P$, $S(PQ)$ is increasing in $S(Q)$, and there exists an increasing function $g_P$ so that $S(PQ) = g_P(S(Q))$ Because for any $s, t > 0$ and $Q$, $S(stQ) = c(st)S(Q) = c(s)c(t)S(Q)$ and $S(Q) > 0$, we have $c(st) = c(s)c(t)$ for all $s, t > 0$. Therefore,

$$c(t) = t^\gamma \text{ for some } \gamma \in \mathbb{R}. \tag{15}$$

For any $t > 0$ and $P, Q$, we have $S(PtQ) = c(t)S(PQ) = c(t)g_P(S(Q))$, and $S(PtQ) = g_P(S(tQ)) = g_P(c(t)S(Q))$. Hence

$$g_P(c(t)S(Q)) = c(t)g_P(S(Q)).$$

For any $P$ and $Q$, we have

$$S(PQ) = g_P\left(S(Q) \cdot \frac{1}{S(Q)} S(Q)\right) = S(Q)g_P(1)$$

by Eq. (15) and taking $t = S(Q)^{-\gamma}$. By taking $Q = \mathbb{I}$ we have $g_P(1) = \frac{S(PQ)}{S(Q)} = \frac{S(P)}{S(\mathbb{I})}$, and prove Eq. (14).

By Eq. (14), $\tilde{S}(Q) := \alpha S(Q)$ is a continuous homomorphism between $GL_d$ and $(\mathbb{R}_{>0}, \cdot)$ so that for all $P, Q$ $\tilde{S}(PQ) = \tilde{S}(P)\tilde{S}(Q)$. Thus, there exists a continuous $f : \mathbb{R} \setminus \{0\} \to \mathbb{R}_{>0}$ so that $\tilde{S}(Q) = f(\det(Q))$. (user856) We now pin down the function $f$. First, $S(tQ) = \alpha f(t^d \det(Q))$ and by Eq. (15), $S(tQ) = \alpha c(t)f(\det(Q)) = \alpha t^\gamma f(\det(Q))$ for all $t > 0$ and $Q$. Given $\beta = \gamma/(2d)$, for all $t > 0$, $f(t) = t^{2\beta} f(1)$ and $f(-t) = t^{2\beta} f(-1)$. Moreover, because $f$ is a homomorphism $f(-1)^2 = f((-1)^2) = f(1) = 1$ and $f(-1) > 0$, we have for all $z \neq 0$, $f(z) = |z|^{2\beta}$ and

$$S(Q) = \alpha f(\det(Q)) = \alpha |\det(Q)|^{2\beta} = \alpha \det(Q^\intercal Q)^\beta.$$

$\square$

## E LEMMAS AND PROOFS FOR SECTION 4.2

**Stratified matching estimator** While the above plug-in estimate can asymptotically preserve all reliability ordering in Theorem 4.2, it lacks provable guarantees for data of finite size. In practice, only a limited number of observations are available, and the data source can be strategic and aims to maximize its reliability score. Definition E.1 provides an estimator that preserves the exact match ordering using finite data which rewards truthful reporting than any other reports.

**Definition E.1.** *Given $\mathcal{X} = [d]$, and $\hat{x}, y$ of size $N$, a* stratified matching estimator *for the Gram determinant score is defined as the following*

1. *Return 0 if the minimum occurrence $\min_{x \in \mathcal{X}} |\{n \in [N] : \hat{x}_n = x\}|$ is less than 2. Otherwise, we randomly select two disjoint index sets $Col, Row \subseteq [N]$ of size $d$ where each label $i \in \mathcal{X}$ occurs in each set exactly once. Then re-index them as two sequences of pairs $(\hat{x}_{i,Col}, y_{i,Col})_{i \in [d]}$ and $(\hat{x}_{i,Row}, y_{i,Row})_{i \in [d]}$ so that $\hat{x}_{i,Col} = \hat{x}_{i,Row} = i$ for all $i \in \mathcal{X}$. We call the first as column sequence and the second as row sequence.*

2. *Randomly sample one permutation $\sigma \in sym(d)$, and return*

$$score(\hat{\boldsymbol{x}}, \boldsymbol{y}) := d! sgn(\sigma) \prod_{i,j \in [d], j = \sigma(i)} \mathbf{1}\left[y_{i,Row} = y_{j,Col}\right] \boldsymbol{q}_{\hat{\boldsymbol{x}}}(i) \boldsymbol{q}_{\hat{\boldsymbol{x}}}(j). \quad (16)$$

Equation (16) approximates the second form of the Gram determinant in Eq. (3) by summing over all permutations. The first step is a stratified sampling to collect one report of each label in $Col$ and $Row$ respectively. The term $\mathbf{1}[y_{i,Row} = y_{j,Col}]$ approximates the inner product between the observation distributions of reports $i$ and $j$, and the extra $\boldsymbol{q}_{\hat{\boldsymbol{x}}}(i) \boldsymbol{q}_{\hat{\boldsymbol{x}}}(j)$ offset the stratified sampling.

The stratified-matching estimator only requires each label to have at least two true data points. If any label occurs fewer than two times, the estimator returns zero and yields a worse score than truthful data. The following result shows that under mild balance conditions, the stratified-matching estimator preserves exact match ordering over linearly independent experiments.

**Proposition E.2.** *Given $\mathcal{X} = [d]$ and $L \geq 1$, the stratified-matching estimator in Definition E.1 preserves exact ordering on $\mathcal{P}_{indep}$, $\mathcal{Q}_L$, and $N = 2Ld$.*

**Proofs for Proposition 4.5**

**Lemma E.3.** *Given $\delta > 0$ and report size $N$,*

$$\Pr\left[\|\bar{\boldsymbol{G}} - \hat{\boldsymbol{G}}\|_2 \leq 4\sqrt{\frac{\log 2d/\delta}{N}} + 2\frac{\log 2d/\delta}{N}\right] \geq 1 - \delta.$$

*Proof of Proposition 4.5.* By Theorem A.2, we have

$$\frac{|\det(\bar{\boldsymbol{G}}) - \det(\hat{\boldsymbol{G}})|}{\det(\hat{\boldsymbol{G}})} \leq \left(1 + \|\hat{\boldsymbol{G}}^{-1}\|_2 \|\bar{\boldsymbol{G}} - \hat{\boldsymbol{G}}\|_2\right)^d - 1.$$

Hence with Lemma E.3 and $\delta = 1/N$, we have $\frac{|\det(\bar{\boldsymbol{G}}) - \det(\hat{\boldsymbol{G}})|}{\det(\hat{\boldsymbol{G}})} = o(1)$, with probability greater than $1 - 1/N$. Additionally, because the random variable $\det(\bar{\boldsymbol{G}})$ is always bounded by 1, the expectation

$$\mathbb{E}[\det(\bar{\boldsymbol{G}})] = (1 + o(1)) \det(\hat{\boldsymbol{G}}) \quad (17)$$

when $\hat{\boldsymbol{G}}^{-1}$ exists. If $\hat{\boldsymbol{G}}$ is not invertible, by (Ipsen & Rehman, 2008, Theorem 2.12),

$$|\det(\bar{\boldsymbol{G}}) - \det(\hat{\boldsymbol{G}})| \leq d\|\bar{\boldsymbol{G}} - \hat{\boldsymbol{G}}\|_2 \max\{\|\hat{\boldsymbol{G}}\|_2, \|\bar{\boldsymbol{G}}\|_2\}^{d-1} = o(1). \quad (18)$$

For all $\boldsymbol{P}$ and $\boldsymbol{Q}, \boldsymbol{Q}'$, if $\det(\hat{\boldsymbol{G}}) = \det(\boldsymbol{Q}^\intercal \boldsymbol{G} \boldsymbol{Q}) > \det((\boldsymbol{Q}')^\intercal \boldsymbol{G} \boldsymbol{Q}') = \det(\hat{\boldsymbol{G}}') > 0$, by Eq. (17) there exists a large enough $N_0$ so that any $\boldsymbol{x}, \hat{\boldsymbol{x}}, \hat{\boldsymbol{x}}'$ with length at least $N_0$ and $\boldsymbol{Q}, \boldsymbol{Q}'$ so that $\mathbb{E}[\det(\bar{\boldsymbol{G}})] > \mathbb{E}[\det(\bar{\boldsymbol{G}}')]$. Similar argument holds for $\det(\hat{\boldsymbol{G}}) > \det(\hat{\boldsymbol{G}}') = 0$ by Eq. (18). Therefore, the plug-in estimator asymptotically preserves all reliability orderings as Theorem 4.2. $\square$

*Proof of Lemma E.3.* Let $N_i = N q_{\hat{\boldsymbol{x}}}(i)$ be the number of report $i$ which is nonzero as $\boldsymbol{Q} \in \mathcal{Q}_{\text{reg}}$. Let $|\mathcal{Y}| = k$, and we can set $\phi : \mathcal{Y} \to \mathbb{R}^k$ be the delta vector $y \mapsto \mathbf{1}_y$. We define $\bar{\boldsymbol{v}}_i = \sum_{n:\hat{x}_n = i} \phi(y_n)$ and $\boldsymbol{v}_i = \mathbb{E}[\bar{\boldsymbol{v}}_i]$ as the sum of (empirical) mean of report $i \in \mathcal{X}$, and error $\boldsymbol{e}_i = \bar{\boldsymbol{v}}_i - \boldsymbol{v}_i$. Hence for all $i, j$, $\bar{\boldsymbol{G}}(i,j) = \frac{1}{N^2} \sum_{n,n':\hat{x}_n = i, \hat{x}_{n'} = j} \langle \phi(y_n), \phi(y_{n'}) \rangle = \frac{1}{N^2} \bar{\boldsymbol{v}}_i^\intercal \bar{\boldsymbol{v}}_j$, $\hat{\boldsymbol{G}}(i,j) = \frac{1}{N^2} \boldsymbol{v}_i^\intercal \boldsymbol{v}_j$, and

$$\bar{\boldsymbol{G}}(i,j) - \hat{\boldsymbol{G}}(i,j) = \frac{1}{N^2} \left(\boldsymbol{v}_i^\intercal \boldsymbol{e}_j + \boldsymbol{e}_i^\intercal \boldsymbol{v}_j + \boldsymbol{e}_i^\intercal \boldsymbol{e}_j\right) \quad (19)$$

To bound the spectral norm of $\bar{\boldsymbol{G}} - \hat{\boldsymbol{G}} \in \mathbb{R}^{d \times d}$, for any $a \in \mathbb{R}^d$ with $\|a\|_2 = 1$, we define $\boldsymbol{v}(a) = \sum_i a_i \boldsymbol{v}_i$, $\boldsymbol{e}(a) = \sum_i a_i \boldsymbol{e}_i \in \mathbb{R}^k$, and $R_{\boldsymbol{v}} = \sup_{\|a\|=1} \|\boldsymbol{v}(a)\|$, $R_{\boldsymbol{e}} = \sup_{\|a\|=1} \|\boldsymbol{e}(a)\|$. By Eq. (19)

$$a^\intercal (\bar{\boldsymbol{G}} - \hat{\boldsymbol{G}}) a = \frac{1}{N^2} (2\boldsymbol{v}(a)^\intercal \boldsymbol{e}(a) + \boldsymbol{e}(a)^\intercal \boldsymbol{e}(a)) \leq \frac{1}{N^2} (2R_{\boldsymbol{v}} R_{\boldsymbol{e}} + R_{\boldsymbol{e}}^2). \quad (20)$$

We first bound $R_{\boldsymbol{v}}$. For all $a$ with $\|a\| = 1$, let $\boldsymbol{V} = N^2 \hat{\boldsymbol{G}} \in \mathbb{R}^{d \times d}$ where $\boldsymbol{V}(i,j) = \boldsymbol{v}_i^\intercal \boldsymbol{v}_j$ which is positive semi definite

$$
\begin{aligned}
\|\boldsymbol{v}(a)\|^2 &= \sum_{i,j} a_i a_j \boldsymbol{v}_i^\intercal \boldsymbol{v}_j \\
&= a^\intercal \boldsymbol{V} a \\
&\leq \sum_i \boldsymbol{v}_i^\intercal \boldsymbol{v}_i && \text{(Rayleigh quotient is upper bounded by the trace)} \\
&= \sum_i N^2 \hat{\boldsymbol{G}}(i,i) && \text{(definition of } \boldsymbol{v}_i) \\
&\leq \sum_i N^2 q_{\hat{\boldsymbol{x}}}(i)^2 && \text{(because } \langle P_x, P_{x'} \rangle \leq 1 \text{ for any } x, x') \\
&\leq N^2 \max_i q_{\hat{\boldsymbol{x}}}(i)
\end{aligned}
$$

Therefore,

$$
R_{\boldsymbol{v}} \leq N \sqrt{\max_i q_{\hat{\boldsymbol{x}}}(i)} \leq N \tag{21}
$$

We bound $R_{\boldsymbol{e}}$ using Chernoff bound. For each $i \in \mathcal{X}$, $\boldsymbol{e}_i = \bar{\boldsymbol{v}}_i - \boldsymbol{v}_i = \sum_{n:\hat{x}_n = i} \phi(y_n) - \mathbb{E}\phi(y_n)$ is sum of $N q_{\boldsymbol{x}}(i)$ independent vectors in $\mathbb{R}^k$, and the norm of each vector is bounded by 1. Therefore, by (Pinelis, 1994, Theorem 3.5), for all $r_i \geq 0$

$$
\Pr[\|\boldsymbol{e}_i\| \geq r_i] \leq 2 \exp\left(-\frac{r_i^2}{2N q_{\boldsymbol{x}}(i)}\right)
$$

Given any $\delta > 0$ and $a$ with $\|a\| = 1$, we set $r_i = \sqrt{2N q_{\boldsymbol{x}}(i) \ln(2d/\delta)}$, and we have

$$
\|\boldsymbol{e}(a)\|^2 \leq \sum_i \|\boldsymbol{e}_i\|^2 \leq \sum_i 2N q_{\boldsymbol{x}}(i) \ln(\frac{2d}{\delta}) = 2N \ln \frac{2d}{\delta}
$$

Therefore,

$$
R_{\boldsymbol{e}} \leq \sqrt{2N \ln \frac{2d}{\delta}} \tag{22}
$$

with probability at least $1 - \delta$. Plugging in Eqs. (21) and (22) to Eq. (20), we have

$$
\|\bar{\boldsymbol{G}} - \hat{\boldsymbol{G}}\|_2 \leq \frac{1}{N^2} \left(2N \sqrt{2N \ln \frac{2d}{\delta}} + 2N \ln \frac{2d}{\delta}\right) \leq \frac{4\sqrt{\ln 2d/\delta}}{\sqrt{N}} + \frac{2 \ln 2d/\delta}{N}.
$$

$\square$

With Lemma E.3, we further provide sample complexity bounds for both multiplicative and additive errors of the plug-in Gram determinant score which are polynomial in all relevant parameters.

**Proposition E.4.** *There exists $\epsilon_0 > 0$, so that the following holds*

- *For any $d \in \mathbb{N}, \delta, \lambda > 0$, and $0 < \epsilon \leq \epsilon_0$, if $\hat{\boldsymbol{G}} \succ \lambda \mathbb{I}$, the plug-in Gram determinant score satisfies*

$$
\Pr[|\bar{S}(\hat{\boldsymbol{x}}, \boldsymbol{y}) - \Gamma| \leq \epsilon \Gamma] \geq 1 - \delta
$$

  *for all $N \geq \frac{100 d^2 \ln \frac{d}{\delta}}{\lambda^2 \epsilon^2}$ where $\Gamma = \det(\hat{\boldsymbol{G}})$.*

- *For any $d \in \mathbb{N}, \delta > 0, 0 < \epsilon \leq \epsilon_0$ and $\hat{\boldsymbol{G}}$, the plug-in Gram determinant score satisfies*

$$
\Pr[|\bar{S}(\hat{\boldsymbol{x}}, \boldsymbol{y}) - \Gamma| \leq \epsilon] \geq 1 - \delta
$$

  *for all $N \geq \frac{100 d^2 \ln \frac{d}{\delta}}{\epsilon^2}$.*

For the multiplicative error, we show that when $\hat{G}$ is invertible, the required sample size is $O(\frac{d^2 \ln \frac{d}{\delta}}{\lambda^2 \epsilon^2})$ where $\lambda$ can be the smallest eigenvalue of $\hat{G}$ as $\hat{G}$ is positive semidefinite. For the additive error, we can get a stronger bound $O(\frac{d^2 \ln \frac{d}{\delta}}{\epsilon^2})$ independent of the minimum eigenvalue.

This result implies Proposition 4.5, which shows that the reliability order is asymptotically preserved for any fixed pair of reports. However, it does not yield a uniform, non-asymptotic guarantee on order preservation, as two report Gram matrices can be arbitrarily close, $\hat{G} \approx \hat{G}'$.

*Proof.* First, Lemma E.3 implies that

$$\|\bar{G} - \hat{G}\|_2 \leq 4\sqrt{\frac{\log 2d/\delta}{N}} + 2\frac{\log 2d/\delta}{N}$$

with probability larger than $1 - \delta$. By taking $N_{\delta,\epsilon} = 100 \cdot \frac{d^2 \ln \frac{d}{\delta}}{\lambda^2 \epsilon^2}$ and $N \geq N_{\delta,\epsilon}$, we can bound the error as

$$4\sqrt{\frac{\log 2d/\delta}{N}} + 2\frac{\log 2d/\delta}{N} \leq 4\sqrt{\frac{\lambda^2 \epsilon^2 \ln 2}{100 d^2}} + 2\frac{\lambda^2 \epsilon^2 \ln 2}{100 d^2} \leq 5\sqrt{\frac{\lambda^2 \epsilon^2}{100 d^2}} = \frac{\lambda \epsilon}{2d}.$$

The first inequality is due to $N \geq N_{\delta,\epsilon}$. The second holds when $\epsilon_0$ is small enough and $\epsilon \leq \epsilon_0$ as $4\sqrt{\ln 2} < 5$. Note that the choice of $\epsilon_0$ can be independent of other parameters, because $\lambda \leq 1$ and $d \geq 1$.

Second, by Theorem A.2, we have

$$\frac{|\det(\bar{G}) - \det(\hat{G})|}{\det(\hat{G})} \leq \left(1 + \|\hat{G}^{-1}\|_2 \|\bar{G} - \hat{G}\|_2\right)^d - 1. \qquad \text{(by Theorem A.2)}$$

$$\leq \left(1 + \frac{1}{\lambda} \cdot \frac{\lambda \epsilon}{2d}\right)^d - 1. \qquad (\|\hat{G}^{-1}\|_2 \leq 1/\lambda)$$

$$\leq e^{\epsilon/2} - 1. \qquad ((1 + x/d)^d \leq e^x)$$

$$\leq \epsilon. \qquad \text{(by taking } \epsilon_0 \leq \ln 2)$$

This proves the first part.

For the second, part, we use Ipsen & Rehman (2008, Theorem 2.12),

$$|\det(\bar{G}) - \det(\hat{G})| \leq d\|\bar{G} - \hat{G}\|_2 \max\{\|\hat{G}\|_2, \|\bar{G}\|_2\}^{d-1}$$

$$\leq d\|\bar{G} - \hat{G}\|_2 \qquad \text{(by Perron–Frobenius theorem)}$$

$$\leq d \cdot \frac{\epsilon}{2d} \leq \epsilon \qquad \text{(by taking } N'_{\delta,\epsilon} = 100 \cdot \frac{d^2 \ln \frac{d}{\delta}}{\epsilon^2})$$

$\square$

**Proof of Proposition E.2** The core idea relies on the multi-linearity of the determinant, and we can approximately get samples of $\hat{G} = Q^\top G Q$ in the detail-free setting. However, one caveat is that we may not have access to multiple independent samples from $\hat{G}$ as $x, \hat{x}$ are deterministic. To circumvent this issue, we first observe that if $\hat{x} = x$, the observations are independently and identically distributed for each label, allowing an unbiased estimator for $\hat{G}$ and thus $\det(\hat{G})$. If $\hat{x} \neq x$, our sampling scheme ensures that the expectation is bounded above by the Gram determinant score. This guarantees that exact match orderings are preserved, as the truthful reports yield higher or scores in expectation compared to any nontruthful reports.

*Proof of Proposition E.2.* By the definition of exact ordering, it is sufficient to show for any $x, \hat{x}$ with $x \succ^x_{\text{EXACT}} \hat{x}$ and $P \in \mathcal{P}_{\text{indep}}$,

$$\mathbb{E}_{y \sim P(x)}[score(x, y)] > \mathbb{E}_{y \sim P(x)}[score(\hat{x}, y)].$$

When the minimum occurrence is at least two, the expectation of Eq. (16) involves three sources of randomness: observation $y$, permutations $\sigma$, and the choice of $Col$ and $Row$. The expectation of

$score(\boldsymbol{x}, \boldsymbol{y})$ only depends on the first two as difference indexing does not change the distribution of score. However, for $score(\hat{\boldsymbol{x}}, \boldsymbol{y})$, the third part will kick in.

Given the index sets $Col, Row$, we define $\boldsymbol{Q}^{Col}, \boldsymbol{Q}^{Row} \in \mathbb{R}^{d \times d}$ so that

$$\boldsymbol{Q}^{Col}(i,j) = q_{\hat{\boldsymbol{x}}}(j) \sum_{n \in Col} \mathbf{1}[x_n = i, \hat{x}_n = j] = q_{\hat{\boldsymbol{x}}}(j)\mathbf{1}[x_{j,Col} = i] \qquad (23)$$

and $\boldsymbol{Q}^{Row}(i,j)$ similarly which are the misreport matrix when restricting reports in $Col$ and $Row$ respectively. As $Col$ can be seen as stratified sampling where each report has exactly one element in $Col$, $\sum_{n \in Col} \mathbf{1}[x_n = i, \hat{x}_n = j] = \boldsymbol{Q}_{\boldsymbol{x}|\hat{\boldsymbol{x}}}(i,j)$, and the expectation over the choice of index is

$$\mathbb{E}[\boldsymbol{Q}^{Col}] = \mathbb{E}[\boldsymbol{Q}^{Row}] = \boldsymbol{Q}. \qquad (24)$$

With the above notation, we first compute the expectation of Eq. (16) conditional on $Col$ and $Row$.

$$\mathbb{E}[score(\hat{\boldsymbol{x}}, \boldsymbol{y}) \mid Col, Row]$$

$$= \mathbb{E}\left[ d!sgn(\sigma) \prod_{k,l \in [d], l=\sigma(k)} \mathbf{1}[y_{k,Row} = y_{l,Col}]q_{\hat{\boldsymbol{x}}}(k)q_{\hat{\boldsymbol{x}}}(l) \mid Col, Row \right]$$

$$= \mathbb{E}\left[ \sum_{\sigma \in sym(d)} sgn(\sigma) \prod_{k,l \in [d], l=\sigma(k)} \mathbf{1}[y_{k,Row} = y_{l,Col}]q_{\hat{\boldsymbol{x}}}(k)q_{\hat{\boldsymbol{x}}}(l) \mid Col, Row \right] \qquad (\text{random } \sigma)$$

$$= \mathbb{E}\left[ \sum_{\sigma \in sym(d)} sgn(\sigma) \prod_{k,l \in [d], l=\sigma(k)} \langle P_{x_{k,Row}}, P_{x_{l,Col}} \rangle q_{\hat{\boldsymbol{x}}}(k)q_{\hat{\boldsymbol{x}}}(l) \mid Col, Row \right] \qquad (Col \cap Row = \emptyset)$$

$$= \mathbb{E}\left[ \sum_{\sigma \in sym(d)} sgn(\sigma) \prod_{k,l \in [d], l=\sigma(k)} \sum_{i,j} \boldsymbol{Q}^{Row}(i,k)\boldsymbol{Q}^{Col}(j,l) \langle P_i, P_j \rangle \mid Col, Row \right] \qquad (\text{by Eq. (23)})$$

$$= \mathbb{E}\left[ \sum_{\sigma \in sym(d)} sgn(\sigma) \prod_{k,l \in [d], l=\sigma(k)} ((\boldsymbol{Q}^{Row})^{\intercal}\boldsymbol{G}\boldsymbol{Q}^{Col})(k,l) \mid Col, Row \right]$$

$$= \mathbb{E}\left[ \det\left((\boldsymbol{Q}^{Row})^{\intercal}\boldsymbol{G}\boldsymbol{Q}^{Col}\right) \mid Col, Row \right]$$

Therefore,

$$\mathbb{E}[score(\hat{\boldsymbol{x}}, \boldsymbol{y})] = \mathbb{E}\left[\det\left((\boldsymbol{Q}^{Row})^{\intercal}\boldsymbol{G}\boldsymbol{Q}^{Col}\right)\right] = \mathbb{E}\left[\det\left((\boldsymbol{Q}^{Row})^{\intercal}\boldsymbol{Q}^{Col}\right)\right]\det(\boldsymbol{G}) \qquad (25)$$

First, when $\hat{\boldsymbol{x}} = \boldsymbol{x}$, because $\boldsymbol{Q} \in \mathcal{Q}_L$, and $N \geq 2Ld$, every label has at least $N \min_i q_{\boldsymbol{x}}(i) \geq 2Ld\frac{1}{Ld-L+1} \geq 2$ reports by Lemma D.5, and the minimum occurrence is at least two. Moreover, $\boldsymbol{Q}^{Col} = \boldsymbol{Q}^{Row} = \boldsymbol{Q}$ are identity matrices regardless the choice of $Col$ and $Row$, by Eq. (25), we have

$$\mathbb{E}[score(\boldsymbol{x}, \boldsymbol{y})] = \det(\boldsymbol{G}). \qquad (26)$$

On the other hand, for $\hat{\boldsymbol{x}}$ with $\boldsymbol{x} \succ^{\boldsymbol{x}}_{\text{EXACT}} \hat{\boldsymbol{x}}$, if the minimum occurrence is less than two, the score would be zero and less than Eq. (29). Otherwise, by Cauchy–Schwarz inequality, we have

$$\mathbb{E}\left[\det\left((\boldsymbol{Q}^{Row})^{\intercal}\boldsymbol{Q}^{Col}\right)\right] \leq \mathbb{E}\left[\det\left((\boldsymbol{Q}^{Row})\right)\right]\mathbb{E}\left[\det\left(\boldsymbol{Q}^{Col}\right)\right] \qquad (27)$$

Formally, consider $\mathcal{I}$ the collection of all possible index set of size $d$ where each label occurs exactly once. Then we can generate $Col$ and $Row$ by sampling two distinct $(i,j)$ element of $\mathcal{I}$ uniformly at random. In particular, if we set $a_i$ be the determinant of the misreporting matrix of the $i$-th index

set in $\mathcal{I}$, the joint distribution of $(\det(\boldsymbol{Q}^{Col}), \det(\boldsymbol{Q}^{Row}))$ equals $(a_i, a_j)$ and

$$\mathbb{E}\left[\det\left((\boldsymbol{Q}^{Row})\right)\right] \mathbb{E}\left[\det\left(\boldsymbol{Q}^{Col}\right)\right] - \mathbb{E}\left[\det\left((\boldsymbol{Q}^{Row})^\mathsf{T}\boldsymbol{Q}^{Col}\right)\right]$$

$$= \left(\frac{1}{|\mathcal{I}|}\sum_i a_i\right)\left(\frac{1}{|\mathcal{I}|}\sum_j a_j\right) - \frac{1}{|\mathcal{I}|(|\mathcal{I}|-1)}\sum_{i\neq j \in \mathcal{I}} a_i a_j$$

$$= \frac{1}{|\mathcal{I}|^2}\sum_i a_i^2 - \frac{1}{|\mathcal{I}|^2(|\mathcal{I}|-1)}\sum_{i\neq j} a_i a_j$$

$$= \frac{1}{2|\mathcal{I}|^2(|\mathcal{I}|-1)}\sum_{i\neq j}(a_i - a_j)^2 \geq 0$$

which proves Eq. (27). Finally, using the first part of Theorem 4.2 and Eqs. (24) to (27)

$$\mathbb{E}[score(\hat{\boldsymbol{x}}, \boldsymbol{y})] \leq \det(\boldsymbol{Q}^\mathsf{T}\boldsymbol{G}\boldsymbol{Q}) = \det(\hat{\boldsymbol{G}}) < \det(\boldsymbol{G}) = \mathbb{E}[score(\boldsymbol{x}, \boldsymbol{y})].$$

$\square$

# F   DETAILS AND PROOFS FOR SECTION 4.3

Now we provide examples to motivate kernelized Gram determinant scores in Definition 4.6.

1. Given any feature map $\phi : \mathcal{Y} \to \mathbb{R}^k$ that maps observations to Euclidean space, we define $K(y, y') = \langle \phi(y), \phi(y') \rangle$ as the standard inner product between the features. A feature map is *injective* if the vectors $\{\phi(y)\}_{y\in\mathcal{Y}}$ are linearly independent. For instance, using the one-hot encoder $\phi : y \mapsto \delta_y$ results in delta-kernel $K(y, y') = \mathbf{1}[y = y']$ and reproduces Definition 4.1.

2. More generally, we can consider implicit feature maps, e.g., Gaussian radial basis function where $K(y, y') = \exp\left(\frac{-\|y-y'\|_2^2}{\sigma^2}\right)$ for $\mathcal{Y} \subseteq \mathbb{R}^k$, or general Hilbert space. (Ziegel et al., 2022)

3. We can use feature maps to incorporate special structure in $\mathcal{Y}$, e.g., predictions of true labels. Formally, given $\boldsymbol{P}$, we say an observation $y$ is a pseudo-posterior with prior $\tilde{q} \in \Delta(\mathcal{X})$ if $y = \{\tilde{\Pr}[\mathrm{x} = x|y]\}_{x\in\mathcal{X}} = \{\frac{P(y,x)\tilde{q}(x)}{\sum_{x'} P(y,x')\tilde{q}(x')}\}_{x\in\mathcal{X}}$ is the posterior of true label under prior $\tilde{q}$. (Kass & Wasserman, 1996) Rather than one-hot encoder, we may consider $\phi(y) = y \in \mathbb{R}^d$ which has smaller and meaningful feature space. We call the associated kernel $K(y, y') = y^\mathsf{T}y'$ with pseudo posterior experiment as *pseudo-posterior kernel*.

We show that kernelized Gram determinant reliability scores also preserve all reliability orderings in Theorem 4.2 for general observation space $\mathcal{Y}$ under three kernel families. First, the result holds for any integrally strictly positive-definite kernel, so admitting arbitrary (possibly infinite or continuous) observation spaces. When $\mathcal{Y}$ is finite, one may use any kernel with injective feature maps. The guarantee also holds when the observations are pseudo-posteriors with arbitrary prior $\tilde{q}$ with full support.

**Theorem F.1.** *Given $\mathcal{X} = [d]$, $\mathcal{Y}$ and $L \geq 1$, the Gram determinant score with any of the following kernels in Definition 4.6 preserves reliability orderings in Theorem 4.2:*

1. *Integrally strictly positive definite kernels—in particular the Gaussian (RBF) kernel on any separable Hilbert space $\mathcal{Y}$.*

2. *Kernels with an injective feature map $\phi : \mathcal{Y} \to \mathbb{R}^k$ and finite set $\mathcal{Y}$.*

3. *Pseudo posterior kernel $K$ with full support $\tilde{q}$*

The proof is mostly identical to that of Theorem 4.2. As the kernel only changes the Gram matrix of labels $\boldsymbol{G}$, it is sufficient to show $\boldsymbol{G}$ is positive definite to reuse Lemmas D.1 and D.3. Similarly, those two estimators in Section 4.2 can also adopt kernels. We provide details in Appendix F.

*Proof of Theorem F.1.* To use Lemmas D.1 to D.3, it is sufficient to show that $\boldsymbol{G}_K$ is positive definite for all $\boldsymbol{P} \in \mathcal{P}_{\text{indep}}$ so that for any nonzero sequence $a : \mathcal{X} \to \mathbb{R}$, the quadratic form is positive,

$$\sum_{x,x' \in \mathcal{X}} a(x)G(x,x')a(x') > 0. \tag{28}$$

First for any integrally strictly positive definite kernel, recall that the kernel mean embedding of $P_x$ is $\phi(P_x) = \mathbb{E}_{y \sim P_x}[\phi(y)] \in \mathcal{H}$, so $\sum_{x,x'} a(x)G_K(x,x')a(x') = \| \sum_x a(x)\phi(P_x)\|^2 \geq 0$ which shows $\boldsymbol{G}_K$ is positive semi definite. If the equality happens, by linearity of integration, $0 = \| \sum_x a(x)\phi(P_x)\|^2 = \iint_{\mathcal{Y}} K(y,y')d\mu(y)d\mu(y')$ where $\mu = \sum_x a(x)P_x$ is a finite signed measure. Therefore, $\mu = \sum_x a(x)P_x = 0$ because $K$ is integrally strictly positive definite. Finally $a(x) = 0$ as columns of $\boldsymbol{P}$ are linearly independent. Therefore the statement holds for integrally strictly positive definite kernels. Additionally, by (Ziegel et al., 2022, Theorem 3.1), the Gaussian kernel is integrally strictly positive definite.

Second, given a feature map $\phi : \mathcal{Y} \to \mathbb{R}^k$, Eq. (28) becomes $\| \sum_{x,y} a(x)P(y,x)\phi(y)\|_2^2$. Because $\boldsymbol{P} \in \mathcal{P}_{\text{indep}}$ and $\phi$ is injective, the quadratic form equals zero if and only if $a(x) = 0$ for all $x$. Moreover, delta kernel is injective, so the statement also holds.

Finally, for any pseudo-posterior observations, Eq. (28) can be written as

$$\left\langle \sum_{x,y} a(x)\boldsymbol{P}(y,x)y, \sum_{x',y'} a(x')\boldsymbol{P}(y',x')y' \right\rangle$$

$$= \sum_{x,x',y,y'} a(x)a(x')\boldsymbol{P}(y,x)\boldsymbol{P}(y',x')\langle \tilde{P}[\mathrm{x}|y], \tilde{P}[\mathrm{x}|y'] \rangle$$

$$= \sum_{x,x',y,y'} a(x)a(x')\boldsymbol{P}(y,x)\boldsymbol{P}(y',x') \sum_{x''} \tilde{P}[\mathrm{x} = x''|y]\tilde{P}[\mathrm{x} = x''|y']$$

Let $b(y) = \sum_x a(x)\boldsymbol{P}(y,x)$, $w(y) = \sum_x \boldsymbol{P}(y,x)\tilde{q}(x)$. Then $\sum_x \tilde{P}[\mathrm{x} = x|y]\tilde{P}[\mathrm{x} = x|y'] = \sum_x \boldsymbol{P}(y,x)\frac{\tilde{q}(x)}{w(y)}\boldsymbol{P}(y',x)\frac{\tilde{q}(x)}{w(y')}$[7] and

$$\sum_{x,x',y,y'} a(x)a(x')\boldsymbol{P}(y,x)\boldsymbol{P}(y',x') \sum_{x''} \tilde{P}[\mathrm{x} = x''|y]\tilde{P}[\mathrm{x} = x''|y']$$

$$= \sum_{y,y'} b(y)b(y') \sum_x \boldsymbol{P}(y,x)\frac{\tilde{q}(x)}{w(y)}\boldsymbol{P}(y',x)\frac{\tilde{q}(x)}{w(y')}$$

$$= \sum_x \tilde{q}(x)^2 \left( \sum_y \boldsymbol{P}(y,x)\frac{b(y)}{w(y)} \right)^2$$

Because $\tilde{q}$ has full support, the quadratic form equals zeros if and only if $\sum_y \boldsymbol{P}(y,x)\frac{b(y)}{w(y)} = 0$ for all $x$. Equivalently, if we set vector $b = \boldsymbol{P}a \in \mathbb{R}^{|\mathcal{Y}|}$ and $\boldsymbol{D}_w$ be the diagonal matrix with $w$, we have $\boldsymbol{0} = b^{\mathsf{T}}\boldsymbol{D}_w\boldsymbol{P} = a^{\mathsf{T}}\boldsymbol{P}^{\mathsf{T}}\boldsymbol{D}_w\boldsymbol{P}$. Since $\boldsymbol{P}$ has full column rank and $w(y) = 0$ when $\boldsymbol{P}(x,y) = 0$ for all $x$, $a(x) = 0$ for all $x$. $\qquad\square$

**Definition F.2** (plug-in Kernelized Gram determinant reliability score). *Given a kernel $K : \mathcal{Y}^2 \to \mathbb{R}$, $\hat{\boldsymbol{x}}, \boldsymbol{y}$ of length $N$, let $\bar{\boldsymbol{G}}_K : \mathcal{X} \times \mathcal{X} \to \mathbb{R}$ where*

$$\bar{\boldsymbol{G}}_K(x,x') = \frac{1}{N^2} \sum_{n,n' \in [N]:\hat{x}_n=x,\hat{x}_{n'}=x'} K(y_n, y_{n'}).$$

*The plug-in kernelized Gram determinant reliability score is $\bar{S}_K(\hat{\boldsymbol{x}}, \boldsymbol{y}) = \det(\bar{\boldsymbol{G}}_K)$*

**Theorem F.3.** *Given $\mathcal{X} = [d]$, finite set $\mathcal{Y}$ and $L \geq 1$, the plug-in Gram determinant score with any bounded kernels in Theorem 4.2 asymptotically preserves reliability orderings in Theorem 4.2.*

---

[7]Here we set $0/0 = 0$ if $w(y) = 0$

**Lemma F.4.** *Given $|K| \leq 1$, $\delta > 0$ and report length $N$,*

$$\Pr\left[\|\bar{\boldsymbol{G}}_K - \hat{\boldsymbol{G}}_K\|_2 \leq 4\sqrt{\frac{\log 2d/\delta}{N}} + 2\frac{\log 2d/\delta}{N}\right] \geq 1 - \delta$$

The above lemma shows that the empirical estimator $\bar{\boldsymbol{G}}$ is close to its expectation $\hat{\boldsymbol{G}}$ in spectral norm. The proof is mostly identical to Lemma E.3. We only need concentration results of the sum of independent random elements in Hilbert spaces as (Pinelis, 1994, Theorem 3.5).

Finally, we can also design an estimator that preserves exact match ordering even with finite length data.

**Definition F.5.** *Given a kernel $K : \mathcal{Y} \times \mathcal{Y} \to \mathbb{R}$, $\hat{\boldsymbol{x}}, \boldsymbol{y}$ of length $N$, a* stratified-matching estimator *estimates the kernelized Gram determinant as the following*

1. *Return $0$ if the minimum occurrence $\min_{x \in \mathcal{X}} |\{n \in [N] : \hat{x}_n = x\}|$ is less than $2$. Otherwise, we randomly select two disjoint index sets $Col, Row \subseteq [N]$ of size $d$ where each label $i \in \mathcal{X}$ occurs in each set exactly once. Then re-index them as two sequences of pairs $(\hat{x}_{i,Col}, y_{i,Col})_{i \in [d]}$ and $(\hat{x}_{i,Row}, y_{i,Row})_{i \in [d]}$ so that $\hat{x}_{i,Col} = \hat{x}_{i,Row} = i$ for all $i \in \mathcal{X}$. We call the first as column sequence and the second as row sequence.*

2. *Randomly sample one permutations $\sigma \in sym(d)$, and return*

$$score(\hat{\boldsymbol{x}}, \boldsymbol{y}) := d! sgn(\sigma) \prod_{i,j \in [d], j = \sigma(i)} K(y_{i,Row}, y_{j,Col}) q_{\hat{\boldsymbol{x}}}(i) q_{\hat{\boldsymbol{x}}}(j). \tag{29}$$

**Theorem F.6.** *Given $\mathcal{X} = [d]$ and $L \geq 1$, the stratified-matching estimator in Definition E.1 with any of kernels in Theorem 4.2 preserves exact ordering on $\mathcal{P}_{indep}$, $\mathcal{Q}_L$, and $N \geq 2Ld$.*

The proof is mostly identical to Proposition E.2

**Remark F.7.** *Our Gram determinant score can be viewed as an application of the peer prediction mechanism introduced in (Kong, 2024), where one agent's report is replaced with the observation $\boldsymbol{y}$. In addition to offering a more fine-grained characterization of the Gram determinant score, as discussed in related work, we also introduce several technical improvements over the original determinant mutual information method. First, the prior approach requires $\mathcal{Y} = \mathcal{X}$ and overlooks potential structure in the observations. As shown in Section 4.3, we address this by introducing kernel methods, allowing us to generalize the score to arbitrary observation spaces $\mathcal{Y}$—a crucial extension for handling continuous observations such as Gaussian variables or image embeddings, as demonstrated in Section 5. Second, our stratified-matching estimators in Definitions E.1 and F.5 are unbiased in the multi-task peer prediction setting of (Kong, 2024), and they reduce the estimator's range from order $(d!)^2$ to $d!$.*

# G ALTERNATIVES TO GRAM DETERMINANT SCORE

## G.1 MORE DATA RELIABILITY SCORES

There is a long line of research on measuring the stochastic relationship between random variables. We may view them as data reliability scores applied to the reported data $\hat{\boldsymbol{x}}$ and observations $\boldsymbol{y}$. In this section, we list some common candidates and illustrate the limitations and possibilities.

**$\Phi$-mutual information**

**Definition G.1** ($\Phi$-divergence (Csiszár, 1964; Morimoto, 1963; Ali & Silvey, 1966)). *Let $\Phi : [0, \infty) \to \mathbb{R}$ be a convex function with $\Phi(1) = 0$. Let $P$ and $Q$ be two probability distributions on a common measurable space $(\Omega, \mathcal{F})$. The $\Phi$-divergence of $Q$ from $P$ where $P \ll Q$[8] is defined as $D_\Phi(P\|Q) := \mathbb{E}_Q[\Phi(P/Q)]$.[9]*

---

[8] $P$ is absolutely continuous with respect to $Q$: for any measurable set $A \in \mathcal{F}$, $Q(A) = 0 \Rightarrow P(A) = 0$.

[9] $P/Q$ is the Radon-Nikodym derivative between measures $P$ and $Q$, and it is equal to the ratio of density function.

We can use these divergences to measure how interdependent two random variables x and y are. Formally, Let $P_{x,y}$ be a distribution over $(x, y) \in \mathcal{X} \times \mathcal{Y}$, and $P_x$ and $P_y$ be marginal distributions of x and y respectively. We set $P_x P_y$ be the tensor product between $P_x$ and $P_y$ such that $P_x P_y(x, y) = P_x(x)P_y(y)$. We call $D_\Phi(P_{x,y} \| P_x P_y)$ the $\Phi$-*mutual information between* x *and* y.

1. Total variation has $\Phi(a)$ as $\frac{1}{2}|a - 1|$.

2. KL-divergence has $a \log a$

3. $\chi^2$-divergence has $a^2 - 1$

4. Squared Hellinger distance has $(1 - \sqrt{a})^2$

In the partial knowledge setting, we can access the $\boldsymbol{J} := \boldsymbol{PQ}$ which can be seen as a joint distribution between reported data and observation $\boldsymbol{J} = P_{x,y}$, and set

$$S_\Phi(\boldsymbol{PQ}) = D_\Phi(P_{x,y} \| P_x P_y).$$

This family of score satisfy the data processing inequality, which is analogous to our *weak* Blackwell dominant ordering so that garbling the report can only decrease the score. Nevertheless, the impossibility results in Section 3 still apply. In addition, they are generally not experiment-agnostic, and lack kernelized extensions as in Section 4.3 for complicated observation space $\mathcal{Y}$.

**Family of symmetric gauge on singular values** Our Gram determinant is essentially a functional on the singular values of $\boldsymbol{J} = \boldsymbol{PQ}$ and sub multiplicative under right multiplication by contraction. One may additionally consider functional on the singular values of the whitened matrix. Formally, given a joint distribution $\boldsymbol{J} := \boldsymbol{PQ}$, let

$$\bar{\boldsymbol{J}} = \boldsymbol{D_y}^{-1/2}(\boldsymbol{J} - \mu_y \mu_{\hat{x}}^\intercal)\boldsymbol{D_{\hat{x}}}^{-1/2}$$

where $\mu_{\hat{x}}$ and $\mu_y$ are marginal distributions and $\boldsymbol{D_{\hat{x}}}, \boldsymbol{D_y}$ are diagonal matrix of them respectively. Given a matrix $\boldsymbol{A}$, $\sigma(\boldsymbol{A})$ denote the singular value list of $\boldsymbol{A}$, we can find a symmetric gauge $\psi$ and define our score as

$$S_\psi(\boldsymbol{J}) = \psi(\sigma(\bar{\boldsymbol{J}})).$$

Let $\bar{\boldsymbol{s}} = \sigma(\bar{\boldsymbol{J}}) = (\bar{s}_1, \dots, \bar{s}_d)$ with $\bar{s}_1 \geq \bar{s}_2 \geq \dots \bar{s}_d \geq 0$.

1. Top-$k$ volume has $\psi_{\wedge k}(\boldsymbol{s}) = \prod_{i=1}^k \bar{s}_i$

2. Maximal correlation $\psi_{\max} = \bar{s}_1$. The maximum correlation can be also written as

$$\max_{(f,g) \in \mathcal{S}} \mathbb{E}[f(x)g(y)]$$

where $\mathcal{S}$ is the collection of real-valued random variables so that $\mathbb{E}f(x) = \mathbb{E}g(y) = 0$ and $\mathbb{E}f(x)^2 = \mathbb{E}g(y)^2 = 1$.

3. Ky-Fan $k$-sum $\sum_{i=1}^k \bar{s}_i$

4. $\chi^2$-mutual information $I_{\chi^2}(x, y) = \sum_{x,y} \mu_{\hat{x}}(x)\mu_y(y)(\frac{\boldsymbol{J}(y,x)}{\mu_{\hat{x}}(x)\mu_y(y)} - 1)^2 = \|\bar{\boldsymbol{J}}\|_F = \sum_i \bar{s}_i^2$

Similarly, the impossibility results in Section 3 still apply and they are generally not experiment-agnostic.

## G.2 EXPERIMENTS ON SCORE COMPARISON

We follow the same data generation process and manipulation policies as in Experiment 1 (Fig. 2), and focus here on comparing four possible reliability scores (Top-$k$ volume with $k = 4$, maximal correlation, KL divergence, and $\chi^2$-mutual information) computed from the empirical joint distribution $\boldsymbol{J} = \boldsymbol{PQ}$.

Across manipulations the larger values of $p$ indicate less corruption, and in practice they are inversely related to the corruption level as measured by $1 - p$, the Hamming distance, and the $\ell_2$ norm between $\boldsymbol{x}$ and $\hat{\boldsymbol{x}}$ (see Figs. 2 and 4). This alignment across multiple metrics demonstrates that the proposed scores all provide robust and informative signals of data quality. Under the Merge 0/1$\rightarrow$

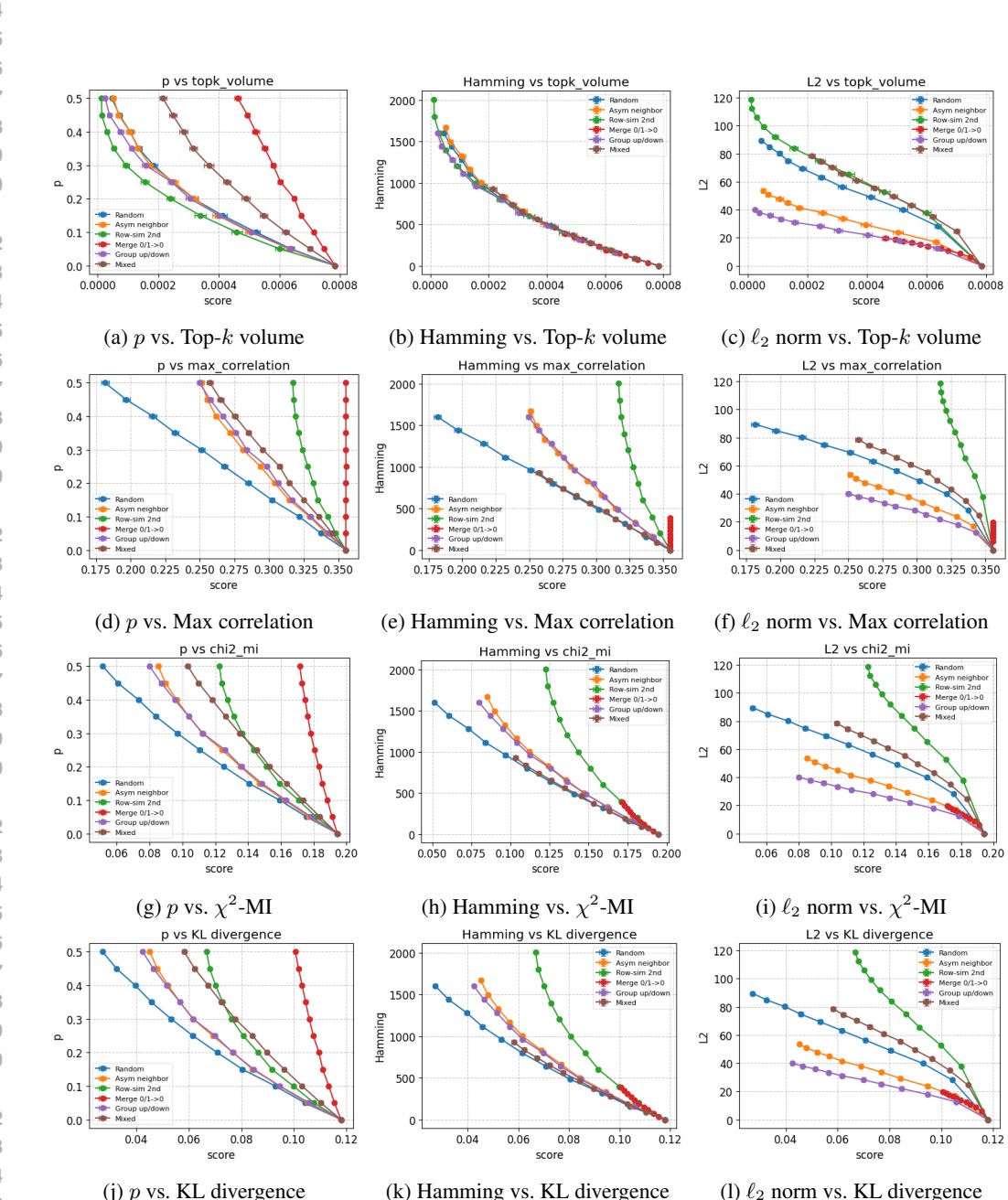

Figure 4: Comparison of the Top-$k$ volume, Max correlation, KL divergence, and $\chi^2$-mutual information scores under different corruption levels and metrics.

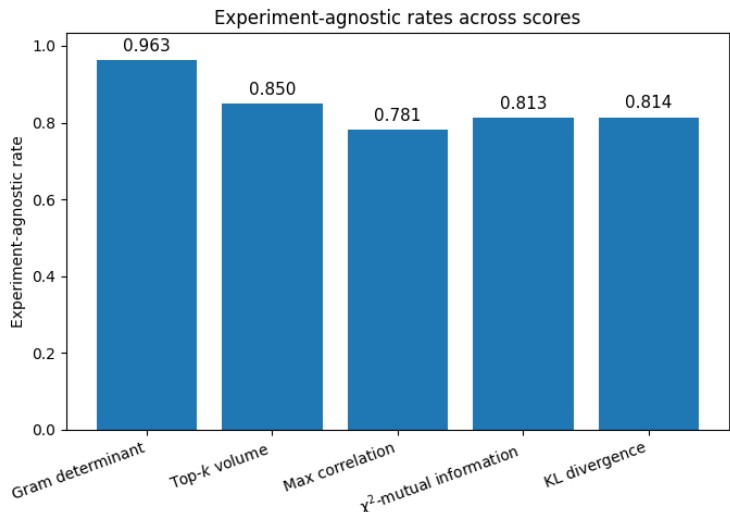

Figure 5: Experiment-agnostic rates across different scoring functions.

0 manipulation, maximal correlation performs poorly, assigning almost identical score to datasets with varying level of manipulation. This is likely due to that maximal correlation depends only on the most significant singular value and misses more fine-grained information. By contrast, the Gram determinant (product of all $d = 5$ singular values) and the top-$k$ volume (product of the largest $k = 4$) perform better, as shown in Fig. 2a-c and Fig. 4a-c respectively. Additionally, we observe cross-manipulation inconsistencies in maximal correlation, $\chi^2$-MI and KL-divergence in Fig. 4e,h,k: when two datasets are manipulated by different methods, they may assign a higher score to the dataset that is further away from the truth according to Hamming distance, a violation to Hamming distance ordering. In contrast, our GDS score (Fig. 2b) and the top-$k$ volume score (Fig. 4b) consistently preserve the Hamming distance ordering across all six manipulations.

Then we evaluate the robustness of each scoring function to changes in the underlying representation. Our goal is to assess whether a score preserves the relative quality ordering between two corruption channels even after the data undergoes an additional, task-specific transformation.

For each trial, we first sample a marginal distribution $\pi_X$ over $\{1, \ldots, d\}$ with $d = 4$, together with a ground-truth conditional model $\boldsymbol{P}_{Y|X} \in \mathbb{R}^{d \times d}$ whose rows are normalized to be stochastic. We then draw two independent corruption channels $\boldsymbol{Q}_1, \boldsymbol{Q}_2 \in \mathbb{R}^{d \times d}$, each representing a distinct noisy mapping from $x$ to corrupted reports. All matrices are row-stochastic, so they define valid conditional distributions.

We generate $1 \times 10^8$ i.i.d. samples by drawing $x \sim \pi_X$, sampling $y \sim \boldsymbol{P}_{Y|X}(\cdot \mid x)$, and obtaining two corrupted versions $\hat{x} \sim \boldsymbol{Q}_1(\cdot \mid x)$ and $\hat{x}' \sim \boldsymbol{Q}_2(\cdot \mid x)$. Collecting these over all draws yields the datasets $\boldsymbol{x}, \boldsymbol{y}, \hat{\boldsymbol{x}}, \hat{\boldsymbol{x}}'$. For each scoring function $S(\cdot)$ (Gram determinant, top-k volume, max correlation, $\chi^2$-mutual information, and KL divergence), we compute

$$S(\hat{\boldsymbol{x}}, \boldsymbol{y}), \quad S(\hat{\boldsymbol{x}}', \boldsymbol{y}), \quad S(\hat{\boldsymbol{x}}, \boldsymbol{x}), \quad S(\hat{\boldsymbol{x}}', \boldsymbol{x}),$$

corresponding to evaluating channel quality before and after the transformation $x \to y$.

A scoring function is said to preserve the ordering between the two channels if

$$\left( S(\hat{\boldsymbol{x}}, \boldsymbol{x}) - S(\hat{\boldsymbol{x}}', \boldsymbol{x}) \right) \left( S(\hat{\boldsymbol{x}}, \boldsymbol{y}) - S(\hat{\boldsymbol{x}}', \boldsymbol{y}) \right) > 0,$$

that is, if the sign of the score difference is invariant under the data-processing transformation. A violation corresponds to a *ranking flip*. The experiment-agnostic rate is defined as $1 -$ flip rate, representing how often the ordering is preserved.

We repeat this procedure for 1000 independent trials and report the average experiment-agnostic rate for each scoring function in Figure 5. Higher rates indicate greater robustness to changes in representation, i.e., a stronger ability to maintain consistent channel rankings across different tasks and observation models.

As shown in Figure 5, the Gram determinant score achieves a substantially higher experiment-agnostic rate than the alternative scoring functions. This empirical advantage is consistent with our theoretical guarantee in Proposition 4.3, which shows that the Gram determinant is uniquely robust to changes in the underlying representation and preserves channel orderings under a broad class of transformations. These results confirm that the Gram determinant score not only performs well in specific synthetic settings, but also provides the most stable and robust measure of reliability across heterogeneous tasks and observation models.

## H ADDITIONAL EXPERIMENTS AND DISCUSSION

### H.1 EXPERIMENT DETAILS

Due to space limitations, we omit some settings in the main paper. Here, we provide the details of how we compute error bars and how we obtain the ranking-accuracy across sample sizes in Fig. 2d.

**Error bars.** Let $M$ be the number of independent trials. For each trial $m \in [M]$, let $score^{(m)}$ denote the determinant score, and similarly let $\mathrm{Hamming}^{(m)}$ and $\ell_2^{(m)}$ denote the Hamming distance and $\ell_2$-norm error, respectively. We compute the sample mean

$$\overline{score} = \frac{1}{M} \sum_{m=1}^{M} score^{(m)}$$

and the standard error of the mean

$$SE(\overline{score}) = \frac{1}{\sqrt{M(M-1)}} \sqrt{\sum_{m=1}^{M} \left(score^{(m)} - \overline{score}\right)^2}.$$

Under approximate normality, we report a 95% confidence interval as $\overline{score} \pm 1.96 SE(\overline{score})$ in Fig. 2a, Fig. 2b and Fig. 2c. The same procedure is applied to $\mathrm{Hamming}^{(m)}$ and $\ell_2^{(m)}$ to yield their error bars in Figs. 2b and 2c.

**Ranking accuracy across sample sizes.** In Fig. 2d, we plot the fraction of trials in which the reversed ranking induced by the determinant score agrees with the ranking induced by each baseline metric—namely the reporting probability $p$, the Hamming distance, and the $\ell_2$-norm—over six noise levels $\mathcal{P} = \{0.0, 0.1, 0.2, 0.3, 0.4, 0.5\}$. Concretely, in each trial $m$ we form three vectors

$$\left(score_p^{(m)}\right)_{p\in\mathcal{P}}, \quad \left(\mathrm{Hamming}_p^{(m)}\right)_{p\in\mathcal{P}}, \quad \left(\ell_{2p}^{(m)}\right)_{p\in\mathcal{P}}.$$

We then check whether the total order of $\left(score_p^{(m)}\right)$ in decreasing order matches the order of $\left(\mathrm{Hamming}_p^{(m)}\right)$ in increasing order (and similarly for $\ell_2$ and for $p$ itself). If they coincide, trial $m$ is counted as a "correct" ranking. The plotted accuracy is

$$\frac{1}{M} \sum_{m=1}^{M} \mathbf{1}\big\{\text{orders agree in trial } m\big\}.$$

A random guess among the 6! possible orderings yields a baseline accuracy of $1/6! \approx 0.00139$.

### H.2 ADDITIONAL EXPERIMENTS

#### H.2.1 COMPARISON ON DELTA AND GAUSSIAN KERNEL

In this section, we still use the dataset and experiment settings from Experiment 1. However, besides the uniform random manipulation introduced in Eq. (6), we consider *normal manipulation*:

$$\hat{\boldsymbol{x}}_k \sim \mathrm{clip}\big(1, d, \mathrm{round}\big(\mathcal{N}(\boldsymbol{x}, \sigma_0)\big)\big). \tag{30}$$

This manipulation introduces localized perturbations to the data. We adopt $\sigma_0 \in \{0.30, 0.37, 0.44, \ldots, 1.00\}$ in our experiments and refer to this type of manipulation as normal

| Figure | Experiments | Manipulation | Kernel |
|--------|-------------|--------------|--------|
| Figure 2 | Random exp | uniform Eq. (6) | delta |
| Figure 7 | Random exp | normal Eq. (30) | delta |
| Figure 8 | Random exp | uniform | Gaussian |
| Figure 9 | Random exp | normal | Gaussian |
| Figure 10 | Gaussian | uniform | Gaussian |
| Figure 11 | Gaussian | normal | Gaussian |
| Figure 12 | Gaussian | uniform | delta |
| Figure 13 | Gaussian | normal | delta |

Table 1: Table for settings of the experiments on synthetic data.

manipulation. For the matched rankings results, the rankings are computed for 6 data points with $\sigma_0 \in \{0.30, 0.44, 0.58, 0.72, 0.86, 1.00\}$.

Moreover, we also use the *Gaussian kernel*

$$K(y, y') = \exp\left(-100 \|y - y'\|_2^2\right)$$

besides the delta kernel $K(y, y') = \mathbf{1}[y = y']$.

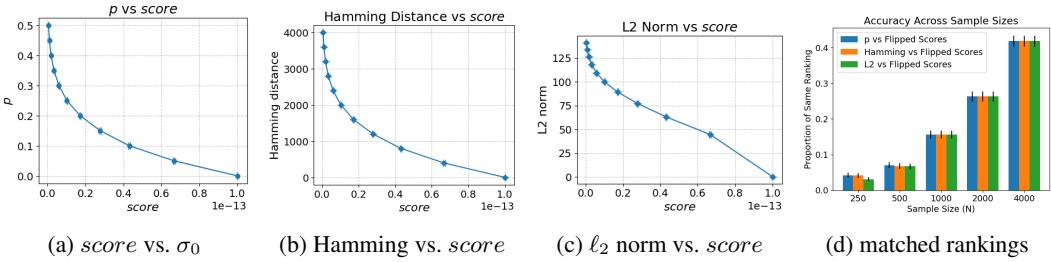

(a) *score* vs. $\sigma_0$     (b) Hamming vs. *score*     (c) $\ell_2$ norm vs. *score*     (d) matched rankings

Figure 6: Experiments of plug-in Gram determinant reliability score with delta kernel on categorical synthetic data with uniform manipulation in Eq. (6).

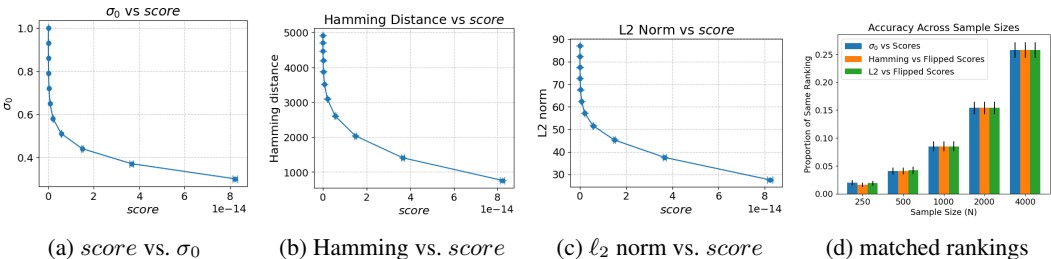

(a) *score* vs. $\sigma_0$     (b) Hamming vs. *score*     (c) $\ell_2$ norm vs. *score*     (d) matched rankings

Figure 7: Experiments of plug-in Gram determinant reliability score with delta kernel on categorical synthetic data with normal manipulation in Eq. (30).

From Figs. 7 to 9, we can conclude that regardless of the kernel used for the Gram determinant score, it consistently behaves well as a measure of reliability. It exhibits a negative correlation with all reliability metrics. As the sample size increases, the ability of the Gram determinant score to align with other reliability metrics also improves. For this categorical dataset, the delta kernel outperforms the Gaussian kernel, achieving both a smaller standard mean error and higher accuracy for matched rankings across sample sizes.

Then we create a new synthetic dataset. Instead of random experiments, each $y_i$ is sampled from a normal distribution $\mathcal{N}(x_i, \sigma)$ centered at $x_i$. We adopt $\sigma = 0.1$ and $d = 4$ in this experiment, and all other experimental settings remain the same as in Experiment 1.

Since $\mathcal{Y}$ lies in the continuous space $\mathbb{R}$ rather than being categorical, we cannot directly apply the plug-in Gram determinant score with the delta kernel. Hence, we use an approximate scor-

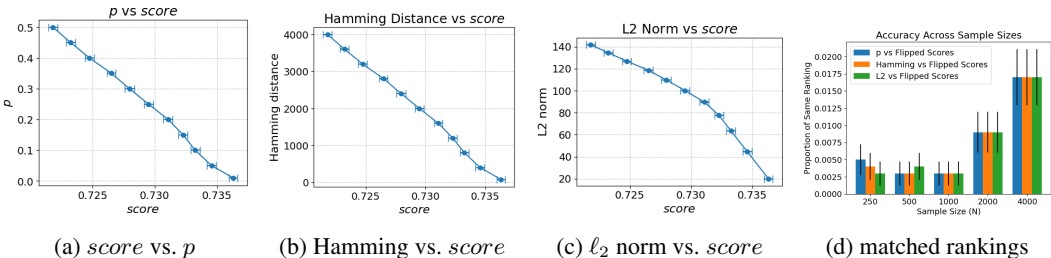

(a) $score$ vs. $p$     (b) Hamming vs. $score$     (c) $\ell_2$ norm vs. $score$     (d) matched rankings

Figure 8: Experiments of plug-in Gram determinant reliability score with Gaussian kernel on categorical synthetic data with uniformly random manipulation in Eq. (6).

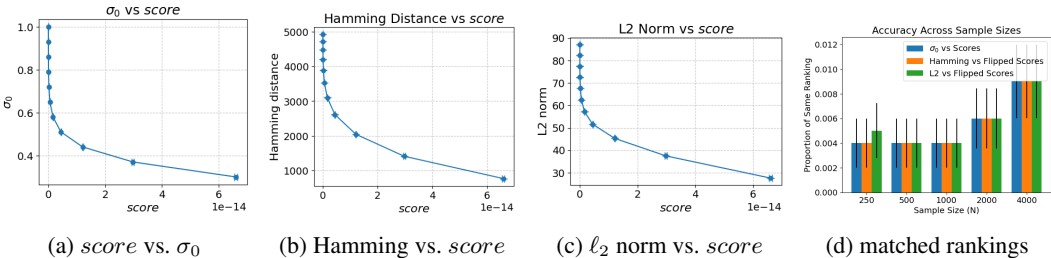

(a) $score$ vs. $\sigma_0$     (b) Hamming vs. $score$     (c) $\ell_2$ norm vs. $score$     (d) matched rankings

Figure 9: Experiments of plug-in Gram determinant reliability score with Gaussian kernel on categorical synthetic data with normal manipulation in Eq. (30).

ing method: we create a new sequence $\bar{\mathbf{y}}$ by bucketing $\mathbf{y}$ into $d$ bins using empirical quantiles, and then apply the plug-in Gram determinant score on $\hat{\boldsymbol{x}}$ and $\bar{\mathbf{y}}$.

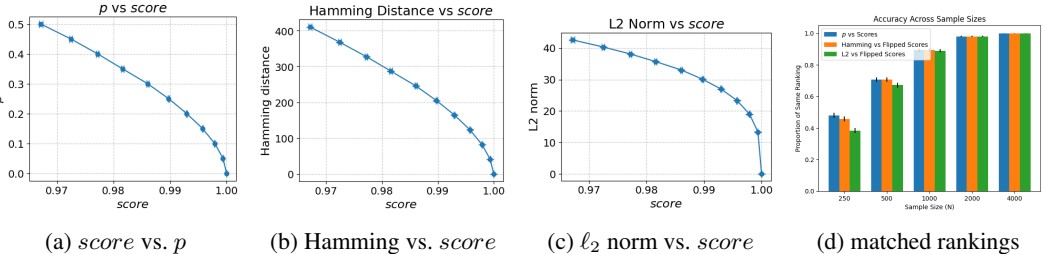

(a) $score$ vs. $p$     (b) Hamming vs. $score$     (c) $\ell_2$ norm vs. $score$     (d) matched rankings

Figure 10: Experiments of plug-in Gram determinant reliability score with Gaussian kernel on Gaussian synthetic data with uniformly random manipulation in Eq. (6).

From Figs. 10 to 13, we observe that both the delta kernel and the Gaussian kernel perform well as reliability measures across all three metrics, under both normal and uniformly random manipulations. In particular, the delta kernel variant using bucketed $\boldsymbol{y}$ achieves consistently strong performance. Empirically, the plug-in Gram determinant score with the delta kernel generally outperforms the version with the Gaussian kernel in most situations, despite the lack of theoretical guarantees for this delta kernel variant. For reported data with small $\ell_2$ norm error, the Gaussian kernel outperforms the approximate delta kernel score.

### H.2.2 ROBUSTNESS UNDER CONDITIONAL LINEAR DEPENDENCE

In this experiment, we investigate how violations of conditional linear independence affect the robustness of our proposed reliability scores. To introduce controlled linear dependence, we construct conditional distributions by generating a stochastic matrix $\boldsymbol{P}$ of a prescribed rank k: we sample k independent basis rows on the simplex and express the remaining rows as random convex combinations of these bases, ensuring that $\mathrm{rank}(\boldsymbol{P}) = \mathrm{k}$.

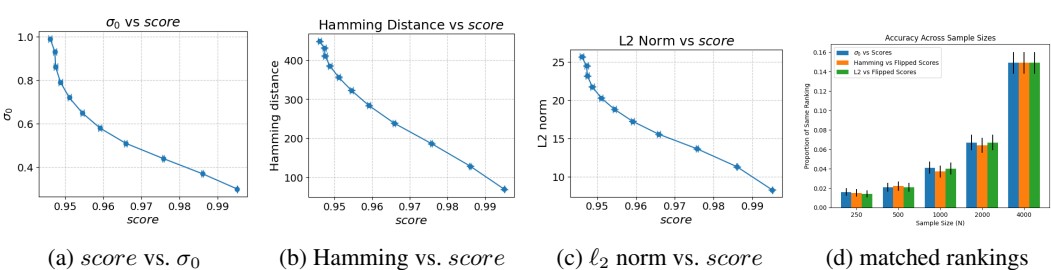

(a) $score$ vs. $\sigma_0$      (b) Hamming vs. $score$      (c) $\ell_2$ norm vs. $score$      (d) matched rankings

Figure 11: Experiments of plug-in Gram determinant reliability score with Gaussian kernel on Gaussian synthetic data with normal manipulation in Eq. (30).

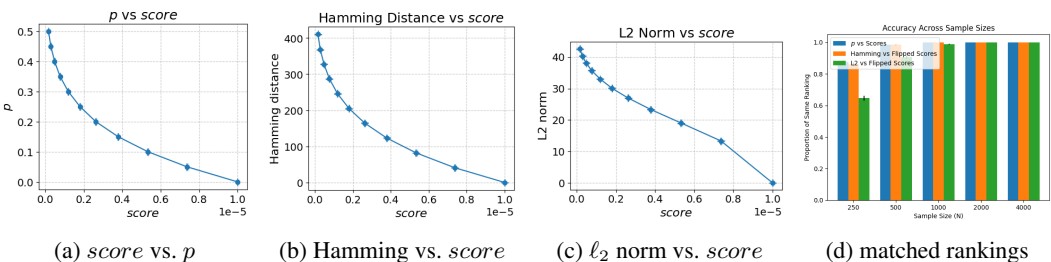

(a) $score$ vs. $p$      (b) Hamming vs. $score$      (c) $\ell_2$ norm vs. $score$      (d) matched rankings

Figure 12: Experiments of plug-in Gram determinant reliability score with delta kernel on Gaussian synthetic data with uniformly random manipulation in Eq. (6).

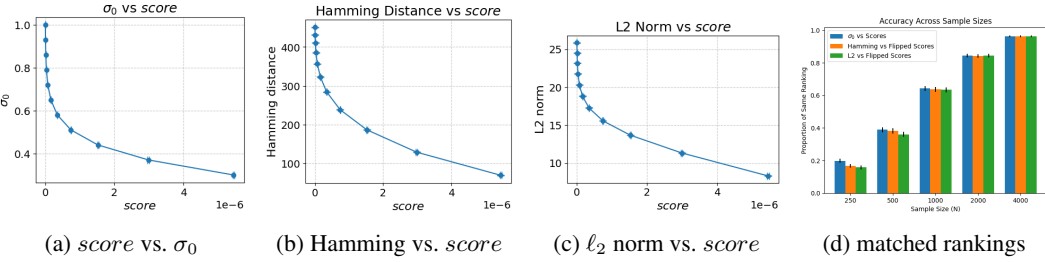

(a) $score$ vs. $\sigma_0$      (b) Hamming vs. $score$      (c) $\ell_2$ norm vs. $score$      (d) matched rankings

Figure 13: Experiments of plug-in Gram determinant reliability score with delta kernel on Gaussian synthetic data with normal manipulation in Eq. (30).

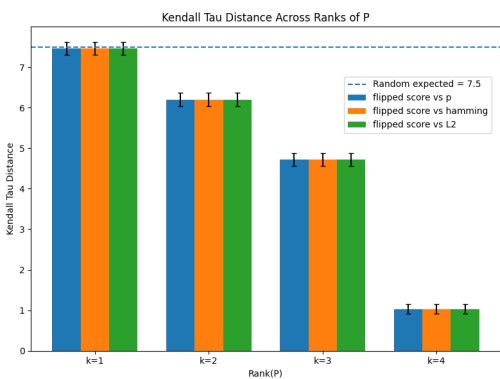 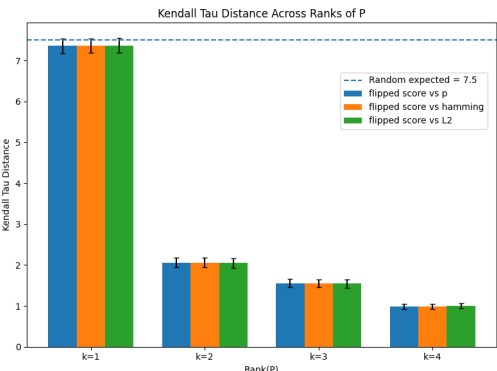

(a) Kendall Tau distance between reliability order and Gram determinant score.

(b) Kendall Tau distance between reliability order and Top-$k$ score ($k = 2$).

Figure 14: Comparison of robustness under rank-deficient conditional structures.

We run 300 independent trials. In each trial, we sample a ground-truth dataset $(\boldsymbol{x}, \boldsymbol{y})$ of size $N = 2000$ with $d = 4$ using the same procedure as in Experiment 1, except that the conditional distribution $\boldsymbol{P}(\cdot \mid x)$ is now determined by the rank-k matrix $\boldsymbol{P}$. To model varying corruption, for $p \in \{0.50, 0.60, \ldots, 1.00\}$ we use uniformly random manipulation to generate perturbed labels

$$\hat{x}_n = \begin{cases} x_n, & \text{with probability } p, \\ Z_n, & \text{with probability } 1-p, \end{cases} \qquad Z_n \sim \text{Uniform}\{1, \ldots, d\}.$$

For each corruption level, we compute the plug-in Gram determinant score and the top-k singular-value score and rank the six corrupted datasets accordingly, allowing us to assess whether each scoring function responds monotonically to increasing corruption even when $\boldsymbol{P}$ is rank-deficient.

From Figure 14a, we observe that even when the conditional distribution of $\boldsymbol{y}$ given $\boldsymbol{x}$ exhibits linear dependence, the Gram determinant score retains nontrivial discriminative power: datasets with higher corruption levels consistently yield lower scores, resulting in a meaningful correlation with the true reliability order. This indicates that the determinant-based score does not rely on full-rank structure in order to capture relative reliability.

When the linear dependence is strong (e.g., the experiment matrices $\boldsymbol{P}$ we construct are explicitly rank-deficient), a top-$k$ singular-value score becomes a natural alternative. As shown in Figure 14b, choosing k equal to the true underlying rank yields performance comparable to the Gram determinant score in the full-rank setting, while being substantially more stable under rank deficiency. This confirms that top-$k$ volume scores can better adapt to structured, low-rank conditional models, especially when only a subset of singular directions carries meaningful information.

### H.2.3 IMBALANCED DATA

In this experiment, we study how data imbalance in the marginal distribution of $\boldsymbol{x}$ affects the behavior of the Gram determinant score. This complements Experiment 1 by examining robustness not only to corruption mechanisms but also to skewed label frequencies, a common characteristic of real-world datasets.

The setting follows Experiment 1 exactly, except that the marginal distribution of $\boldsymbol{x}$ is now imbalanced: we draw $x_n = 1$ with probability $0.7$, and sample uniformly from $\{2, 3, 4, 5\}$ with probability $0.3$. The conditional distribution $\boldsymbol{y} \mid \boldsymbol{x}$ and all corruption schemes remain unchanged.

As shown in Figure 15, the Gram determinant score continues to separate datasets of different reliability levels even under substantial class imbalance. The monotonic decrease of the score with increasing corruption probability $p$ is preserved, and higher scores still correspond to lower Hamming error and smaller $\ell_2$ deviation. However, compared to the balanced case in Experiment 1, the score exhibits higher variance due to the reduced effective sample size in underrepresented classes.

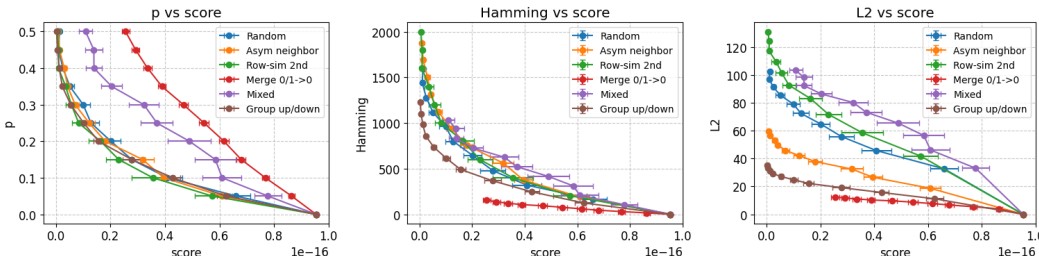

(a) Gram determinant score vs. corruption probability $p$.

(b) Hamming error vs. Gram determinant score.

(c) $\ell_2$ deviation vs. Gram determinant score.

Figure 15: Performance of the Gram determinant score under data imbalance.

This indicates that while the Gram determinant score remains informative under imbalance, larger dataset sizes are beneficial for stabilizing the estimate.

## H.3 ADDITIONAL DISCUSSION

While the proposed Gram determinant score shows strong empirical performance across both synthetic and real-world settings, several caveats deserve attention.

**Discretization versus kernelization.** In our synthetic experiments with Gaussian label distributions, we found that both the kernelized Gram determinant score (using a Gaussian kernel) and the regular Gram determinant score (based on bucketization on $\mathbf{y}$) performed similarly well, with no significant difference in effectiveness. This suggests that discretization, despite being a relatively crude approach, can sometimes work better than or similar to more elaborate kernel methods. In practice, however, not all datasets admit a natural discretization strategy. For example, in image datasets such as CIFAR-10, the lack of an intuitive discretization makes kernelized versions of the Gram determinant score particularly valuable.

**Assumptions about conditional independence in Experiment 3.** A key limitation of Experiment 3 is the reliance on the conditional independence assumption, which is difficult to validate in real-world applications. In practice, employment data may be indirectly adjusted from withheld tax records. In the unemployment dataset, we lack ground-truth employment data and only have access to three fiscal time series from which scores are computed. This prevents us from directly checking whether conditional independence holds. Consequently, the reported scores for these employment series should be interpreted only as indicative references, rather than definitive measures of reliability for formal or practical use.

**Comparison with alternative scores.** We compared the Gram determinant score to five existing scoring methods in Appendix G. All of them showed broadly consistent behavior: their rankings aligned well with Hamming distance and $\ell_2$-norm error. We also attempted to demonstrate the advantage of the Gram determinant score as an "experiment-agnostic" method. However, because we only had access to samples $\hat{\mathbf{x}}$ with corresponding $\mathbf{y}$, the underlying joint distribution matrix $\mathbf{PQ}$ was unknown, and any estimator we used introduced additional variance, the Gram determinant score could not exhibit a clear advantage in this regard. This limitation makes it more difficult to establish the clear superiority of our approach over the alternatives discussed in Appendix G, particularly in finite-sample regimes.

**Application of the Gram Determinant Score in Practice** Although verifying the formal conditions to preserve reliability orderings may be challenging in practice, several heuristics can offer guidance. Strongly imbalanced reported labels—for example, when one class is reported far more frequently than others—may fail to provide information for rare labels to reliably distinguish their observations. The conditional independence assumption is more credible when the observation is revealed only after reports (or kept blinded), so reporters cannot tailor reports to the observations. Persistently small determinants of the empirical Gram matrix may reflect poor reliability or weak

stochastic dependence between the reported data and observations. These diagnostics are not formal tests, but they offer practitioners useful signals about whether the theoretical requirements are plausibly satisfied in applied settings.