# OpenReview forum: "Data Reliability Scoring"
_ICLR.cc/2026/Conference — Submitted to ICLR 2026_

### Official Review · Reviewer_2ZZc · 2025-10-22

**Soundness:** 3
**Presentation:** 3
**Contribution:** 3
**Rating:** 6
**Confidence:** 2

**Summary:**

The paper addresses the problem of evaluating data reliability when ground-truth data are unobserved and only reported data and observations are available. The authors propose a reliability score, the Gram Determinant Score (GDS) defined as the determinant of the Gram matrix of observation distributions, capturing the information diversity among reported data. The paper shows that GDS preserves key reliability orderings and is independent of the observation process. Two estimators are proposed (plug-in and stratified matching), along with a GDS with kernels that generalizes the method to continuous or structured observation spaces.

**Strengths:**

1.	The paper provides theoretical results on preservation of reliability orderings (Exact, Blackwell, Hamming).
2.	The proposed score admits a geonetrical interpretation.
3.	The GDS with Kernel kernelized extends GDS for application to high-dimensional or structured data (e.g., feature embeddings).

**Weaknesses:**

1.	The results hold under restrictive assumptions.
2.	Estimation of Gram matrices can be expensive for large $d$ or $N$.
3.	The theoretical strengths of GDS might not easily turn into practical guidance.
4.	Comparaison with other metrics is not provided.

**Questions:**

1.	The paper does not provide a comparison of the proposed Gram Determinant Score (GDS) with existing reliability metrics. It would be informative to see how GDS performs relative to measures such as mutual information or other established scores.

2.	The computational complexity of the method is not evaluated. In particular, it is unclear whether the kernelized version can scale efficiently to large datasets or high-dimensional embeddings.

3.	In Experiment 2, the kernel is chosen as $K(y,y’) = \langle y,y’ \rangle$, but the paper does not justify this choice. Could alternative kernels affect the ranking or monotonicity of the score? Are there specific properties a kernel must satisfy for the method to remain valid? Furthermore, which kernel choices are most appropriate for real-world structured data, such as images, text, or signal embeddings?

4.	It is unclear whether the Gram Determinant Score can be normalized across datasets to allow direct comparisons of reliability between heterogeneous datasets. For practical applications, it would also be useful to establish whether a meaningful threshold exists above which a dataset can be considered “reliable.” Clarifying these points would enhance both the interpretability and the practical applicability of the method.

This work makes a conceptually novel and theoretically elegant contribution to the study of data reliability by introducing a unified geometric measure that encompasses several classical reliability orderings. While the theoretical results are compelling and the determinant-based approach is original, the study would benefit from further empirical validation and a more thorough investigation of its scalability and robustness in practical, real-world settings.

---

> ### Author Response · Authors · 2025-11-24
>
> **1. On the linear dependence condition of $\mathbf P$**
>
> Please refer to the [first point in our response](https://openreview.net/forum?id=b6miYNcjag&noteId=dcPPl09b9d) to review JUwe.
>
>
>
> **2. Comparison with existing reliability metrics**
>
> Please refer to the [seventh point in our response](https://openreview.net/forum?id=b6miYNcjag&noteId=aipvQfMDCH) to review YGfJ
>
> **3. Computational complexity of the proposed method, especially the kernalized version.**
>
> Given the empirical frequency matrix $\bar{\mathbf{G}} \in \mathbb{R}^{d \times d}$, we can compute its determinant via Gaussian elimination in $O(d^3)$ time. In addition, Han, Malioutov, and Shin (2015) provide an approximation algorithm that runs in $O(d^2)$. Hence, the computational cost remains manageable even for large $d$.
> When the linear-independence condition is violated, we may instead use the top-$k$ volume score (described in Appendix G.1), which corresponds to taking a low-rank approximation. We can further use SVD to compute the top $k$ eigen values, and hence the top-$k$ volume score, in $O(d^3)$ times.
> The size of the dataset only affects the computation of the empirical distribution, which can be done in linear time $O(N)$.
>
> For the kernelized version, scalability is preserved because the kernel Gram matrix remains $d \times d$. High-dimensional observation spaces influence only the cost of evaluating the kernel (which is precisely the strength of kernel methods), not the subsequent determinant computation. Moreover, we can accelerate the estimation of the empirical Gram matrix by subsampling reports.
>
> **4. The Kernel choice in Experiment 2**
>
> In our CIFAR-
> 10 experiment, we choose the linear kernel $K(\mathbf{y},\mathbf{y}')=\langle \mathbf{y}, \mathbf{y}' \rangle$ because the raw ${\mathbf y}$ values are continuous and extremely high-dimensional. We adopt this commonly used linear kernel to map such observations into a more
> manageable discrete space while retaining the relevant structural information. Kernalization is especially suitable when there is no natural discretization strategy for the space of $\mathbf y$, such as in our CIFAR-10 experiment. We have more discussion on this in Appendix H.3.
>
> For our experiment 1, Appendix H.2.1 of our submission has compared the performance of the delta kernel and the Gaussian kernel on datasets where ${\mathbf y}$ is generated either from Gaussian distributions or from random experiments. We observe that the Gaussian kernel performs noticeably better when the observation ${\mathbf y}$ is Gaussian, particularly in terms of the probability of correct rank alignment. Thus, although different kernels yield the same expected ranking, the *variance* of the score estimator can be significantly reduced by using a kernel that better matches the underlying data-generating process. In practice, if partial information about the experiment is available, one can design kernels that better capture the structure of ${\mathbf y}$. The precise conditions required for optimal kernel design remains an open question.
>
>
> **5. Whether the Gram Determinant Score can be normalized across datasets to allow direct comparisons of reliability between heterogeneous datasets**
>
> Please refer to the [fourth point in our response](https://openreview.net/forum?id=b6miYNcjag&noteId=oT4vUxrRdg) to review JUwe.

---

### Official Review · Reviewer_YGfJ · 2025-10-31

**Soundness:** 3
**Presentation:** 2
**Contribution:** 3
**Rating:** 2
**Confidence:** 3

**Summary:**

The paper introduces a way to score how reliable different reported datasets are. It uses extra signals or observations that are related to the true labels and uses them to tell which dataset is closer to the truth. It formalizes how to say that dataset A is closer to the true data than dataset B, then defines a new score called the Gram Determinant Score to capture that. The score is computed from the joint information between the reported labels and the extra signals, and under some conditions it gives the same ranking no matter which observation process you used. Experiments on synthetic data, CIFAR-10 style image embeddings, and on an employment dataset show that the score goes down when the labels are corrupted, so it can detect which datasets are better and which are noisier.

**Strengths:**

- The problem is clearly set up. They say what it means for one reported dataset to be better than another and connect it to standard ideas like Blackwell orderings.

- The method has an intuitive picture. Clean data gives a Gram matrix with a bigger determinant. Noisy data makes it smaller.

- The main theorem is strong. If the observation processes are linearly independent, this score gives the same ranking across them and is basically unique.

**Weaknesses:**

- The strongest results need linearly independent observation processes and some structure on the reporting noise. In real data those conditions may not hold exactly.

- The score is defined to preserve certain orderings, but in practice you never see those orderings because you never see the true labels. So it can be hard to know if it is right for your case.

- Determinants of Gram matrices can be numerically small and unstable, especially in higher dimensions.

- The experiments are convincing but still mid scale. We do not see behavior for very large label spaces or very heavy class imbalance.

- There is not a wide empirical comparison with other label quality or data pruning methods, so it is hard to tell how big the gain is.

**Questions:**

- How sensitive is the Gram Determinant Score if the observation process is not perfectly linearly independent but only close to it? Can you show a robustness curve?

- In practice how many samples do we need before the score ranking is stable, especially with many classes?

- How should we pick the kernel and its parameters in the embedding setting?

- Can the score be fooled if a subset of reports is adversarial or all collapsed to a popular class?

- What is the computational cost for large d and can we use low rank or randomized approximations?

- Did you compare to simpler agreement or mutual information based scores on the same datasets?

---

> ### Author Response · Authors · 2025-11-24
>
> **1. On the linear dependence condition of $\mathbf P$**
>
> Please refer to our [first response](https://openreview.net/forum?id=b6miYNcjag&noteId=dcPPl09b9d) to review JUwe.
>
> **2. The score is defined to preserve certain orderings, but in practice you never see those orderings because you never see the true labels.**
>
> We agree that true labels are often unavailable—this is precisely why a theoretically grounded reliability measure is valuable. The order-preservation guarantee of GDS ensures that less reliable reports are never rated higher than more reliable ones, even without ground truth. Thus, GDS offers a robust comparative notion of reliability. In our unemployment-rate experiment, it effectively distinguishes report quality without any access to true labels, demonstrating that the resulting comparisons are both meaningful and actionable.
>
>
> **3. How many samples are needed before the score ranking becomes stable (sample complexity)?**
>
> We provide additional theoretical and experimental results for this question in our updated paper.
>
> On the theory side, we add Proposition E.4 (Appendix E) which provides sample-complexity bounds for both multiplicative and additive errors of the plug-in Gram determinant score. For the multiplicative error, when $\hat{\mathbf G}$ is invertible, the required sample size is polynomial in all relevant parameters:
> $
> O\left(\frac{d^{2}\log(d/\delta)}{\lambda^{2}\epsilon^{2}}\right),
> $
> where $\lambda$ denotes the smallest eigenvalue of $\hat{\mathbf G}$ (recall that $\hat{\mathbf G}$ is positive semidefinite). For the additive error, we obtain an even stronger bound,
> $
> O\left(\frac{d^{2}\log(d/\delta)}{\epsilon^{2}}\right),
> $
> which holds for arbitrary $\hat{\mathbf G}$ and is independent of its minimum eigenvalue.
>
> Empirically, we further evaluate stability using the Kendall-Tau distance between the GDS reliability ranking and other ground-truth-based orderings in Figure 2(d).
> When the sample size reaches $8000$, for ranking $6$ datasets, the Kendall Tau distance between the GDS reliability ranking and any of the three ground-truth-based rankings is on average $1$, with very small standard deviation---indicating stability of the ranking produced by the GDS.
>
> **4. How should we pick the kernel and its parameters in the embedding setting?**
>
> First, Appendix H.2.1 of our submission has compared the performance of the delta kernel and the Gaussian kernel on datasets where $\mathbf y$ is generated either from Gaussian distributions or from random experiments. We observe that the Gaussian kernel performs noticeably better when the observation ${\mathbf y}$ is Gaussian, particularly in terms of the probability of correct rank alignment. Thus, although different kernels yield the same expected ranking, the *variance* of the score estimator can be significantly reduced by using a kernel that better matches the underlying data-generating process. In practice, if partial information about the experiment is available, one can design kernels that better capture the structure of ${\mathbf y}$. The precise conditions required for optimal kernel design remains an open question.
>
> Second, kernels also serve as an effective tool for dimensionality reduction. For example, in our CIFAR-10 experiment, we choose the linear kernel $K(\mathbf{y},\mathbf{y}')=\langle \mathbf{y}, \mathbf{y}' \rangle$ because the raw ${\mathbf y}$ values are continuous and extremely high-dimensional. Kernelization provides a principled way to map such observations into a more manageable representation while retaining the relevant structural information. This is especially suitable when there is no natural discretization strategy for the space of $\mathbf y$, such as in our CIFAR-10 experiment. We have more discussion on this in Appendix H.3.

---

> ### Author Response · Authors · 2025-11-24
>
> **5. Can the score be fooled if a subset of reports is adversarial or all collapsed to a popular class?**
>
> Our theoretical results (Theorem 4.2) show that this is not possible. These manipulations --- manipulating a subset of reports in an adversarial way or collapsing multiple classes into a single class --- fall exactly within our Blackwell ordering, which captures post processing operations. An adversarial agent who observes the true report $\mathbf{x}$ and chooses to always misreport class label $j$ as label $i$ is represented by $\mathbf{Q_{\hat{x}|x}}(i,j) = 1$ and $\mathbf{Q_{\hat{x}|x}}(j,j) = 0$. This is a post processing on the true data $\mathbf{x}$.
> More complex adversarial manipulations, such as partial misreporting, are also post-processing.  Further, the Blackwell ordering is closed under the composition of such post-processing. Thus, our theorem guarantees that the manipulated data receives a lower expected GDS than the unmanipulated data.
>
>
> Empirically, this was also confirmed in our experiments. In both Experiment 1 and Experiment 2 in Section 5, we evaluate the case where classes $0$ and $1$ are merged into a single class $0$, which represents an extreme form of collapsing into a popular category. In all settings, the Gram determinant score reliably detects this manipulation and assigns a lower score to the corrupted dataset.
>
> **6. What is the computational cost for large $d$ and can we use low rank or randomized approximations?**
>
> Given the empirical frequency matrix $\bar{\mathbf{G}} \in \mathbb{R}^{d \times d}$, we can compute its determinant via Gaussian elimination in $O(d^3)$ time. In addition, Han, Malioutov, and Shin (2015) [1] provide an approximation algorithm that runs in $O(d^2)$. Hence, the computational cost remains manageable even for large $d$.
>
> When the linear-independence condition is violated, we may instead use the top-$k$ volume score (described in Appendix G.1), which corresponds to taking a low-rank approximation. We can further use SVD to compute the top $k$ eigen values, and hence the top-$k$ volume score, in $O(d^3)$ times.
>
> **7. Comparison to simpler agreement or mutual information based scores on the same datasets**
>
> We add additional comparisons between GDS and other candidate metrics (top-$k$ volume score, maximal correlation, $\chi^2$-mutual information, and KL divergence) for datasets degraded by one of six types of manipulations in Appendix G. Figure 4 (the results for GDS are in Figure 2) shows that all metrics more or less preserve the ground-truth-based ordering across datasets manipulated by the same method, with the exception of maximal correlation which fails to order datasets manipulated by the merge $0/1\to 1$ strategy. However, for datasets degraded by different types of manipulations, maximal correlation, $\chi^2$-MI, and KL divergence all have cross-manipulation inconsistencies: when two datasets are manipulated by different methods, they may assign a higher score to the dataset that is further away from the truth according to Hamming distance, a violation to Hamming distance ordering.  In contrast, our GDS consistently preserves the Hamming distance ordering across all six types of manipulation.
>
> We additionally report the experiment-agnostic satisfying rate for all metrics in Figure 5. The GDS achieves a substantially higher rate than the other four metrics ($0.963$ vs. $0.78 - 0.85$). This is consistent with our theoretical result in Proposition 4.3 that GDS (when computed exactly) is the unique reliability score whose ranking of datasets is independent of the choice of the experiment $\mathbf y$.
>
> **8. The experiments...very heavy class imbalance.**
>
> In Appendix H.2.3, we report additional experiments on GDS for datasets where the distribution of labels is skewed. Under imbalanced data, the variance of GDS increases, and more data is needed for a stable ranking to be achieved, while the general ranking is largely preserved.
>
>
> [1]: Han, Insu, Dmitry Malioutov, and Jinwoo Shin. "Large-scale log-determinant computation through stochastic Chebyshev expansions." International Conference on Machine Learning. PMLR, 2015.

---

### Official Review · Reviewer_JUwe · 2025-11-02

**Soundness:** 3
**Presentation:** 3
**Contribution:** 3
**Rating:** 6
**Confidence:** 2

**Summary:**

This paper tackles the novel problem of reliability scoring: assessing dataset quality without ground truth. The authors formalize this setting for data collected from potentially noisy or strategic sources and define several ground-truth–based reliability orderings (Exact Match, Blackwell, Hamming/Distance) as benchmarks for evaluating reliability metrics.

They show fundamental impossibility results, proving that no score can universally preserve all such orderings. To overcome this, the authors propose the Gram Determinant Score (GDS), a geometric measure that quantifies the “volume” spanned by observation distributions — smaller volumes indicate greater deviation from truth. The GDS enjoys strong theoretical properties: it preserves key reliability orderings under mild conditions, is experiment-agnostic (ranking consistency across observation mechanisms), and generalizes naturally to continuous domains via kernelization.

Experiments on synthetic categorical data, CIFAR-10 embeddings, and U.S. employment statistics confirm that the GDS correlates well with ground-truth reliability and consistently ranks more trustworthy datasets higher.

**Strengths:**

The paper formally introduces the setting of reliability without ground truth, which is both theoretically interesting and practically relevant for data collected from uncertain or biased sources (e.g., social, economic, or self-reported data).

GDS has a clean intuition: it measures the “volume” of observation distributions, which naturally shrinks as the data deviate from truth. This connects statistical structure to an interpretable geometric property.

Experiments cover both synthetic (controlled corruption), vision (CIFAR-10 embeddings), and real-world (employment data) domains, demonstrating the score’s consistency and practical usability.

**Weaknesses:**

The mathematical framing may be too abstract for practical deployment; real-world users might find the link between Q,P, and reliability difficult to interpret or estimate.

The theoretical results rely on independence and linearity assumptions in the observation model P; it is unclear how robust the score remains when these are violated.

While the three experiments are convincing, all datasets are relatively small-scale or well-structured; results on larger, noisier real-world datasets (e.g., survey or crowd-sourced data) would strengthen claims of generality.

**Questions:**

Can GDS be intuitively understood as a variance or entropy measure over observation space? How might practitioners interpret a “high” or “low” Gram Determinant Score in practical settings?

In settings where the observation process P is unknown or partially known (e.g., in survey data), how could one practically estimate or approximate P for computing GDS?

---

> ### Author Response · Authors · 2025-11-24
>
> **1. How to compute GDS when the observation process $\mathbf P$ is unknown or partially known**
>
> A key advantage of the Gram Determinant Score (GDS) is that it does not require knowledge of the underlying stochastic experiment $\mathbf P$ or the misreporting matrix $\mathbf Q$ to estimate. While Section 4.1 uses $\mathbf P$ and $\mathbf Q$ to analyze and interpret the theoretical properties of GDS, the estimators defined in Sections 4.2 and 4.3 --- which preserve the same reliability orderings --- take as inputs only the reported data $\hat{\mathbf x}$ and observations $\mathbf y$. The intuition is that GDS depends only on the product $\mathbf P \mathbf Q$, which represents the joint distribution of the reported data and the observations of the experiment, a quantity that can be directly and efficiently estimated from the empirical joint frequency of $\hat{\mathbf x}$ and $\mathbf y$.
>
>
> **2. On the linear dependence condition of $\mathbf P$**
>
> First, we emphasize that linear independence of columns of $\mathbf P$ (i.e. $\mathbf P$ having full column rank) is a characterization of when reliability scoring is possible, not an assumption that we impose. Proposition 3.1 formally shows that if P is rank-deficient, no score—ours or any other—can preserve the Blackwell ordering. Thus, the condition is inherent in the structure of the problem rather than specific to our method.
>
> Second, this linear independence condition is very mild. When the observation space $\mathcal{Y}$ is high-dimensional (e.g., composed of multiple measurements or signals with real values), collinearity among the columns of $\mathbf P$ is highly unlikely---the set of linearly dependent matrices has Lebesgue measure zero. In practice, one can design or augment the observed variable $y$ to increase informativeness. Each $y$ may be vector-valued, allowing practitioners to combine multiple partial measurements. For instance, in the COVID-19 example, combining death counts, ambulance-call records, and hospital occupancy rates into a composite $y$ effectively restores rank.
>
> Third, we include additional empirical evaluation in Appendix H.2.2 of our updated paper. The first table below and Figure 14(a) report the *Kendall Tau distance* between the GDS induced ranking of the $6$ datasets and several ground-truth-based orderings of the datasets when ${\mathbf P}$ is explicitly rank-deficient. We see that the low column rank of $\mathbf P$ indeed increases the Kendall Tau distance significantly, indicating inaccurate ranking. However, a top-$k$ volume score (inspired by the GDS and described in Appendix G.1) provides a natural alternative. As shown in the second table below and Figure 14(b), choosing $k$ equal to the underlying rank yields a performance close to the full-rank GDS while remaining stable under rank deficiency. This demonstrates that top-$k$ volume score adapts well to structured low-rank settings.
>
> **Gram determinant score: Kendall Tau distance (mean ± s.e.)**
>
>  | Rank k | Score vs p     | Score vs Hamming | Score vs L2     |
>  |--------|----------------|------------------|------------------|
>  | 1      | 7.46 ± 0.16    | 7.46 ± 0.16      | 7.46 ± 0.16      |
>  | 2      | 6.20 ± 0.17    | 6.20 ± 0.17      | 6.20 ± 0.17      |
>  | 3      | 4.72 ± 0.16    | 4.72 ± 0.16      | 4.72 ± 0.16      |
>  | 4      | 1.03 ± 0.13    | 1.03 ± 0.13      | 1.03 ± 0.13      |
>
>
>  **Top-k volume score: Kendall Tau distance (mean ± s.e.)**
>
>  | Rank k | Score vs p     | Score vs Hamming | Score vs L2     |
>  |--------|----------------|------------------|------------------|
>  | 1      | 7.36 ± 0.18    | 7.36 ± 0.18      | 7.37 ± 0.18      |
>  | 2      | 2.06 ± 0.12    | 2.06 ± 0.12      | 2.05 ± 0.12      |
>  | 3      | 1.56 ± 0.10    | 1.55 ± 0.10      | 1.54 ± 0.10      |
>  | 4      | 0.99 ± 0.06    | 0.98 ± 0.06      | 1.01 ± 0.06      |

---

> > ### Author Response · Authors · 2025-11-24
> >
> > **3.Can GDS be intuitively understood as a variance or entropy measure over observation space?**
> >
> > The GDS quantifies the diversity of the observation distributions associated with different reported types. Intuitively, when reports are close to true data, each type induces a distinct pattern of observations, which results in a large volume of the joint frequency matrix $\mathbf P \mathbf Q$.
> >
> > This contrasts with Shannon mutual information (or the more general $\Phi$-mutual information defined in Appendix G.1), which measures deviations of the joint distribution from the product of marginal distributions. It is less directly tied to the structural distinctiveness of the observation profiles across reported types. For example, if an agent deterministically merges two underlying classes into one class in the reported dataset, the GDS will assign $0$ to the reported dataset because the resulting Gram matrix loses full rank. Mutual information, however, is less sensitive to such manipulation and may still assign a positive score for the reported dataset. In Appendix G.2, we provide more experimental results comparing GDS and a few other metrics including mutual information.
> >
> > **4. How might practitioners interpret a high or low Gram Determinant Score in practical settings?**
> >
> > Our current model assumes no knowledge of the experiment $\mathbf P$ and data manipulation $\mathbf Q$, and we try to understand what is possible in this most challenging setting. The magnitude of the GDS depends jointly on $\mathbf P$ and $\mathbf Q$ and therefore does not admit a universal threshold across heterogeneous datasets for determining whether a dataset is reliable.
> >
> > We think it is an important and promising future direction to explore, with additional knowledge of $\mathbf P$ or $\mathbf Q$ (which is available in many domains), how to develop reliability measures that have universal interpretation across heterogeneous datasets. For instance, for the binary symmetric example in Appendix D.1, if we know that the gap in observation probabilities satisfies $p_1-p_2> \rho$, then the GDS provides a quantitative bound on the misreporting rate $1-2\delta = \Theta(\frac{\sqrt{GDS}}{\rho})$.

---

### Author Response · Authors · 2025-11-24
**General Response**

We thank all reviewers for their feedback!

The main contribution of our paper is theoretical. We formulate the problem of data reliability without ground truth, prove fundamental impossibility results, and introduce the Gram Determinant Score (GDS) as a principled solution. GDS is the first reliability measure shown to preserve multiple ground-truth–based orderings and to be experiment-agnostic, a unique property we establish formally. The experiments serve primarily to illustrate and support the theoretical predictions rather than to exhaust empirical evaluations. A natural future direction is to conduct large-scale experiments in multiple domains to extensively evaluate the empirical performance of the proposed method.

A common point raised by all reviewers concerns the linear independence condition of ${\mathbf P}$. We emphasize that this condition is a characterization, not an imposed assumption, and it is not restrictive given the flexibility to choose observations $\mathbf{y}$. In the updated paper, we provide additional experiments to demonstrate the robustness of GDS to violations of this condition, as well as some new theoretical and empirical results addressing reviewers' specific comments.

We provide details on these points in our responses to individual reviews below. All changes are highlighted in blue in the updated paper.

---

### Meta-Review · Area_Chair_mpHN · 2025-12-31

**Summary:**

Relability scoring without the ground truth is a non-trivial, but practical challenge. The authors introduce Gram determinant score as a metric for the volume of class-conditional observation distributions. Under mild assumptions, it preserves exact-match and Blackwell orderings and closely approximates Hamming orderings. The strengths include:

- The problem is clearly presented.

- The geometric intuition: Clean data provides a Gram matrix with a bigger determinant. Noisy data makes it smaller.

- The main theorem: If the observation processes are linearly independent, this score gives the same ranking across them and is basically unique.

Unfortunately, the reviewers were not enthusiastic, with the initial scores of 2, 6 and 6.

**Reviewer Concerns:**

Several key concerns proved impossible to address during the rebuttal:

- the mild assumption remains poorly explained and difficult to test on real data.

- the experiment design is poor. None of the experiments uses strategic responses. Indeed, wow does CIFAR aid in illustrating robustness to strategic responses?

- Claims of experiment agnosticism are purely based on limited empirical evidence, although could perhaps be substantiated using the Johnson–Lindenstrauss lemma?

- comparison with other metrics is not provided.

**Reviewer Scores:**

I doubt that the reviewers would change their scores, based on the terse rebuttal.

---

### Decision · Program_Chairs · 2026-01-26

Reject